# Expected effects of a global transformation of agricultural pest management

Niklas Möhring [1] ✉, Malick N. Ba[2], Anna Rafaela Cavalcante Braga[3], Sabrina Gaba[4], Vesna Gagic[5], Per Kudsk [6], Ashley Larsen [7], Robin Mesnage [8], Urs Niggli[9], Matin Qaim [10], Pepijn Schreinemachers[11], Christian Stamm [12], Wim de Vries [13] & Robert Finger [14] ✉

Ambitious policy goals to reduce pesticide use and risks have been established at global and regional levels. Here, we provide an assessment of the expected effects of such a global transformation of agricultural pest management. We develop a holistic assessment framework covering economic, human health, food security, social, and environmental effects and conduct a global survey with 473 experts from key disciplines and major agricultural production regions. This is an important step to identify leverage points for advancing pesticide policies and focusing future research efforts. Our results demonstrate that transforming agricultural pest management could be an important nexus for addressing multiple sustainability challenges. We find the highest expected benefits for the environmental and human health domains and the lowest for the economic and food safety domains. For regions with low income and low pesticide use, we find higher benefits and less trade-offs of the transformation than for intensive production systems in Europe and North America. Finally, a transformation is not free of costs and our results indicate that it will require a combination of new and locally adapted pest management solutions, research and support for their implementation on the ground, and an enabling policy environment.

Most agricultural crops are affected by a broad range of animal pests, plant diseases, and weeds, which can lead to significant crop losses in terms of quantity and quality. Despite current efforts in pest management, average yield losses due to pests and pathogens are substantial. They are estimated to vary between 26% and 40% (ref. 1; based on literature and field trial data) and between 17% and 30% (ref. 2; based on expert assessments) for the five major food crops (wheat, rice, maize, potato, and soybean) worldwide with a large variation across regions. Further, quality losses can be important[3]. Effective management of agricultural pests is thus essential for farmers' livelihoods and for producing sufficient and diverse food for a growing population on a limited amount of land. The current pest management systems mainly rely on the use of pesticides with a global consumption of 3.7 million tons in 2023[4]. We here specifically focus on pre-harvest pesticides in agriculture, which typically provide affordable and efficient pest management solutions that can be applied in a wide range of

[1]Production Economics Group, University of Bonn, Bonn, Germany. [2]World Vegetable Center, Cotonou, Benin. [3]Department of Chemical Engineering, Universidade Federal de São Paulo, São Paulo, Brazil. [4]USC 1339 Centre d'Etudes Biologiques de Chizé- Résilience, INRAE CNRS La Rochelle Université, Villiers-en-Bois, France. [5]Department of Agriculture and Fisheries Queensland, Brisbane, QLD, Australia. [6]Department of Agroecology, Aarhus University, Flakkebjerg, Denmark. [7]Bren School of Environmental Science & Management, UC Santa Barbara, Santa Barbara, CA, USA. [8]Department of Medical & Molecular Genetics, Kings's College London, London, UK. [9]Institute of Agroecology, Aarau, Switzerland. [10]Center for Development Research (ZEF), University of Bonn, Bonn, Germany. [11]World Vegetable Center, Bangkok, Thailand. [12]Eawag, Swiss Federal Institute of Aquatic Science and Technology, Dübendorf, Switzerland. [13]Environmental Systems Analysis Group, Wageningen University and Research, Wageningen, The Netherlands. [14]Agricultural Economics and Policy Group, ETH Zurich, Zurich, Switzerland. ✉e-mail: mohring@uni-bonn.de; rofinger@ethz.ch

production contexts[5]. However, pesticides can adversely affect biodiversity as well as ecosystem functioning and services (e.g., refs. 6–8). Ecosystem services such as natural pest control[9–11], insect-pollination[12–14] and soil functions[15] contribute to agricultural production. Additionally, pesticides can have adverse health effects, especially on farmers and bystanders[8,16–18]. Finally, resistance to widely used active ingredients are increasing, narrowing available chemical pest management strategies[19].

In the future, the importance of effective pest management will further increase with elevated global food demand, changing consumption patterns, and expected changes in the distribution and abundance of insect pests and plant pathogens under climate change (e.g., refs. 20–24).

In response to health and environmental concerns, substantial policy targets for pesticide reduction have been set on policy agendas globally. For example, the Post-2020 Global Biodiversity Framework of the United Nation sets a reduction target of 50% for overall pesticide risks by 2030[25]. The European Union member states further aim to reduce "the use and risk of chemical pesticides and the use of more hazardous pesticides" by 50% (see ref. 26 for an overview and discussion of EU policy targets). Both emphasize the reduction of environmental and health risks from pesticide use (see refs. 25,27,28 for extensive discussions on this question).

For decades, agricultural production systems worldwide have been promoted in which pesticides are partially or entirely substituted with other means of pest control, e.g., biological, mechanical, breeding, or agronomic solutions. This includes integrated pest management[29–32], agroecological crop protection[33] and organic agriculture[34,35]. However, global efforts to reduce and substitute pesticide use remain fragmented, lacking a coherent strategy to mitigate pesticide pollution and adverse effects[36]. Achieving the targets requires a shift in policies. There is also little evidence of significant progress towards achieving pesticide risk and use reduction targets (e.g., refs. 26,37), with exceptions such as Denmark[38]. On the contrary, studies report high levels[39,40] and even increasing trends (e.g., refs. 27,41,42) in the toxicity and risk of pesticides used in a number of major agricultural regions worldwide.

A central question for research, policy, and industry is how to effectively and efficiently support pesticide risk reduction, while minimizing crop loss due to pests. In order to establish priorities and to design effective and efficient reduction strategies—an understanding of the current status, potential effects, and trade-offs of a pest management transformation is required. However, data on the potential effects of a transformation are scarce and fragmented despite the high relevance of this question. Literature discusses only specific cases of pests, pest management practices, production systems, or regions (Table 1). A few studies exist on a European level (see refs. 43–47 for an overview). Some of these studies have been criticized because, (i) they are based on estimates of a small number of experts or results from a limited number of field experiments under controlled conditions, and because they not sufficiently consider (ii) potential environmental spillover effects of pesticide reduction on agricultural production, (iii) potential improvements in efficiency of crop health management, (iv) farmer's decision-making, (v) the availability of novel substitutes or means to improve pesticide use efficiency, or (vi) the wide range of context-specific costs, benefits and trade-offs of pesticides and their substitutes (e.g., ref. 47). This limits their ability to holistically compare potential effects across the different domains of impact and global regions of production.

Here, we provide a global, holistic assessment of the effects expected from transforming agricultural pest management. We develop an assessment framework for the transformation of global pest management covering all relevant dimensions of sustainability, i.e., economic, human health, food security, social, and environmental. We then identify 24 indicators to assess the effects and trade-offs of a

transformation of global pest management systems and verify the choice of indicators in several steps (see the "Methods" section). Importantly, we also include indicator domains and single indicators in which the literature has previously set clear expectations regarding the direction of effects (e.g., refs. 2,40,48). We then conduct a global online survey (in English, Chinese, Spanish, Portuguese, French, and German, covering global experts) to obtain expert assessments of expected changes. We compare changes in indicators from a baseline scenario to a *sustainable pest management* scenario with no or minimal use of pesticides, but alternative pest management practices. We employ a robust, multi-stage procedure to design, test, implement, and analyze our global survey. Large-scale global expert surveys have, for example, previously been employed to assess obstacles for integrated pest management in developing countries[30], the state of global plant health[49] to estimate global yield losses from pests[2] or to determine suitable global carbon prices[50]. We assess the expected direction of the effects at the global level (negative or positive), heterogeneity of effects across indicators and regions worldwide and analyze the potential drivers of the expected effects. We obtain responses from 473 respondents, who are mainly senior scientists from leading institutions in the field worldwide, and cover key disciplines, global food production hotspots and countries with a combined share of 88%, 85%, and 90% of global cereal, fruit, and vegetable production, respectively (see supplementary note 2). We find that a transformation of global pest management could contribute to multiple sustainability challenges. However, expected effects are heterogeneous across global regions and a transformation is not free of costs. We assess drivers and adequate policy bundles for a transformation and discuss their implementation.

## Results
### Expected direction of the global effects
We lack global estimates and regional heterogeneity data for all 24 key indicators considered in our study (see Table 1). Therefore, prior expectations regarding the sign and importance of effects per indicator are limited and very context-specific. In general, data on pesticide use and pest management practices are scarce (e.g., ref. 51). Existing studies typically focus on individual pesticides or alternatives for individual pests, pathogens, or crops at the regional or case study level. Furthermore, the existing assessments mainly highlight the potential costs of the reduction of pesticides. But they fail to quantify the potential of alternative pest management practices to mitigate these costs (e.g., refs. 40,48). This aspect is crucial, because pesticides are often effectively controlling pests and a reduction of pesticides must be accompanied by the implementation of alternative measures of pest management to protect crops. Thus, essential information for the identification of priorities, trade-offs, and synergies and the definition of strategies for the advancement of sustainable pest management is currently missing.

For the domains of environmental toxicity and the pollution of surface water and soils by pesticides, quantitative estimates of pesticide risks exist for several aggregation levels, including global assessments (e.g., refs. 40,48). But even studies in this domain typically do not provide estimates of the pollution reduction potential of alternative pest management practices. Further, the literature documents negative effects of pesticides on human health across all indicators in the domain. However, examining the scale and scope of the effects on human health is more challenging (for ethical reasons) and is mainly reliant on epidemiological studies (see refs. 8,17,18 for exceptions). Thus, the extent of these effects is less clear for human health than in the environmental domain. The availability of different active substances, the requirements for pesticide application and protection, and the enforcement of rules largely differ across countries (e.g., refs. 52,53). Furthermore, pesticide residues in food are found to be mostly below maximum residue levels in regions with strict standards,

**Table 1 | Key indicators, references, and expected directions of effects (from survey responses) of a global agricultural pest management transformation**

| Labels | Indicators | Descriptions | Key references | Direction of expected global effects |
|---|---|---|---|---|
| **Domain 1: environment** | | | | |
| ENV1 | Drinking water | Pesticide pollution of drinking water | 98,99 | +++ |
| ENV2 | Soils | Pesticide pollution of soils | 15,40,100 | +++ |
| ENV3 | Marine ecosystems | Pesticide damage to marine ecosystems | 101,102 | ++ |
| ENV4 | Freshwater ecosystems | Pesticide damage to freshwater ecosystems | 48,101 | +++ |
| ENV5 | Biodiversity loss | Contribution of pesticides to biodiversity loss | 6–9,103,104 | +++ |
| **Domain 2: human health** | | | | |
| HH1 | Health of farm workers | Acute and long-term effects on the health of farm workers | 52,105,106 | +++ |
| HH2 | Health of residents | Acute and long-term effects on the health of residents and bystanders | 8,16–18 | ++ |
| HH3 | Health of consumers | Acute and long-term effects on the health of consumers | 107,108 | ++ |
| HH4 | Healthy food choices | Contaminated food that distorts food choices (e.g., fewer vegetables or fruits) | 59 | +++ |
| **Domain 3: food and nutrition security** | | | | |
| FS1 | Food quantity | Provision of food in sufficient quantity | 1,2,109 | + |
| FS2 | Healthy diets | Provision of diverse and nutritious food | 1,2,110 | ++ |
| FS3 | Safe food | Provision of safe food and feed | 54,55,111 | +++ |
| FS4 | Resilience hunger | Resilience to extreme pest events resulting in hunger and food insecurity | 21,112 | + |
| **Domain 4: economic prosperity and livelihood of farmers** | | | | |
| ECON1 | Cost-efficient | Provision of affordable and efficient pest management solutions | 30,113–115 | + |
| ECON2 | Short-run farmer income | Enabling production systems that can sustain the livelihood of farmers in the short run | 29,115,116 | + |
| ECON3 | Long-run farmer income | Enabling production systems that can sustain the livelihood of farmers in the long run | 12,15,56,57 | ++ |
| ECON4 | Economic resilience | Economic resilience to extreme pest events | 21,117 | + |
| ECON5 | Productivity growth | Agricultural productivity growth (threatened by pest damages) | 2,21,24,56 | ++ |
| ECON6 | Indirect costs | Indirect costs of pest management (e.g., decreasing soil productivity and pollination, increasing resistances) | 28,118–120 | +++ |
| **Domain 5: social equality and security** | | | | |
| SOC1 | Local equality | Equal distribution of all costs and benefits of pest management at the local scale (between actors in a region) | 31,116,117,121 | + |
| SOC2 | Global equality | Equal distribution of all costs and benefits of pest management at the global scale (between countries) | 2,36,60 | + |
| SOC3 | Adaptation capacity | Capacity of *all* farmers to adapt to future increases in pest pressure | 21,117,121,122 | ++ |
| SOC4 | Equal access pest management | Equal access of *all* farmers to suitable education, tools, and technologies for pest management | 29,30,123 | + |
| SOC5 | Safe work | Safe working conditions (e.g., gear and education) for *all* farm workers in pest management | 124,125 | ++ |

We only indicate the key references of relevance for the selected indicators instead of providing a comprehensive literature overview. The column "Direction of expected global effects" displays the results of the global survey of experts ($N$ = 473; respondents were allowed to refuse to respond if they lacked the expertise on a group of or single indicators; see Supplementary Table 4 for the sample sizes per indicator). The experts assessed the potential effects of the reduction of pesticide use and transformation to sustainable pest management based on the 24 indicators using a scale ranging from 10 to +10. We indicate the general direction of the expected effects (notably, all were positive (+) in relation to the expected improvement in the category) and the expected magnitude of change compared with those of other indicators. +, ++, and +++ indicate that the expected effect of an indicator belongs to the lowest, middle, or highest third, respectively, of the distribution of the expected effects across all indicators. See Supplementary Fig. 3 for the distribution of all indicator values and Supplementary Fig. 4 for the absolute mean values per indicator.

e.g., EU countries. But they can be important concerns in countries with laxer standards and controls (e.g., refs. 54,55).

In general, prior expectations on the direction of effects in the domains of economy and food security are ambiguous. They may change along the gradient of analyzed production systems, regions, and time horizons. Pesticides are typically efficient, cheaper and easier to apply compared with alternatives. But they can negatively affect agricultural productivity (e.g., ref. 56), especially if incorrectly applied. They can, for example, reduce ecosystem services (e.g., refs. 9,12,15). Additionally, pesticide resistances are increasing, making important active ingredients less effective (e.g., ref. 57). Further, crop quality may not only be endangered by the lack of crop protection but also by the overuse of pesticides, especially in countries with less regulations[58]. In turn, this may lead to a lack of consumer trust in certain foods and affect diet quality (e.g., ref. 59).

Indicators in the domains of social equality and security are least explored in the literature despite their potential importance. This is especially relevant in regions with high levels of existing inequality and the leakage of pesticide emissions; leakage meaning trade leading to the global reallocation of intensive production when regional incentives differ during a transformation in pest management (e.g., refs. 36,60). Finally, the relevance of the 24 indicators is expected to vary across the diverse regions, agricultural production systems, and pest management practices (see references in Table 1).

Our results on the direction of the expected global effects mainly align with expectations derived from the literature. The results emphasize the potential role of sustainable pest management in addressing several sustainability challenges (Table 1). Exceptions include the domains "Economics and Farmer Livelihood" and, to a lesser degree, "Food and Nutrition Security", in which experts

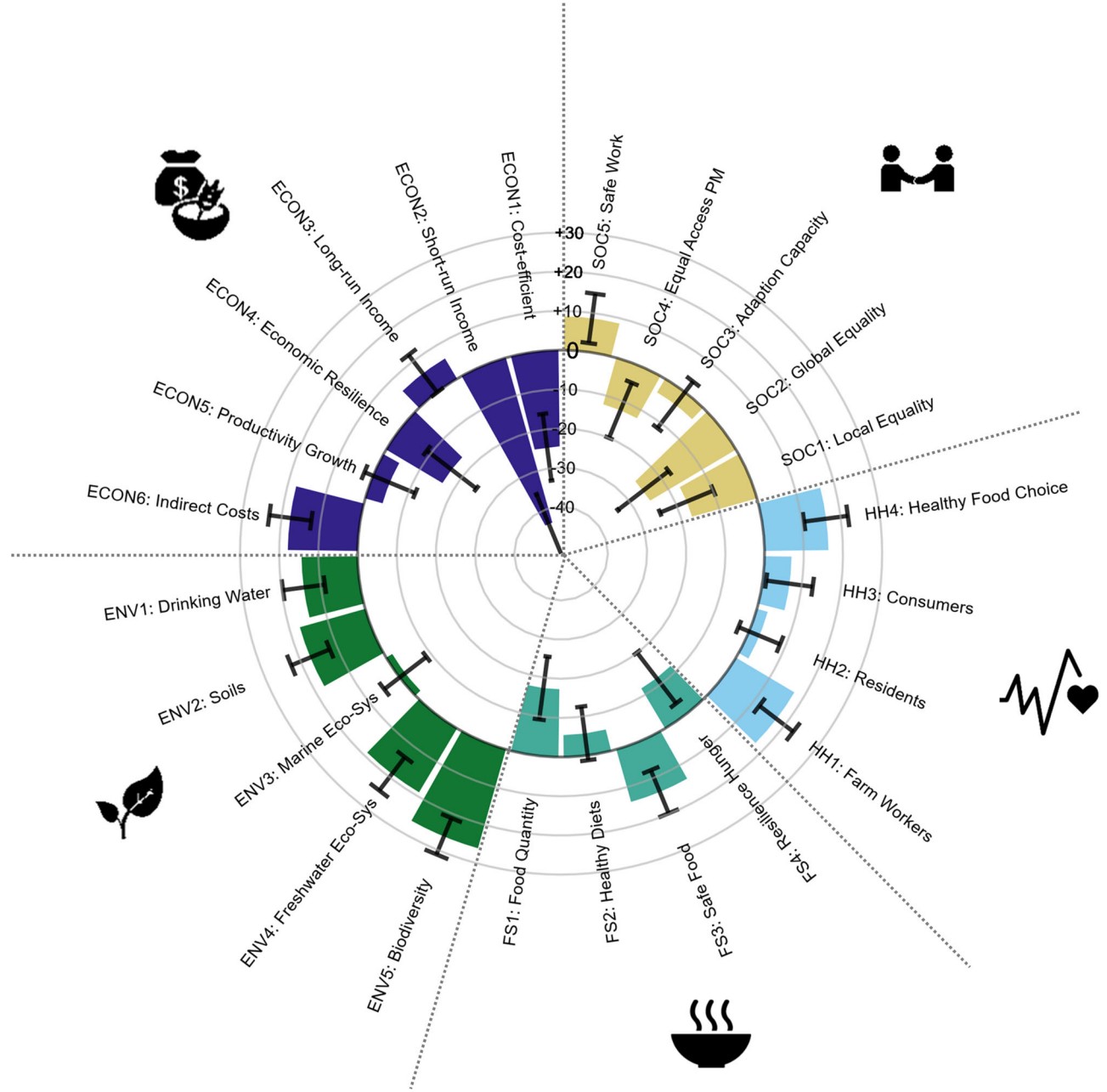

**Fig. 1 | Heterogeneity in the expected effects of a global transformation of agricultural pest management across indicators.** The Figure shows the heterogeneity in expected effects of a global transformation to sustainable pest management. Data comes from a global survey on experts ($N = 473$). The experts assessed the potential effects of the reduction of pesticide use and transformation to sustainable pest management in relation to the 24 indicators across five domains using a scale from 10 to +10, respectively. Colored bars depict the deviation from the global means (mean of all responses) for all indicators in percent and black error bars depict 95% confidence intervals per indicator, respectively. The following symbols indicate the five indicator domains (clockwise from bottom right): steaming bowl (food and nutrition security), plant with leaves (environment), bag of money and wheat stalk (economy and farmer livelihood), people shaking hands (social security and equality), and diagram with a heart (human health). Icons' source: OCHA. PM stands for "Pest Management".

expected more positive effects than those documented in the literature. This result is striking given that pesticides can often effectively protect crops, are easy to use and relatively affordable (e.g., refs. 5,26). However, if the responses are examined beyond their mean values, we find that they are heterogeneous across domains and respondents (e.g., 43% of individual respondents expected negative effects for at least one indicator). Furthermore, each indicator received negative responses that ranged from 2% (HH1) to 26% (ECON2) out of all responses per indicator (Supplementary Fig. 3). These results remain stable across different groups of respondents (e.g., by discipline or

sampling strategy and between academic and non-academic experts (Supplementary Figs. 5–7)).

## Highest potential of sustainable pest management in lower income regions

The heterogeneity of expected benefits across indicators (Fig. 1) and regions (Fig. 2) confirms the importance of a global assessment and a comparison of potential effects beyond the case study level.

We find that the observed differences in mean responses across the 24 indicators and within the five domains are plausible and align

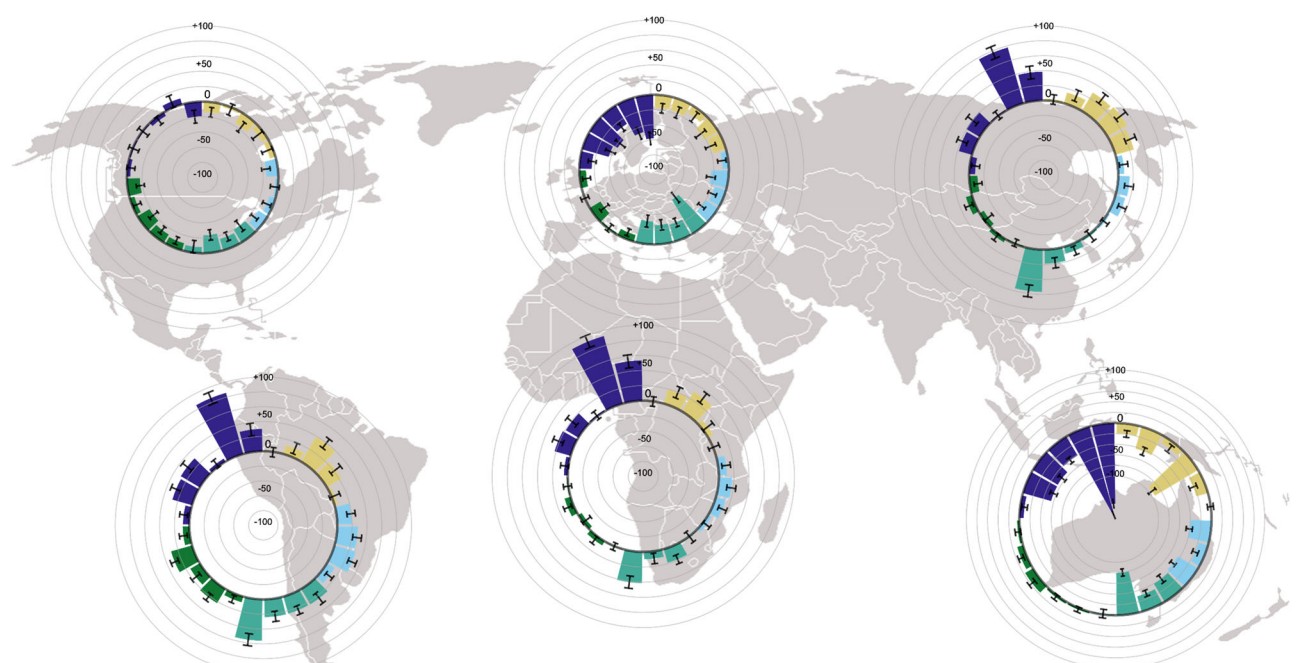

**Fig. 2 | Heterogeneity in the expected effects of a global transformation of agricultural pest management across regions.** The figure presents the results of a global survey of experts. Results are split upper continent (sample sizes are North America = 49, South America = 38, Europe = 128, Africa = 73, Asia = 90, and Oceania = 12). The experts assessed the potential effects of a reduction of pesticide use and transformation to sustainable pest management for 24 indicators in five domains on a scale from −10 to +10, respectively. Colored bars indicate the means of the responses per indicator and continent relative to the expected effects at the global levels in percentage, respectively. Black error bars display 95% confidence intervals. Each bar indicates the results for a single indicator, and the indicators are ordered in the same manner as that in Fig. 1. The following colors indicate the indicator domains (clockwise from bottom right): food and nutrition security (emerald), the environment (green), economic prosperity and Livelihood of farmers (indigo blue), social security and equality (beige) and human health (turquoise). Icons' source: OCHA. The map is based on a world map by Wiz999 under a CC BY-SA 3.0 license.

with expectations from the literature (Fig. 1). First, the potential benefits are, on average, lowest in the domains of "Economics and farmer livelihood" and "Food security". Second, within these domains, the potential is especially low for short-term farmer incomes (ECON2) and costs of sustainable pest management (ECON1) as well as for resilience to shocks of incomes (ECON4) and food supply (FS4). At the same time, the study finds higher potential benefits of the transformation for long-term farmer incomes (ECON3) and the development of agricultural productivity growth (ECON5). Finally, we find that the highest average benefits across all domains are expected in the environmental domain, where the positive effects of pesticide reduction are also best established in the literature (Table 1).

The regional heterogeneity in the level of the expected effects aligns with findings from the literature (Fig. 2). Experts on production systems in Europe and North America, on average, envision lower potential benefits due to sustainable pest management—across nearly all indicators. Production Systems in Europe and North America, on average, are very productive and operate with high material and capital input. In contrast, experts in South America, Africa, and Asia, on average, identified higher levels of potential benefits. Especially for these regions, experts, on average, detect potential win–win scenarios due to the reduction of pesticide use and the transformation of pest management. The potential for the mitigation of adverse health impacts was rated lowest in Europe, which reflects the currently stringent pesticide regulations on human health in Europe and a recent tightening in pesticide regulations. This led to a ban of more toxic pesticides than in other regions worldwide[61]. Along this line, the potential benefits for human health were rated substantially higher for North America, a region with a similar input intensity but different regulations[61].

Furthermore, the expected benefits of the transformation to sustainable pest management are significantly correlated with country-level indicators of human development (the United Nations Human Development and Planetary pressure-adjusted Human Development Indices) and progress on related Sustainable Development Goals (SDGs) (Supplementary Table 2, see "Methods" section). We consistently find that higher expected benefits of the transformation are correlated with (i) lower human development indicators, (ii) lower average income (measured in 2015 Gross Domestic Product per capita), and (iii) low achievement of SDGs (Supplementary Table 2). One exception is the environmental domain, in which we found the potential benefits to be equally high across all countries and regions. The magnitude of the Kendall rank correlation coefficients between the indicators assessed in the survey and the external indicators lies between absolute values of 0.00 and 0.25, which indicates weak to moderate correlations. Strong correlations would have been unlikely, because the assessed indicators relate to the specific effects of the transformation. Other factors may also play an important role for achieving SDGs. For example, for the biodiversity indicators, land use intensity and pollution from industry or fertilizers play a role additional to pesticide use[10]. Potential reasons for a high success of ecological and integrated pest management practices in the global south are the (lacking) availability of suitable chemical options, resource scarcity of farmers, and cheaper alternative solutions. Additional reasons can be pesticide misuse and synergistic and ecologically beneficial effects of more diverse and pesticide-reduced production systems[62–64].

Finally, out of all indicators and regions, respondents' expectations were most heterogeneous for Europe, North America, and Oceania and the domains of Economics and Farmer Livelihood and Food and Nutrition Security (see Supplementary Notes 3). This

indicates that experts had diverging opinions on expected effects especially for these indicators and regions. The finding aligns with our expectation of heterogeneous effects in these domains. Note that the averages are aggregated on a large scale and that important heterogeneity within regions exist (Supplementary Fig. 14). This result highlights the importance of looking into potential drivers of expected effects. The descriptive analyses in this section do not account for context- and respondent-specific characteristics that potentially drive expectations of effects. Thus, we analyze these in the next section using regression analysis.

### Controlling for respondent- and context-specific characteristics

To assess the drivers of the expected effects, we spatially matched the survey data with a wide set of country- and regional-level data on key characteristics of agricultural production systems (see "Methods"). We then analyzed the data using multiple regression analyses (see Fig. 3 and Supplementary Table 3 for detailed regression results). This allows us to control for key characteristics of the production system (e.g., level of pesticide risk, pest pressure). Further, the survey data allows us to control for key respondent characteristics and thus mitigate potential biases, for example, from different disciplinary backgrounds and expertise of participants.

The regression analyses revealed important patterns. First, we consistently found diminishing marginal gains of sustainable pest management across all indicators (the variable Level Sustainable PM in Fig. 3). That is, higher current implementation levels are associated with lower marginal benefits of sustainable pest management across all domains. For example, a one-point higher level of current implementation is associated with a 0.15-point lower potential of the transformation to increase the long-term income of farmers (regression coefficient = −0.15 [−0.29; −0.01] (95% CI)). Importantly and in line with expectations, this relation is reversed for expected effects on the short-term income of farmers (ECON2) (regression coefficient = 0.15 [−0.04; 0.33] (95% CI)). In general, experts expect the lowest potential of the transformation for arable production systems and the highest potential for the more pesticide-intensive horticultural production systems. In horticultural production systems, sustainable pest management is especially expected to provide benefits for food security and social domain indicators. This finding reflects the large potential of sustainable pest management for improving the productivity of horticulture, especially for small-holder and self-sufficient producers in low-income regions (e.g., ref. 65).

In production systems with currently high environmental risk due to pesticide use, the transformation is expected to be less beneficial for the reduction of biodiversity loss (ENV5; regression coefficient = −0.42 [−0.79; −0.06] (95% CI)) and the restoration of the confidence of consumers in the absence of food contamination (HH4; regression coefficient = −0.48 [−0.89; −0.06] (95% CI)). At the same time, it is expected to be more beneficial for the reduction of global distributional inequality (SOC2; regression coefficient = 0.74 [0.18; 1.30] (95% CI)). We interpret these findings on biodiversity loss, consumer confidence, and inequalities as an indication of the legacy effects of pesticide contamination. Case studies on pesticide contamination of soils or water bodies, as well as ecosystems, have previously reported such legacy effects (e.g., refs. 66–68).

For the economics domain (ECON), the highest potential benefits are expected for production systems with low attainable yields but higher pest pressure. In addition, for the food security domain, we find the highest potential benefits in production systems with low attainable yields. We interpret these results to suggest a different strategy for agricultural development of pest management. This especially holds for regions in Africa with currently low levels of pesticide use and high potential benefits of sustainable pest management (e.g., refs. 63,69,70). The potential effects of a transformation are lower in regions with high yields. This shows the challenges of a transformation in Europe, North America or Oceania.

The environmental domain is least affected by the characteristics of the production system. Potential benefits are expected to be equally high across global production systems. This may be related to the interconnectedness of global ecosystems[71] or potential trade-offs between increased intensities and regulations across regions (e.g., ref. 55). For human health, we found expected benefits of the transformation, especially in production systems with low attainable yields. This result is in line with expectations from the literature. That is, education and equipment for handling pesticides safely, as well as regulations, can especially be improved in countries with small-holder and self-sufficient farmers (e.g., ref. 52).

Social equality and security is the domain covered least by the literature so far. We find strong expected differences between the types of production systems when comparing the potential of the transformation to solve social challenges (arable crops lower and horticulture higher potential). Experts further expect that resolving distributional conflicts, for example, the local and global distribution of the costs and benefits of pest management (SOC1 and SOC2), will be especially difficult in production systems with high attainable yield and pest pressure. For example, we find that a 1% higher potential pest damage is related to a 0.09 point decrease in the potential to improve the local distribution of costs and benefits of pest management (SOC1; regression coefficient = −0.09 [−0.18; −0.01] (95% CI)).

The results further highlight the importance of controlling for the characteristics and expectations of respondents in the regression analysis. Respondents with higher levels of self-indicated expertise in the economic, environment, and human health domains also expected higher levels of benefits from the transformation in these domains. Furthermore, experts who rated certain indicator domains to be more important expected higher levels of benefits for an indicator in the domain. This finding holds consistently across all indicators.

Finally, the research field of the respondents shapes their expectations. Socio-economists consistently rate the potential effects of the transformation least optimistic compared with other research fields (except for the social security and equality domain). Ecologists rate the potential effects as most optimistic. This contrast underscores the different discourses in various fields of research and the importance of covering a broad sample of experts across disciplines in the research on sustainable pest management systems. We find that the expectations of experts outside academia do not significantly differ from those of researchers (except for indicators within the environmental domain), which supports the sampling strategy of our study.

### How to transform pest management systems

Despite our results of multiple expected benefits from a transformation to sustainable pest management, the pathway to such a transformation is not easy or free of costs. In the second part of the analysis, we therefore ask what is required to transform global pest management systems.

To address this question, the participants rated the importance of different measures for the achievement of a full transformation to sustainable pest management by freely distributing 100 points (indicating importance) across a list of predefined measures. We here report the average points attributed per measure across all participants. For all regions and production systems, we consistently find that "identifying and providing access to effective and cost-efficient substitutes for pesticides" (i.e., accounting for prices and efficiency in protecting crops) is the topmost priority for experts to support the transformation (22/100 points, SD = 0.14) followed by "education and extension on pest management" (18/100 points, SD = 0.13), "economic support" (13/100 points, SD = 0.11), "legislative support" (12/100 points, SD = 0.10) and "awareness building" (i.e., of unintended effects)

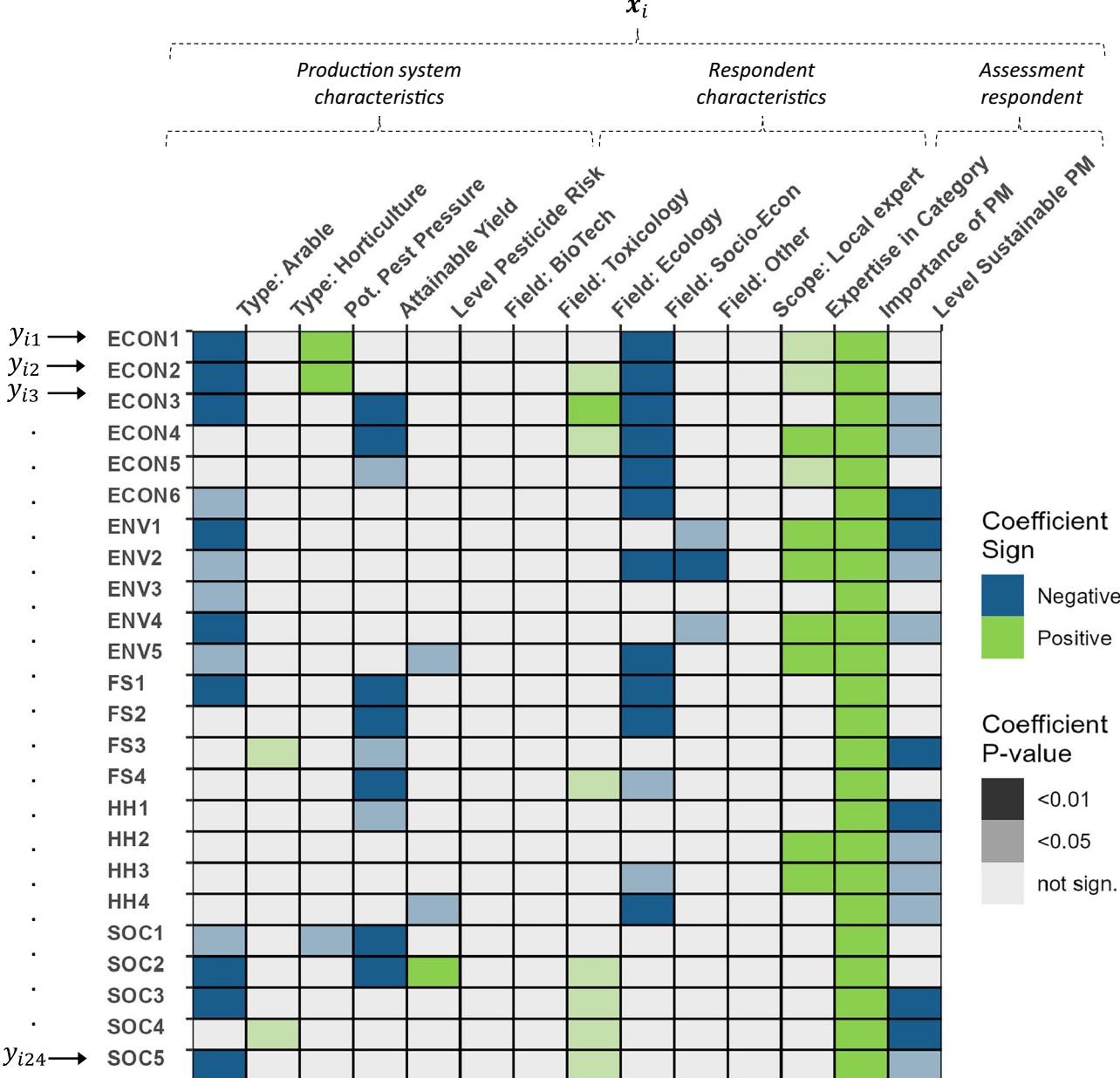

**Fig. 3 | Drivers of the expected effects of a global transformation of agricultural pest management.** The figure illustrates the drivers of the expected effects from a transformation of global pest management based on regression analyses (OLS) for each of the 24 indicators ($j = 1, \ldots 24$). Response variable ($Y_{ij}$) are the expert assessments of the potential effects of a transformation to sustainable pest management systems on a scale from −10 to +10. We account for key characteristics of the production system (type of farming, potential pest pressure, attainable yield, current level of pesticide risk used) and control for important respondent characteristics (field of research and scope of expertise), self-elicited expertise in the indicator category, elicited importance of pest management for the indicator category, and elicited current implementation level of sustainable pest management in the production system. The reference categories for the categorical variable type of production system are general production and for field of research pest management sciences. We then test the significance of the

regression coefficients with two-sided $t$-tests against the null hypothesis that the coefficient estimates are zero. Colors of fields indicate signs of point estimates (blue = negative effect on indicator value, green = positive effect on indicator value, grey = non-significant effect ($p$-value > 0.05). A higher intensity of color indicates the higher statistical significance of the point estimates. Signs of point estimates with $p$-values > 0.05 are not shown (grey fields). Abbreviations refer to: ECON1 Cost-efficient, ECON2 Short-run income, ECON3 Long-run income, ECON4 Economic resilience, ECON5 Productivity growth, ECON6 Indirect costs, ENV1 Drinking water, ENV2 Soils, ENV3 Marine ecosystems, ENV4 Freshwater ecosystems, ENV5 Biodiversity, FS1 Food quantity, FS2 Healthy diets, FS3 Safe food, FS4 Resilience hunger, HH1 Farm workers, HH2 Residents, HH3 Consumers, HH4 Healthy food choice, SOC1 Local equality, SOC2 Global equality, SOC3 Adaption Capacity, SOC4 Equal access to pest management, SOC5 Safe Work.

(12/100 points, SD = 0.10) as the top five responses (see Supplementary Fig. 8 for an overview of all responses).

## Discussion

Our results on key measures for supporting a transformation highlight that support for the development of effective, cost-efficient, and

reliable, alternative solutions for pest management should be the topmost priority at the global scale. A research gap currently exists in terms of alternative solutions for the management of many pests and production systems. Agronomic developments for improved and diversified production systems that prevent pests can be combined with new opportunities from digital innovations, New Genomic

Technics, and the development of novel biopesticides[72–74]. This is a promising avenue, which has already shown the possibility to increase productivity while significantly reducing pesticide use and risk[75,76].

Pesticides are usually easily and broadly applicable in different contexts. Substituting them often requires combinations of several alternative management actions (e.g., adjustment of the crop rotation and crop variety together with mechanical pest control), at various scales (e.g., the field and landscape scale), adapted to the production context (e.g., growing conditions and pest pressure) (see refs. 26,31,33 for a discussion). Our results on key measures for supporting a transformation (Supplementary Fig. 8) highlight that the *technical feasibility* of alternative practices for pest management is currently insufficient for an efficient transformation. They further demonstrate that locally adapted bundles of technical and socio-economic support measures for the implementation of sustainable pest management systems are crucial. These can range from extension and education to awareness building, economic support, and legislative efforts. Wuepper et al.[77], for example, highlighted the global importance of policies for the reduction of pesticide risk, and Möhring et al.[25] show the importance of policy mixes for supporting this transformation. This can start, for example, by providing information and facilitating access to pest management by farmers. But in places where big trade-offs remain this will not be sufficient and a transformation will likely jointly require the development of new farming systems, novel technologies, and economic instruments to overcome barriers[26]. The recent emergence of production systems that operate without synthetic pesticides in Europe shows that the combination of integrated pest management approaches, together with a mix of private and public funding, can enable large-scale systems that offer a "third way" between conventional and organic agriculture[78].

However, different approaches will be required for globally heterogeneous regions and agricultural production systems. For example, natural conditions, the strength of institutions for implementing and enforcing rules, the capacity of governments to support farmers and their knowledge base, access to markets and technology, and the economic capacity of farmers to purchase inputs and implement practices differ strongly across regions and will determine appropriate strategies. Advancing on this question requires a holistic and place-based approach. It should not only consider farmers, but also the important, systemic role of food-value chain actors and industry in the global transition to sustainable pest management[26,79–81]. Importantly, novel cropping systems shall be evaluated by efficiency metrics (e.g., yield per unit of input) and impacts (e.g., on ecosystems and ecosystem services)—not by the type of production (e.g., whether and how much pesticides are used)[82].

We here create a multidisciplinary framework to assess the potential effects of a global transformation to sustainable pest management across the five domains "environment", "human health", "food and nutrition security", "economic prosperity and livelihoods", and "social equality and security". We find that expected benefits of a transformation towards sustainable pest management are especially pronounced for long-term effects (Fig. 1). This indicates that the respondents expect a transformation of pest management systems to be a valuable investment in the future. The results of the regression analyses (Fig. 3) emphasize the importance of undertaking a cautious transition for highly productive regions, as well as the diminishing marginal gains of sustainable pest management. Global benefits from a transformation are expected, especially in the environmental and health domains. Importantly, the results caution that future benefits for some of these indicator domains may be lower when a transformation is further delayed. An important objective of the analysis was to compare potential effects across impact domains and regions and to assess drivers of this heterogeneity. Generally, we find a large similarity in expected effects for Europe, Oceania and North America on the one hand, and South America, Africa and Asia on the other hand (Fig. 2). Further, the regression results imply expected benefits of choosing a less pesticide-intensive pest management strategy, especially in regions in Africa with currently low levels of pesticide use (Fig. 3).

Finally, the important scale mismatches in the distribution of potential benefits and costs from a transformation of pest management must be addressed across different local actors (e.g., in watersheds or landscapes[83]), countries (e.g., addressing leakage between the global South and North[60]), and actors of food systems (e.g., farmers, farm workers, industry, and consumers[26]). Our finding of lower distributional and social benefits in systems with high attainable yields and high pest pressure indicates that the benefits of a transformation are distributed heterogeneously across regions worldwide. This emphasizes the importance of a coherent global strategy that avoids leakage. That is, if lower pesticide footprints in one part of the world imply lower productivity, which is compensated by higher imports—this may increase the intensity and footprint of production and land use in other parts of the world. Such mechanisms may lead to environmental and human health impacts simply shifting from one country to another and shall be avoided[84,85]. Future regional and international agreements should address the question of pesticide leakage in an interconnected world (e.g., ref. 36). This can avoid environmental and social impacts, for example, from Europe, being shifted to other parts of the world via changes in trade-flows (e.g., ref. 84). Our assessment highlights pathways for advancing this debate.

Global expert assessments can provide important insights into the expected effects of large-scale transformations, as well as regional hotspots and leverage points. Global and regional policy goals for pesticide reduction have been set, but assessments on the potential effects of such a transformation are still missing. Filling this gap is a necessary first step on the path to achieving global policy goals of pesticide risk reduction. Our data and results can serve as an important baseline for urgent societal discussions, the design of policy and support measures and the direction of future research efforts.

There are limitations to our study. An inherent limitation of expert assessments is that results are dependent on the chosen definitions in the survey. Biases may occur depending on how individuals understand and perceive the questions. We here addressed this by (i) an external and internal validation of our study design before its execution, (ii) choosing a large and diverse sample of experts (different geographic regions, professions and disciplines), (iii) controlling for important respondent characteristics in our regression analysis, and (iv) by choosing a research design that focuses on heterogeneity and drivers instead of absolute indicator levels (see Fig. 4 for an overview). Additionally, we define the target of the transformation as a system with minimal or no pesticide use, which closely aligns to the well-known concept of Integrated Pest Management. The definition might be understood differently in different production contexts. We therefore also assess and control for its current implementation in the regression analysis. However, we do not explicitly refer to "pesticide risk" reduction, since this term is often not clearly defined and assessments in terms of pesticide risk reduction are less common in the field, and therefore harder to judge for experts from various disciplines. Still, despite our best efforts, we cannot rule out remaining biases completely.

Emerging literature in the field will contribute to consolidating empirical evidence, also for specific regions. Future research should consider other important characteristics of production systems, such as farm and farmer characteristics and attitudes to different types of pest management, which we could not include in our global survey. For example, small-holder maize farmers in sub-Saharan Africa face a completely different socio-economic context than large-scale commercial maize farmers in Southern Africa.

Given the broad geographic and interdisciplinary scope of our work, our results are not intended to quantify absolute effect sizes for given indicators and regions. The results can therefore only be a start, identifying focal points for more regional- and local- in-depth analysis.

Studies that provide information on the choice and development of locally adapted practices and solutions will be crucial for the development of sustainable pest management. This will require farm-level surveys and research. Future assessments should take potential changes in land use and resulting direct and indirect effects, for example, on greenhouse gas emissions into account. Further, future research should complement our results from expert assessments with results from statistical estimates based on observed data or simulation models. For example, on changes of farmer decision-making, global prices, and land use decisions. These have been successfully combined in adjacent fields, e.g., for impacts of climate change[86]. But currently, suitable methods to combine these are missing in the field of sustainable pest management and their implementation across scales is hampered by limited data availability.

## Methods

We employ a robust, multi-stage procedure to design, implement, and analyze our global survey (Fig. 4). First, we defined baseline and target scenarios of pest management for the assessment. Second, to provide a holistic evaluation of the potential effects of sustainable pest management, we created an interdisciplinary framework of indicators for its assessment. Third, based on the framework, we then created and tested an online survey. Fourth, we distributed the survey to relevant experts worldwide and let them assess and quantify potentials and trade-offs from a reduction of pesticide use and a transformation to sustainable pest management systems. Fifth, we combined survey data with various external data sources. Finally, we analyzed the combined dataset. We defined several robustness checks and sensitivity analyses throughout.

We restrict the focus of our assessment to pest management in agricultural production, excluding other areas of pesticide use, such as national biosecurity measures or post-harvest pest management.

### Definition of a baseline and a target scenario of pest management

First, we defined a baseline and a target scenario of pest management for the assessment to set expectations of respondents and make survey results comparable. This required a definition of sustainable pest management. Despite the frequent use of this terminology in research and policy documents, we have not identified a common definition. Existing studies often focus on certain concepts, such as Integrated Pest Management or Agroecology or specific practices or technologies (see main text). However, fulfilling existing policy goals for a substantial reduction of pesticide use and risk is not necessarily linked to the adoption of a single concept or practice, but might rather be achieved by the adoption of a multitude of concepts, approaches, technologies, and practices in parallel (e.g., ref. 26). To provide an umbrella for this variety of approaches we chose to create a broad and explicitly policy-relevant definition. We therefore refer to Sustainable Pest Management and not only, for example, Integrated Pest Management in order to provide a more inclusive definition that also captures other important approaches, such as agroecological pest management. We tested before and after conducting the survey that our intentions aligned with participants' perceptions ("Methods" section below, Supplementary Materials Section A4). To align with our research question, this definition thus had to have two central elements: (i) reflect sustainable use, as defined by the Brundtland commission, i.e., "meeting society's current and future needs", and (ii) be policy relevant, i.e., substantially reducing pesticide use and substituting it with alternative crop protection measures.

We defined the baseline scenario as the current status of pest management in each production system. Pest management can be highly diverse depending on the region. In order to control for this heterogeneity in our analyses, we thus additionally collected key characteristics of the respective production and pest management systems in the survey.

We defined sustainable pest management (the target scenario) as "pest management systems with no or minimal use of pesticides, using (a combination of) … alternative pest management practices…their implementation leading to a substantial reduction of current pesticide use levels." We refer to this target scenario as the sustainable pest management scenario. Here, no or minimal use refers to the notion of pesticide use as a last resort, for example, as defined in Integrated Pest Management principles. We explicitly used a subjective definition of pesticide reduction (no or minimal use of pesticides and substantial reduction) and not an explicit numeric goal, as (i) regional reduction goals and reference scenarios might differ globally, (ii) unified recommendations for a reduction of pesticide use often do not exist, and (iii) these formulations reflect the spirit of the above-discussed pesticide reduction goals. Note that we explicitly do not refer to pesticide risk reduction, since this term is often defined differently, for example, using different indicators[25], or has different connotations. Further, assessments in terms of pesticide risk reduction are less common in the field, and therefore harder to judge for experts from various disciplines (e.g., ref. 25).

To make the definition clear and practically applicable to respondents, we further specified key substitutes for pesticides that fall under this definition. More specifically, we explained in the survey that to reduce losses from pests, including plant pathogens, farmers may use single or combinations of alternative preventive and curative measures for pest control, for example, along the lines of integrated crop protection or agroecological principles (e.g., refs. 33,83). Important examples of such measures include biocontrol, bio-pesticides, mechanical/technological solutions (e.g., mechanical weed control, smart and precision farming), breeding solutions (e.g., resistant and adapted varieties), and adaptations in crop management (e.g., adapted or spatially and temporally diversified crop rotations, field hygiene measures, measures favoring natural solutions for pest control). A variety of different measures exists, but their use is very context dependent and currently not widely adopted, despite their recognition (see above). The scope and scale of these measures might differ and range from improving pesticide use efficiency to substituting pesticides, and completely redesigning production systems on a field-, farm-, or landscape level. Integrated pest management principles are even mandatory to consider for farmers in some regions (e.g., cross-compliance guidelines for participation in EU direct payment schemes) but are typically loosely defined and rarely implemented effectively (e.g., ref. 31).

Note that this broad definition may also entail respondents from different backgrounds and experiences associating different practices with a switch to sustainable pest management. We capture this heterogeneity by covering a large and diverse sample of respondents with experience across different production systems, regions, and disciplines.

The availability and implementation of effective and cost-efficient, alternative pest management measures may further vary throughout the diverse global production systems. We therefore also ask about the current level of adoption of sustainable pest management in the production system and region, and control for it in our analysis. In our analysis, we then use this information to assess how heterogeneity in expertise and respondents' characteristics and expectations relates to the respondents' expectations.

### Assessment framework for the potential effects of a transformation to sustainable pest management

We identify relevant indicators for providing a holistic evaluation of the potential direct effects and trade-offs of transforming pest management systems.

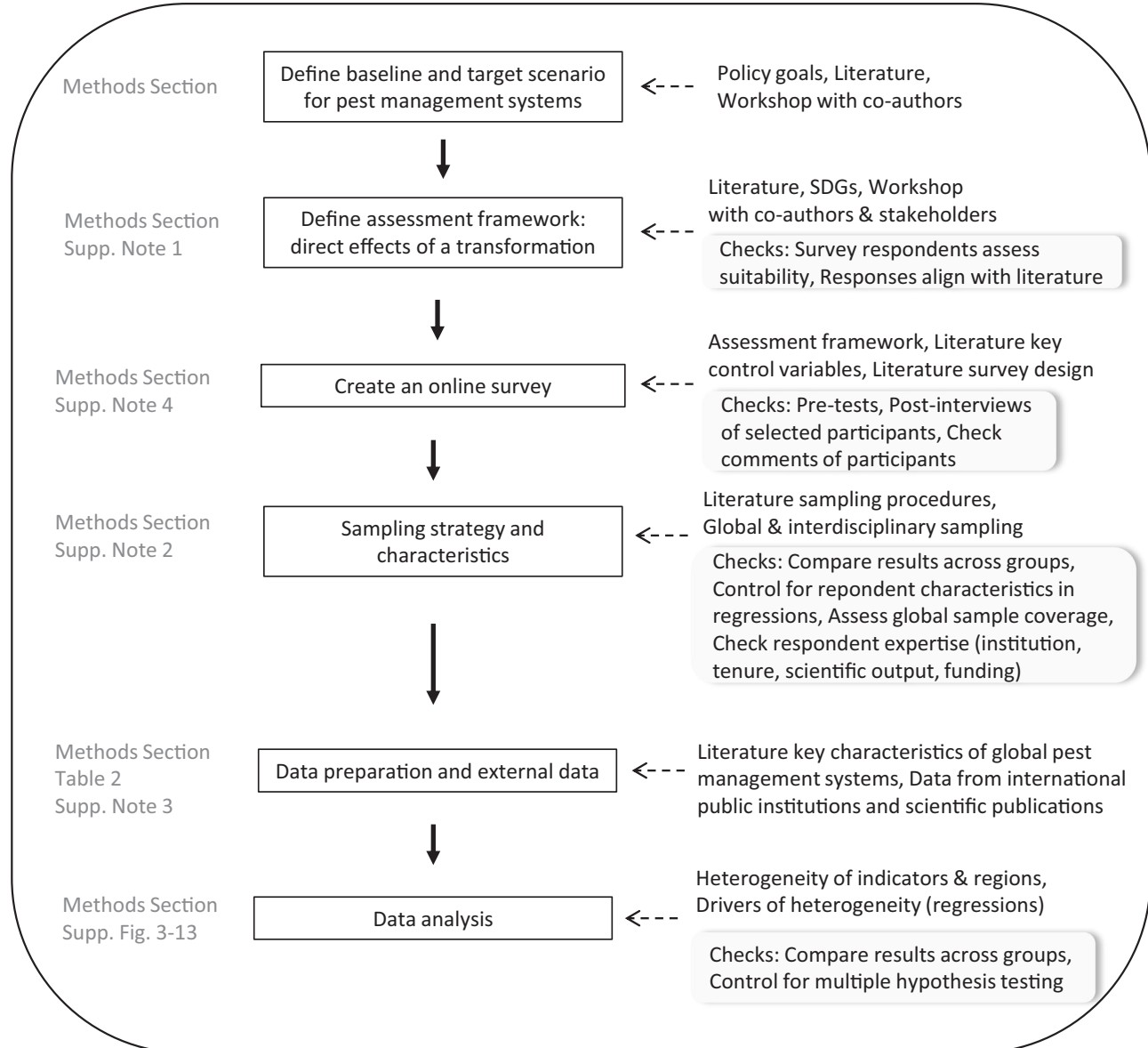

**Fig. 4 | Procedure for survey design, implementation, and analysis.** The Figure shows the procedure for survey design, implementation, and analysis. The main steps of the procedure are shown in boxes in the middle. The right column shows utilized methods and data for each step. For each step robustness checks are further indicated in grey boxes underneath the main methods in the right column (where applicable). The left column indicates in which chapters of the "Methods" section the steps are described in detail, and where robustness checks can be found.

We start by identifying the relevant dimensions for assessing transformations in agricultural production systems. To this end, we look both at high-level studies and review studies on related topics concerning the sustainable transformation of agriculture from the last two decades. For example, Seufert and Ramankutty[87] identify production, environmental, producer, and consumer dimensions in their literature review on the performance of organic agriculture. Deguine et al.[33] mention environmental, economic, social, human health, and food security dimensions in their review of the literature on agroecological crop protection. Waddington et al.[29] analyses potential economic, social, human health, environmental, and production-related impacts in their systematic review on the effectiveness of farmer field schools (the introduction of practices such as integrated pest management) in low- and middle-income countries. These dimensions also largely overlap with the agriculture-related challenges that are covered in the SDGs. In line with the identified and reviewed literature (see Table 1), we therefore chose five indicator dimensions. First, the environment, as pesticides are an important source of environmental pollution. Second, human health, as pesticides have been shown to have adverse effects on agricultural and non-agricultural populations. Third, food and nutrition security, due to the key role of crop protection for food production. Fourth, economic prosperity and livelihoods of farmers, due to the important role of crop protection for economically viable farming. Fifth, social security and equality due to heterogeneity in pesticide exposure and distribution of costs and benefits.

In order to identify detailed indicators on potential direct effects of a transformation, we then created overviews of key literature on potential effects of pesticides and a transformation of global pest management systems from the last two decades in these five domains, based on the expertise of co-authors and targeted searches in literature databases.

Based on these overviews, we identified suitable indicators of potential costs and benefits for each category (see Table 1 for an

overview of references per indicator and the main text for a detailed discussion). We focused on indicators, which are related to direct effects. This means, for example, we are not assessing effects from potential land use changes, e.g., on greenhouse gas emissions. In order to provide a detailed overview of effects across all five domains and for the heterogeneous global production systems, we required indicators to meet the following criteria: (i) high societal and policy relevance, (ii) relevant for the diverse global agricultural production systems at local, regional and global scales (iii) together representing the range of potential effects in the respective category, and (iv) sufficiently distinct per category to allow for differentiated responses in the survey. Similar criteria have also been applied by previous studies in related fields: Pe'er et al.[88] for example, also focus on key sustainability issues and societal demands in their expert assessment of the European Union's Common Agricultural Policy. Seufert and Ramankutty[87], and Mesnage et al.[51] point out the importance and scarce availability of data for different global regions in this research field, and Stantcheva[89] highlights the importance of formulating precise questions in expert surveys.

In total, we identified 24 key indicators (Table 1). Following the approach of Seufert and Ramankutty[87], we also include indicators that have so far received limited attention in literature (for example, due to scarce data availability). Global assessments of single indicators from our framework exist - but only for the effects of pesticides and not for a transformation to sustainable pest management (see for example, refs. 40,48 for global studies on pesticide pollution in soils and water bodies and section 2 of the main text for a more detailed discussion for each domain). Note that to create a holistic overview of potential effects and trade-offs across all indicator domains, we also included indicators that highlight a similar challenge from a different, domain-specific angle. For example, HH3 (health effect on consumers) and FS3 (safe food), as well as HH4 (healthy food choices) and FS2 (healthy diets). We believe that this approach does not affect our analysis of results, as we (i) interpret results by indicator category, (ii) use multiple, separate regression analysis, and (iii) check for multiple hypothesis testing in our analysis. We focus on direct effects of changes in pest management, where literature indicates a potentially significant relation to key societal challenges, for example, those covered in the United Nations SDGs (see Supplementary Table 6). We acknowledge that changes in pest management could have even broader, indirect effects, for example, if land use decisions of farmers are affected. This is an important topic for future research, but was not in the scope of our analysis.

The literature overviews and the resulting assessment framework, comprising the 24 selected key indicators, were verified in several steps before and after conducting the survey, following recommended steps for expert assessments. Stantcheva[89] for example, highlights the importance of pre-tests for surveys, and Boijke et al.[90] discuss the importance of the verification of the survey by respondents.

Before the survey, both literature overviews and selected indicators were first discussed in several iterations with the interdisciplinary and international group of co-authors. Second, the framework was again critically appraised based on first results in a two-day workshop at ETH Zurich in summer 2022, (i) with co-authors and (ii) with other selected participants from the field (research, industry, and general public).

Finally, we also tested the validity of the assessment framework and the survey after it was conducted: In the survey, we asked respondents to assess the relevance of the chosen indicator domains for assessing potential cost and benefits of pest management. Its relevance was confirmed by respondents (Supplementary Materials, Section A1). Further, we analyzed responses and tested if hypotheses from the literature were reflected in the results (see the main text). Moreover, we went through comments of respondents and addressed their concerns in our analysis (Supplementary Materials, Section A4). Both analyses confirmed the validity of the assessment framework.

**Creating an online survey based on the assessment framework**
Based on this framework, we designed, tested, and conducted a global survey of experts from relevant disciplines and fields on the expected effects of transforming pest management.

We created an online survey in the software Limesurvey based on the assessment framework. For the survey design, we followed general recommendations and best practices for survey design (e.g., ref. 89). More specifically, for questions with a natural unipolar distribution (only an answer of zero or positive values are possible), we chose sliders with an odd number, as recommended by Stantcheva[89]. Further, we chose a scale of (0–10 = 11 values), as this scale is easily comparable and transformable (for example, to a scale of 0–100) but gives a lower number of values for choice[89]. For questions where we expected both positive and negative responses, we chose "Bipolar ordinal scales [...] along two opposite dimensions, with a zero-point located in the middle of the scale, [...]" (ref. 89, pp. 20–21) to avoid priming of respondents.

The survey consisted of five main parts (see Supplementary Materials, Section A5 for the complete survey text), which are described below.

Key characteristics of the production context and respondent (allows to identify drivers and correct for potential biases in the analyses): Assesses the (i) type of expertise: occupation and field of research, (ii) production system expertise: type of production system, (iii) scope of expertise: local, regional, or global and location of production system of expertise, and (iv) self-elicited expertise per indicator domain: from 0 (no expertise, category was excluded for the expert) to 10 (very high expertise). This information was needed to exclude respondents from domains where they have no expertise and conduct robustness checks for domain-specific expertise, since our survey covered different domains and experts with different disciplinary backgrounds. It allowed us to control for potential biases related to different levels of expertise in the regression analyses.

Relevance of indicators for pest management (allows to verify the selection of indicators and control for participants' expectations): Assesses the perceived importance of pest management for agriculture's contribution to each of the five indicator domains: from 0 (not relevant) to 10 (highly relevant). This information was needed to conduct robustness checks on how heterogeneity in the perceived importance of indicator domains across individual respondents, agricultural production systems, or regions may affect results. It was further used to assess the validity of the chosen indicator domains for assessing changes in pest management systems. Importantly, respondents were specifically asked to assess the importance of pest management as far as it concerns the contribution of agriculture to these challenges. Note that this question was not mutually exclusive, meaning that other agricultural drivers, which were not of concern for the survey, might be of a similar or even higher importance for some of the indicator domains.

The current status of sustainable pest management (allows to control for the heterogeneous baseline scenarios): Assesses the status of sustainable pest management according to the given definition: from 0 (no implementation) to 10 (complete implementation) in the respective region and agricultural system of expertise. Generally, there is a lack of data on the implementation of pest management practices. Existing indicators, for example, on pesticide use or the amount of land under organic farming[4] are often missing or incomplete (e.g., ref. 51). Even where available, these alone are not adequate indicators of sustainable pest management practices. For example, a high implementation of integrated pest management practices may not show up as a high share of organic farming in a region or cropping

system. Note that responses might be interpreted in different ways, depending on the context of the production system. Interpretations might for example, differ between regions and agricultural systems with (i) high levels and (ii) a lack of sufficient pest management. For the latter regions, low pesticide use by default not necessarily indicates a sufficient implementation of sustainable pest management practices, but a general lack of crop protection. Whereas it is a good indicator for regions with high levels of crop protection.

The potential effects of a transformation to sustainable pest management: Assesses the expected effects of a complete shift to sustainable pest management: from −10 ( = severe decrease in indicator target) to (+10 = complete achievement of indicator target) in their region and agricultural production system of expertise for the 24 indicators in our assessment framework. Responses to these questions may thus be interpreted as the potential effects of a full shift to sustainable pest management (our target scenario defined above) and can be interpreted as upper-bound effects of a transformation of global pest management systems. This approach allows us to identify the expected direction of the effects (negative or positive) and compare their expected levels across the groups of indicators, regions, and agricultural systems. However, note that the response should not be interpreted in absolute terms.

Tools and policies for a transformation to sustainable pest management: Assesses which measures would be required for a complete shift to sustainable pest management. Respondents freely distribute 100 total points on a broad range of potential measures, where more points indicate a higher importance of a measure.

The survey was validated and pre-tested in several steps: First, it was pre-tested and checked for consistency by all co-authors, which represent the diversity of disciplinary backgrounds and global regions targeted in our sampling strategy (see below). This led to a more precise wording and reformulation of baseline and target scenarios and the questions for assessing the 24 indicators. In the next step, the survey contents were translated with a deep learning software into five other languages, starting from the English version (Chinese, French, German, Portuguese, Spanish). These cover the majority of global agricultural production, assuming that scientific experts speak at least the most common foreign language per country. Each translation was then checked and corrected by a native speaker working in the research field. Finally, the survey was again pre-tested and checked for consistency in each language by a native speaker. The survey was coded in such a way that no backward translation of results was necessary (common structure and question numbers for all language layers).

All respondents were informed about data storage and anonymity, rights to their data, and the purpose of the survey. Their understanding and approved consent was assured before answering the survey. Finally, the survey was assessed and approved by the Ethics Council of ETH Zurich before its distribution (approval number 2022-N-34). The survey was sent out with two reminders for non-respondents. It was open from the end of March 2022–October 2022.

After conducting the survey, we further contacted three selected participants from different research disciplines to assess their perceptions and understandings of the used definitions and survey questions. The respondents confirmed the alignment of our research questions and the language used in the survey. We further used their answers to improve the description and understanding of survey results. Moreover, we went through comments of respondents and addressed their concerns in our analysis (Supplementary Materials, Section A4).

## Sampling strategy and sample characteristics

Previous expert assessments on global or regional levels have either sampled their experts from (i) a predefined list of experts or participants of a workshop (e.g., refs. 30,88), (ii) members of a key organization in the field (e.g., ref. 2), or (iii) authors of peer-reviewed articles in the field identified from literature databases (e.g., refs. 50,91). We here did not solely rely on one of those sampling procedures but used all three procedures. We additionally checked for potential inconsistencies in results across procedures to increase the reliability of our results (Supplementary Fig. 7).

Our target population were leading research experts for pest management globally, and for all agricultural regions (North America, South America, Europe, Africa, Asia, Oceania), agricultural production systems (general, arable crops, horticultural crops), and all relevant scientific disciplines. The latter included Pest Management Sciences (e.g., Crop Pathology, Weed Sciences, Entomology, Agronomy), Ecology (Agroecology), Toxicology (e.g., Human Toxicology, Environmental Toxicology, and Environmental Sciences), and Social Sciences or Economics (see Table 2 for an overview). We focused on scientific experts, as they are generally most familiar with evaluating current and future scenarios, as well as the potential effects and trade-offs on a regional or country-level. But we also checked results against a subsample of non-academic experts.

We collected a total number of 473 responses and achieved subsamples of $N > 30$ for all key regions and disciplines, except for the region of Oceania ($N = 12$, also has a smaller population of relevant experts than the other regions). Our sample size is thus amongst the highest across comparable studies conducted in the field or adjacent fields (see Supplementary Note 2). Further, survey respondents cover (i) key global hotspots of food production, (ii) and countries with a combined share of 88%, 85%, and 90% of global cereal, fruit, and vegetable production. The large majority of survey respondents are further tenured, senior researchers with a high scientific output in the field, working at public research institutions. See Supplementary Note 2 and Table 2 for a detailed description of sampling procedures and characteristics of survey respondents.

## Data preparation and external data

We first anonymized survey data and excluded 30 responses that had not at least completed section 3 of the survey (on potential effects of a transformation), leaving us with a sample of 473 responses.

To empirically evaluate heterogeneity and drivers of expected effects across the gradient of agricultural production systems and regions, we merged the survey data with external data on regional development (Human Development Index, Planetary pressure-adjusted Human Development Index, GDP per capita), progress on associated SDGs, and characteristics of the farming system. These include potential pest damages (pest pressure is a key variable for farmers crop protection choices[1]), attainable crop yields (an indicator of agricultural development and potential losses to pests[92]), agricultural productivity in output value per hectare[4], and current pesticide pollution (an indicator for the type and intensity of current pesticide use[40]). The databases used here are reliable, since they are either based on information from international, public organizations or scientific publications and have previously been used in global analyses (e.g., refs. 35,77). Further, we selected SDG indicators fitting the respective domains of our 24 indicators (see Supplementary Table 6) to assess the relation between current state of SDGs and potential improvement through a transformation of pest management. We further assessed the relation with the Human Development Index and the Planetary pressure-adjusted Human Development Index. They are indicative of which countries would benefit most from a transformation.

We merged data as follows: The geographic scope of expertise indicated by experts is either (i) the country level, (ii) the continental level, or (iii) the global level. All external data are available on the country level, for some data sources aggregate values per continent and globally are already provided. We dealt with missing data on different aggregation levels as follows. If aggregated data on a

continent or global level was not provided or partially missing, aggregate values were computed based on means of country-level values and weighted by variable-specific measures, in line with source-specific methodology (e.g., aggregate mean values of GDP per capita of a continent are weighted by country-level population). All responses were then matched, based on their specific aggregation-level. Non-matching country-level data were coded missing (NA). Due to the low number of NAs in external data, we do not expect an impact of data availability on our analyses. See Table 2. for an overview and descriptive statistics of all variables used in the analyses.

## Data analysis: descriptive analyses

In Table 1 we report the direction of mean effects per indicator, as well as their magnitude compared to mean values for all other indicators. We first compute sample means of indicators (Eq. (1)).

$$\bar{x}_{Sample, j=1} = \frac{1}{M*N} * \sum_{j=1}^{M} \sum_{i=1}^{N} x_{ij} \qquad (1)$$

Where $x_{ij}$ is the expected effect for indicator $j$, as indicated by respondent $i$ and $\bar{x}_{Sample, j=1}$ is the sample mean for indicator $j = 1$. We then repeat this procedure for all 24 indicators and report in Table 1, (i) the general direction of the expected effect per indicator (positive or negative sample mean), and (ii) if the sample mean lies in the lower, middle, or upper third of the distribution of the sample means of all indicators (see Supplementary Fig. 3 for the distribution of all indicator values). Note that we created an indicator variable for whether respondents have no expertise in a domain, i.e., respondents selected {expertise = 0}. Those responses are then coded missing (NA) for all indicators in this category for this respondent and are not accounted for (see Supplementary Fig. 12 for the distribution of expertise per domain and Supplementary Table 4 for the number of complete responses per indicator).

In Fig. 1, we present the relative deviation of mean responses per indicator to the mean for all indicators as follows (Eq. (2)).

$$R1_{j=1} = \frac{\frac{1}{N} * \sum_{i=1}^{N} x_{i, j=1} - \frac{1}{M*N} * \sum_{j=1}^{M} \sum_{i=1}^{N} x_{ij}}{\frac{1}{M*N} * \sum_{j=1}^{M} \sum_{i=1}^{N} x_{ij}} * 100 \qquad (2)$$

Where $x_{ij}$ is the expected effect for indicator $j$, as indicated by respondent $i$ and $R1_{j=1}$ is the relative difference in percent, between the global mean for indicator $j = 1$ and the global mean over all indicators in percent. This procedure is then repeated for all 24 indicators.

In Fig. 2, we present the relative deviation of mean responses per indicator between regional and global levels as follows.

$$\bar{x}_{j=1, Region = Europe} = \frac{1}{P} * \sum_{i=1}^{P} x_{i, j=1, Region = Europe} \qquad (3)$$

$$\bar{x}_{j=1, Region = Global} = \frac{1}{Q} * \sum_{i=1}^{Q} x_{i, j=1, Region = Global} \qquad (4)$$

$$R2_{j=1, Region = Europe} = \frac{\bar{x}_{j=1, Region = Europe} - \bar{x}_{j=1, Region = Global}}{\bar{x}_{j=1, Region = Global}} * 100 \qquad (5)$$

Where $x_{ij}$ is the expected effect for indicator $j$, as indicated by respondent $i$ and $\bar{x}_{j=1, Region = Europe}$ (Eq. (3)) and $\bar{x}_{j=1, Region = Global}$ (Eq. (4)) are the respective means for indicator $j = 1$ from respondents with expertise on European and global scales respectively. $R2_{j=1, Region = Europe}$ (Eq. (5)) is then relative difference in percent, between the mean response for Global and European levels for indicator $j = 1$. This procedure is repeated for all 24 indicators and six regions (North America, South America, Europe, Africa, Asia, Oceania).

We conduct various robustness checks on our descriptive results and main results remain robust throughout all checks (see Supplementary Figs. 5–7).

## Relations between country characteristics and indicators

We computed Kendall rank correlation coefficients (using complete observations) between expected effects per indicator and external, country- and regional-level data on (i) important, general indicators of human development (the United Nations' Human Development Index, the United Nations' Planetary pressures-adjusted Human Development Index and GDP per capita), (ii) as well as the state of related SDG indicators.

For assessment framework-SDG indicator pairs, we first identify for each assessment framework indicator, respectively, if an SDG target that comprises this potential effect exists (see Supplementary Table 6 for an overview). For these SDG targets, we then extract data on progress towards their achievement, indicated by levels of the respective SDG indicators (see Table 2 and Supplementary Table 6 for an overview), and assess their correlation with indicators from the assessment framework. Note that we did not identify country-level data on the progress of SDG Targets 2.4 (Adaptable, sustainable, and resilient agricultural practices) and 10.3 (Ensure equal opportunity and reduce inequalities of outcome). We expected that indicators from our framework and SDG indicators do not strongly correlate, since SDG indicators are often affected by a diverse set of drivers. The level of the SDG indicator for biodiversity is, for example, not only affected by the choice of agricultural pest management in a region but also by land use choices or pollution from industry[6].

We chose Kendall rank correlations as a nonparametric measure of rank correlation. It is more appropriate for data that is not normally distributed or has significant outliers (e.g., ref. 93). We tested significance of correlations with a non-parametric two-sided test against the null hypothesis of a zero correlation (e.g., ref. 94). We computed Kendall rank correlations and tests for significance using the "cor.test" function of the R base package "stats" in version 4.2.1[95].

This allowed us to differentiate potentials per region and production system, but also to test how far results on the expected effects of sustainable pest management are correlated to the state of related SDGs indicators. There are different rules of thumb on how to interpret the size of correlation coefficients[96]. suggest, based on the percentiles of a meta-analysis of empirical studies in cognitive and behavioral sciences, that correlation coefficients of ±.1, ±.2, and ±.3 should be interpreted as "weak", "moderate," and "strong" correlations.

## Regression analyses

To assess drivers of differences in the expected effects, while controlling for characteristics of respondents, we conducted multiple linear regression analyses (one for each of the 24 indicators from the assessment framework) using OLS as follows (Eq. (6)).

$$\boldsymbol{y}_{ij} = \boldsymbol{x}_{ij} \boldsymbol{\beta}_j^x + \boldsymbol{z}_{ij} \boldsymbol{\beta}_j^z + \boldsymbol{\varepsilon}_{ij} \qquad (6)$$

Where $y_{ij}$ is the expected effect for indicator $j$ and respondent $i$, and $x_{ij}$, $z_{ij}$, $\beta_j^x$, $\beta_j^z$ are vectors of variables of farming system and respondent characteristics and their regression coefficients, respectively. $\varepsilon_{ij}$ is the error term. We then test significance of regression coefficients with two-sided $t$-tests against the null hypothesis of regression coefficients being zero. We conducted regression analyses and tests for significance using the "lm" function of the package "stats" in R version 4.2.1[95].

We account for key characteristics of the production system. Namely, the type of production system, the percentage of maximum potential losses to pests in the region, the level of average attainable yields in the region, the level of environmental risks of currently used pesticides in the region, and the current implementation level of sustainable pest management collected in the survey. We control for

**Table 2 | Overview of all variables used in the analysis (N= 473)**

| Variable | Description | Source | Mean | SD | Min | Max |
|---|---|---|---|---|---|---|
| **External Data Sources** | | | | | | |
| GDP per capita | GDP (Gross Domestic Product) per capita in 2015 in $1000 US at constant 2015 prices. | World Bank, 2022[126] | 19.76 *$10^3$ | 20.23 *$10^3$ | 0.35 *$10^3$ | 105.5 *$10^3$ |
| HDI | United Nations Human Development Index in 2021. | United Nations, 2022[127] | 0.77 | 0.14 | 0.40 | 0.96 |
| PHDI | United Nations Planetary pressure-adjusted Human Development Index in 2021. | United Nations, 2022[127] | 0.67 | 0.09 | 0.39 | 0.82 |
| SDG_1.5 Indicator 1 | The total number of people affected by natural disasters per 10.000.000 in 2020. | World in Data, 2022[92] | 1.18 *$10^3$ | 3.25 *$10^3$ | 0 | 32.04 *$10^3$ |
| SDG_2.1 Indicator 1 | Prevalence of undernourishment in percentage of population in 2019. | World in Data, 2022[92] | 7.65 | 6.09 | 2.50 | 48.20 |
| SDG_2.3 Indicator 1 | Agricultural production in US$ per labor unit in 2019. | World in Data, 2022[92] | 85.59 *$10^3$ | 312.22 *$10^3$ | 0.32 *$10^3$ | 275.82 *$10^3$ |
| SDG_2A Indicator 1 | The agriculture shares of government expenditures, divided by the agriculture share of GDP. | World in Data, 2022[92] | 1.07 | 1.03 | 0.02 | 4.60 |
| SDG_3.9 Indicator 3 | Mortality rate of unintentional poisonings from hazardous chemicals per 100.000 in 2019. | World in Data, 2022[92] | 0.93 | 0.77 | 0.10 | 3.80 |
| SDG_6.3 Indicator 2 | Proportion of bodies of water with good ambient water quality. | World in Data, 2022[92] | 68.17 | 18.69 | 0 | 100 |
| SDG_8.4 Indicator 2 | Domestic material consumption (kg) per unit of GDP (constant 2015 US$) in 2019. | World in Data, 2022[92] | 1.52 | 1.38 | 0.10 | 9.14 |
| SDG_8.8 Indicator 1 | Non-fatal occupational injuries per 100,000 employees in 2015 | World in Data, 2022[92] | 1.37 *$10^3$ | 1.14 *$10^3$ | 0.13 *$10^3$ | 8.92 *$10^3$ |
| SDG_12.3 Indicator 1 | Food waste per capita (kg) at retail level in 2019. | World in Data, 2022[92] | 14.33 | 5.66 | 3.12 | 78.82 |
| SDG_12.4 Indicator 1 | Parties meeting their commitments and obligations in transmitting information as required by the Rotterdam Convention on hazardous waste and pesticides in 2020. | World in Data, 2022[92] | 83.06 | 13.61 | 8.62 | 98.28 |
| SDG_14.1 Indicator 1 | The share of satellite imagery pixels measuring chlorophyll-a within a country's exclusive Economic zone above the 90th percentile of the global baseline (2000-2004) in 2020. | World in Data, 2022[92] | 4.14 | 2.53 | 0.11 | 25 |
| SDG_15.1 Indicator 2 | The average proportion of freshwater Key Biodiversity Areas covered by protected areas in 2020. | World in Data, 2022[92] | 46.19 | 22.07 | 4.17 | 99.97 |
| SDG_15.5 Indicator 1 | The Red List Index [0,1] in 2020. | World in Data, 2022[92] | 0.82 | 0.09 | 0.62 | 0.99 |
| Relative attainable yield | The mean attainable yield of major crops grown in the region (out of maize, rice, wheat, potatoes, soybeans, sugarcane, cassava, sorghum), relative to their global maximum attainable yield. | World in Data, 2022[92] | 0.69 | 0.12 | 0.44 | 0.94 |
| Potential pest pressure | Potential maximal damages of pests and pathogens to major crops in the region. | Oerke, 2006[1] | 70.88 | 6.01 | 60 | 85 |
| Level Risk Score of Used Pesticides | Environmental risk of current pesticide use, measured with the Risk Score. | Tang et al., 2021[40] | 2.34 | 0.77 | 0 | 4.01 |
| **Data from the survey** | | | | | | |
| Implementation sustainable pest management | Current level of implementation of sustainable pest management (0-10), according to the above definition, in the production system of expertise. | Survey, Möhring et al., 2025 | 5.12 | 2.35 | 0 | 10 |
| Field of research | | Survey, Möhring et al., 2025 | | | | |
| BioTech | Research on alternative pest management solutions (biological, mechanical, etc.). | | 22 (5%) | | | |
| Ecol | Research on (agro-)ecology. | | 56 (12%) | | | |
| Other | Respondent from Extension services, farming, policymaking, or industry. | | 78 (16%) | | | |

**Table 2 (continued) | Overview of all variables used in the analysis (N= 473)**

| Variable | Description | Source | Mean | SD | Min | Max |
|---|---|---|---|---|---|---|
| PMS | Research in pest management sciences (e.g., weed sciences, entomology, plant pathology, agronomy) | | 226 (48%) | | | |
| Soc-Eco | Research in Social Sciences or Economics. | | 57 (12%) | | | |
| Toxi | Research in Toxicology (Human/Environmental) or Environmental Sciences. | | 34 (7%) | | | |
| Indicator assessment framework | Potential of sustainable pest management (-10 to +10) to… | Survey, Möhring et al., 2025 | | | | |
| FS1 | …increase the provision of food in sufficient quantities. | | 4.65 | 4.66 | -10 | 10 |
| FS2 | …increase the provision of diverse and nutritious food. | | 5.31 | 3.91 | -8 | 10 |
| FS3 | …increase the provision of safe food and feed. | | 6.41 | 3.45 | -8 | 10 |
| FS4 | …increase the resilience to extreme pest events resulting in hunger and food insecurity. | | 4.96 | 4.76 | -10 | 10 |
| Indicator assessment framework | Potential of sustainable pest management (-10 to +10) to… | Survey, Möhring et al., 2025 | | | | |
| ENV1 | …reduce current pesticide pollution of drinking water. | | 6.44 | 3.26 | -10 | 10 |
| ENV2 | …reduce current pesticide pollution of soils. | | 6.61 | 3.25 | -10 | 10 |
| ENV3 | …reduce current pesticide damages to marine ecosystems | | 5.53 | 3.81 | -10 | 10 |
| ENV4 | …reduce current pesticide damages to freshwater ecosystems. | | 6.70 | 3.24 | -10 | 10 |
| ENV5 | …reduce current effects of pesticide use on biodiversity loss. | | 7.11 | 3.05 | -6 | 10 |
| Indicator assessment framework | Potential of sustainable pest management (-10 to +10) to… | Survey, Möhring et al., 2025 | | | | |
| HH1 | …reduce pesticides' current acute and long-term effects on health of farm workers. | | 6.61 | 3.18 | -7 | 10 |
| HH2 | …reduce pesticides' current acute and long-term effects on health of residents and bystanders. | | 5.79 | 3.32 | -8 | 10 |
| HH3 | …reduce pesticides' current acute and long-term effects on health of consumers. | | 6.02 | 3.44 | -8 | 10 |
| HH4 | …restore trust in contaminated food - distorting food choices (e.g., fewer vegetables or fruits). | | 6.55 | 3.26 | -10 | 10 |
| Indicator assessment framework | Potential of sustainable pest management (-10 to +10) to… | Survey, Möhring et al., 2025 | | | | |
| SOC1 | …improve the equal distribution of all costs and benefits from pest management (e.g., profit gains vs. adverse environmental and health effects) on local scales (between actors in a region). | | 4.66 | 3.80 | -10 | 10 |
| SOC2 | …improve the equal distribution of all costs and benefits from pest management on global scales (between countries). | | 4.19 | 4.14 | -10 | 10 |
| SOC3 | …improve the capacity of all farmers to adapt to future increases in pest pressure. | | 5.40 | 4.05 | -10 | 10 |
| SOC4 | …improve equal access of all farmers to suitable education, tools, and technologies for pest management. | | 4.94 | 3.97 | -10 | 10 |
| SOC5 | …ensure safe working conditions (e.g., gear & education) for all farm workers in pest management. | | 6.12 | 3.39 | -10 | 10 |
| Indicator assessment framework | Potential of sustainable pest management (-10 to +10) to… | Survey, Möhring et al., 2025 | | | | |
| ECON1 | …increase the provision of affordable and efficient pest management solutions. | | 4.25 | 5.00 | -10 | 10 |
| ECON2 | …enable production systems that can sustain the livelihood of farmers in the short run. | | 3.17 | 4.78 | -10 | 10 |
| ECON3 | …enable production systems that can sustain the livelihood of farmers in the long run. | | 5.97 | 3.55 | -9 | 10 |
| ECON4 | …increase economic resilience to extreme pest events. | | 4.69 | 4.69 | -10 | 10 |
| ECON5 | …increase agricultural productivity growth threatened by pest damages. | | 5.37 | 4.06 | -9 | 10 |
| ECON6 | …reduce indirect costs of pest management (e.g., decreasing soil productivity and pollination, increasing resistances). | | 6.63 | 3.18 | -10 | 10 |

**Table 2 (continued) | Overview of all variables used in the analysis (N= 473)**

| Variable | Description | Source | Mean | SD | Min | Max |
|---|---|---|---|---|---|---|
| Type of crop | | Survey, Möhring et al., 2025 | | | | |
| Arable | Arable crops | | 134 (28%) | | | |
| General | All crops | | 184 (39%) | | | |
| Horticulture | Horticultural crops | | 155 (33%) | | | |
| Scope of expertise | | Survey, Möhring et al., 2025 | | | | |
| larger | On a level of a continent or globally. | | 212 (45%) | | | |
| specific | On a level of national or local production systems. | | 261 (55%) | | | |
| Importance category Food Security | Importance (0-10) of pest management for agricultural challenges in this category. | Survey, Möhring et al., 2025 | 7.77 | 2.09 | 0 | 10 |
| Importance category Environment | Importance (0-10) of pest management for agricultural challenges in this category. | Survey, Möhring et al., 2025 | 6.98 | 2.20 | 0 | 10 |
| Importance category Human Health | Importance (0-10) of pest management for agricultural challenges in this category. | Survey, Möhring et al., 2025 | 6.55 | 2.61 | 0 | 10 |
| Importance category Social Equality | Importance (0-10) of pest management for agricultural challenges in this category. | Survey, Möhring et al., 2025 | 6.29 | 2.57 | 0 | 10 |
| Importance category Economy and Livelihoods | Importance (0-10) of pest management for agricultural challenges in this category. | Survey, Möhring et al., 2025 | 7.92 | 1.82 | 1 | 10 |
| Strength expertise Food Security | Strength of expertise (0-10) in the category. | | 5.58 | 2.95 | 0 | 10 |
| Strength expertise Environment | Strength of expertise (0-10) in the category. | | 6.73 | 2.48 | 0 | 10 |
| Strength expertise Human Health | Strength of expertise (0-10) in the category. | | 5.12 | 3.01 | 0 | 10 |
| Strength expertise Social Equality | Strength of expertise (0-10) in the category. | | 3.98 | 2.91 | 0 | 10 |
| Strength expertise Economy and Livelihoods | Strength of expertise (0-10) in the category. | | 5.35 | 2.87 | 0 | 10 |
| Invitation type | | Survey, Möhring et al., 2025 | | | | |
| Network and organizations | Selected senior researchers and members of key international organizations in the field of pest management. | | 199 (42%) | | | |
| Literature | Invited corresponding author based on literature search of peer reviewed articles in the field of sustainable pest management. | | 274 (58%) | | | |

Note that respondents were allowed to opt out of answering if they did not have expertise for a domain or single indicators of the assessment framework in the survey. See Supplementary Table 4 for sample sizes per indicator.

important characteristics of the respondents, including the field of research, the scope of the expertise (local farming systems or regional/global expertise), the self-elicited expertise in the category of the indicator, and the self-elicited relevance of pest management for the category. This allowed us to point out differences and trade-offs in potential effects that are key for informing policy makers, researchers, and food-value chain actors. It further allowed us to compare hypotheses from the literature with survey results, and thus to further confirm the consistency of responses.

## Robustness checks

Since we simultaneously test multiple hypotheses in our analysis, there was a risk that the type I error rates (significance levels) no longer reflect the type I error rates of the multiple tests, i.e., that a multiple hypotheses testing problem arises. Bonferroni correction is a conservative method to correct the probability threshold and control the occurrence of false positives (e.g., ref. 97). It works by dividing the original probability threshold $\alpha$ by $n$, where $n$ is the number of tests within a *family of indicators*. If we defined the family of indicators as their respective domains (Food Security, ... etc.), the "safe" threshold using the Bonferroni correction (for $\alpha = 0.05$) is indicated in Fig. 3 by the $\alpha < 0.01$ threshold for all domains (except for the Economics category with 6 indicators, where it is $\alpha = 0.0083$). This indicates that results remain very stable. Notably, out of our main results, only the evidence for legacy effects of toxic pesticides would be weaker. If we define the family of indicators as all 24 indicators used in the assessment framework, the "safe" threshold using the Bonferroni correction (for $\alpha = 0.05$), would be $\alpha = 0.05/24 = 0.0021$. To give an overview on the robustness of results under this assumption, we provide an additional graph (Supplementary Fig. 13). We again find that our main results are robust.

## Reporting summary

Further information on research design is available in the Nature Portfolio Reporting Summary linked to this article.

## Data availability

A documentation of the complete survey text is provided in the supplementary materials of the manuscript. The data used in this study are available in the bonndata database under accession code https://doi.org/10.60507/FK2/XWSS9W.

## Code availability

A documentation of the complete R Code used for the analysis is provided together with the manuscript. It can be used together with the freely available data (see data availability) to reproduce all analyses and figures. The code used in this study are available in the bonndata database under accession code https://doi.org/10.60507/FK2/FE09XJ.

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

## Acknowledgements

We acknowledge support in the creation and translation of the survey by Philippe Mathys, Sarah Vogelsanger, Yanbing Wang, Viviana Garcia, Alice Dos Santos, and Hang Xiong. We thank Emmanuel Tolani and Juliette Morel for their support in collecting data. We thank Katarina Kliestenec, Nadine Trottmann, and Jeanne Tomaszewski for the support in organizing the project workshop, and all workshop participants for their contributions, especially Eileen Ziehmann, Lucca Zachmann, Viviana Garcia and Chloe McCallum. We thank Christian Pohl for facilitating one part of the workshop. The research included co-authors from all geographic areas targeted in the research and the targeted, relevant disciplines. Roles and responsibilities were agreed amongst co-authors ahead of the research. The survey has been approved by the ETH Zurich ethics council. No local authorizations were collected since the survey was hosted and stored at ETH Zurich and no local research was conducted on the ground. Regional research relevant to the study has been considered in discussions and citations, assured by co-authors from the respective regions. The authors acknowledge funding as follows: Swiss National Science Foundation Grant IZSEZ0 209440 (R.F.), INRAE Metaprogram SuMCrop (S.G.), Fresh and Secure Trade Alliance (FASTA, AM22000) (V.G.), and German Research Foundation (DFG) under Excellence Strategy Grant EXC-2070-390732324-PhenoRob (N.M., M.Q.). This publication was supported by the Open Access Publication Fund of the University of Bonn.

## Author contributions

N.M. and R.F. conceptualized the study. N.M., M.B., A.B., S.G., V.G., P.K., A.L., R.M., U.N., M.Q., P.S., C.S., W.V., and R.F. designed the assessment framework. N.M., M.B., A.B., S.G., V.G., P.K., A.L., R.M., U.N., M.Q., P.S., C.S., W.V., and R.F. designed and prepared the survey. N.M. conducted the survey and prepared the survey and external data. N.M. analyzed the data. NM visualized the results. NM and RF supervised the project. N.M., M.B., A.B., S.G., V.G., P.K., A.L., R.M., U.N., M.Q., P.S., C.S., W.V., and R.F. wrote the original manuscript and contributed to review and editing.

## Funding

## Competing interests

The authors declare no competing interests.
