## [Transparent Peer Review file · Nature Communications]

Expected Effects of a Global Transformation of Agricultural Pest Management

Corresponding Author: Dr Niklas Möhring

Version 0:

Reviewer comments:

Reviewer #1

(Remarks to the Author)

Dear Authors,

I have had the pleasure of reviewing your manuscript titled "Expected Effects of Transforming Agricultural Pest Management across Global Scales" submitted to Nature Communications. Your work provides a comprehensive global assessment based on survey evidence from 517 experts across key disciplines and regions worldwide. The study evaluates the heterogeneity of expected impacts across five domains: economic, human health, food security, social, and environmental.

This study is novel in its holistic approach and global scope, employing rigorous and well-executed statistical methods. This makes me believe that this is an important paper for the scientific community dealing with transforming agricultural pest management systems and mitigating the burden of pesticides on sustainable agriculture, the environment, and social health.

The results suggest that indicators related to human health and environmental quality have higher expected benefits than those related to short-term economic gains and food security. These interesting results emphasize the need for a holistic and multidisciplinary approach to transform them into actionable and tailored regional strategies, especially for policymakers.

Hence, I recommend this manuscript for publication in Nature Communications, but I do so subject to major revisions to enhance readability and focus. As it stands, it is dense, which makes reading it sometimes difficult and unclear. The manuscript would benefit from a more streamlined presentation to enhance readability, which includes reducing redundancy and focusing on the most critical findings and implications. Additionally, simplifying the methodology section by summarizing the key steps and referring to supplementary materials for detailed descriptions is advisable. These general recommendations are elaborated in more detail in the specific comments provided to the authors.

I look forward to seeing the revised version of your manuscript.
Thank you very much.

General comments

These general comments are valid for all sections of the study. While the English is generally good, the sentences tend to be lengthy with some grammar mistakes and quite a lot of repetition of terms. The text must be simplified and, above all, synthesized, made more direct, and easier to read. For example, breaking down complex and long sentences into shorter ones and avoiding redundant phrases would improve readability.

I believe it would be beneficial to have the paper reviewed by someone who was not involved in its writing, as there is a lot of repetition that hinders the text flow. Simplification is possible in most sections, so aim for clear, concise communication by avoiding long sentences.

Additionally, instead of having the last paragraph in the main text discuss limitations, please write a dedicated section for limitations. This will provide a clearer and more structured discussion of the study's constraints and how they might impact

the results and conclusions.

At the start (L87/88), the authors question “how policy and industry is how to effectively and efficiently support these policy goals of strongly reducing pesticide use, while minimizing crop losses to pests.” It would complement and strengthen the paper to add a section addressing the policy aspect. This section could specify potential current and future approaches to address this complex issue.

The titles of all subsections should be shorter and more punchy to capture attention effectively. As they stand, their importance is lost in the wiggle of words. For example, “Large data gaps on the potential effects of transforming global agricultural pest management - despite global reduction targets” (L72/73) could be: “Data gaps in global pest management transformation despite reduction targets” or something along these lines.

I have provided a non-exhaustive list of suggestions. I recommend that the authors apply them throughout the text:

- Although the work is about pest management, please avoid overusing this term as it makes the text heavy to read. Use the IPM acronym consistently for Integrated Pest Management.
- The phrase "expected effects" is used repeatedly; consider varying your language.
- Ensure consistency in terminology and punctuation.
- Streamline repetitive phrases describing the survey information.
- Ensure subject-verb agreement in sentences about missing information.
- Avoid using a wide spread of parentheses, e.g., (over), (reducing), etc.
- Remove unnecessary hyphens for better readability.
- Consider using acronyms or varying language to avoid repetition.

#####

Specific comments

Abstract

The abstract could benefit from a clearer structure, with a more distinct separation between the background, methods, results, and conclusions. While the abstract mentions "leverage points" and "combinations of actions", specific examples are missing. In addition, some sentences are lengthy and complex, making them difficult to follow. Breaking them down into shorter, clearer sentences would improve readability. For example, in L37-39 could be changed to: “Our study assesses the expected effects across five domains: economic, human health, food security, social, and environmental. We examine these effects in major agricultural production regions worldwide”

L35 (and 99): suggestion to modify to “Here, we provide the first global assessment...”

L38 (and 102): suggestion to modify to “...five domains (economic, human health, food security, social, and environmental) in major agricultural production regions.”

L42: I think that instead of “... a global transformation to sustainable pest management ...”, the text would flow better with “... transforming agricultural pest management could be an important nexus for addressing multiple sustainability challenge”

L37/39: The phrase "key disciplines and regions worldwide" is somewhat redundant with "main agricultural production regions worldwide" mentioned later. Rephrase one of the two.

L46: The phrase "controlling for important production system- and participant characteristics" could be clearer if rephrased, as the hyphenation is strange. Actually throughout the text the authors use hyphenation (too) often. Please revise

Main text

L60-61: "Effective management of agricultural pests is thus essential for food security and farmers' livelihoods." This point is repeated in the context of future importance without adding new information. Consider merging or refining for brevity.

L65: "Pesticides often provide affordable and efficient pest management solutions, which can be applied in a wide range of production contexts." The comma before "which" is unnecessary.

L67-68. Change to ““Additionally, heavy pesticide use can diminish long-term agricultural productivity.”

L69-71: "For example, due to negative effects on ecosystems services essential for agriculture, such as biocontrol potential (13-15), pollination (16) and soil functions (17), as well as increasing resistances of pests to widely used active substances (18)." This sentence is a fragment and should be restructured to complete the thought.

L75:78. Too long, it should be split into shorter and clearer sentences to improve readability

L80: "Not clear, what those “other means” are?.

L81-84: “However, global efforts to reduce and substitute pesticide use remain fragmented, lacking a coherent strategy to mitigate pesticide pollution and adverse effects (26). There is also little evidence of significant progress toward achieving pesticide risk and use reduction targets (e.g., 27, 28)”.

L87-88: This sentence could be more concise for better readability

L88:91. Too long, it should be split into shorter and clearer sentences to improve readability

L107-109: This sentence could be more concise for better readability. For example: "We first identify suitable indicators to assess the effects and trade-offs of reducing pesticide use and transforming global pest management systems." Use this as a guideline for other similar sentences throughout the text to align with the high writing standards of this journal.

L112-116: This sentence could be more concise for better readability

L116-120: (and other cases) no need to enumerate. "We identified 24 indicators across five domains: environment (ENV),"

L121:125. Too long, it should be split into shorter and clearer sentences to improve readability. Both scenarios must be clearly defined.

L130: potentials of transforming to "potential for transforming"

L134: "across different indicators" could be simplified to "across indicators.". idem for other cases

L145-148: This sentence could be more concise for better readability

L148-149: where?

L151-153: please simplify to "We lack global estimates and regional heterogeneity data for all 24 key indicators of sustainable pest management (see Table 1)."

L202-204: Please support it with references

L2015-2019. Not clear. I do not see the connection with the previous sentence.

L307: such as?

Material and Methods

The methods described in the Materials and Methods section are generally solid and sound. The study design is comprehensive, data collection procedures are robust, data analysis techniques are appropriate, and validation methods are thorough. However it should be simplified and made clearer, and placing additional details in the supplementary material. The first part, where the surveys are described, should be straightforward and concise. A diagram would help the reader understand the process better. As it stands, it does not flow well.

Specifically:

L696-697: I am still wondering what is the "other survey and external data"

L710-....: The process of defining the baseline scenario could be described in more detail to understand how the current situations of pest management in different regions were identified and characterized.

L710-....: avoid using so often the word "elicit*". Idem for "concepts, approaches, technologies,"

L754-757. The main points are unclear. Could you clarify the primary message and its significance?

L771-... The study effectively identifies relevant dimensions for assessing transformations in agricultural production systems by reviewing high-level studies and literature on related topics. However, providing a brief summary of each dimension's relevance and its contribution to the overall assessment could enhance clarity.

In addition, although the selected indicators (societal and policy relevance, relevance across diverse production systems, representativeness of potential effects, and distinctiveness per category) are well-defined and appropriate. I expected to see example of other indicators that were considered but not selected, and explaining why. This could provide additional context and validation for the chosen indicators.

L836-... The creation of the online survey based on the assessment framework is well-executed and methodologically sound. The survey design follows best practices, is structured to gather comprehensive data, and includes thorough validation and pre-testing processes. However, additional details on the development of survey questions, interpretation of responses, and specific feedback from pre-testing would enhance the methodology's clarity.

The survey is divided into five main parts: expert information, perceived importance of pest management, current state of sustainable pest management, potential effects of transformation, and potential tools and policies for transition. Adding a brief rationale for each part and its contribution to the overall research goals would also improve clarity. Providing more detail on how the self-elicited expertise scores were validated and used in the analysis would also strengthen this section.

L921-... I commend the authors for their extensive work in selecting the target population. Focusing on scientific experts ensures that respondents are knowledgeable and capable of evaluating current and future scenarios, as well as potential trade-offs. However, including specific criteria for selecting experts and ensuring balanced representation from each region and discipline would be advisable. As well as providing more information on how non-academic experts were selected and how their responses compared to those of academic experts.

L944... Finally the external data. First point, I think this section should be moved up in the article, as external data is referenced several times earlier without clear direction, leaving the reader uncertain. This section could also be expanded. Providing more information on the sources and reliability of the external data, and any potential limitations or biases. Additionally, clarifying the process for handling cases where aggregated data were unavailable or partially missing, and how this might impact the analysis.

The authors have selected several relevant external variables (e.g., HDI, GDP per capita, potential pest damages, attainable crop yields, agricultural productivity, current pesticide pollution). However, a discussion on the rationale behind selecting these specific variables and their relation to the expected effects of sustainable pest management would improve the understanding of the analysis.

Data analysis

The data analysis methods described are statistically relevant and sound. The use of descriptive analyses, Kendall rank correlation, and multiple linear regression with appropriate control variables ensures a robust approach to understanding the potential effects of sustainable pest management. The inclusion of multiple hypotheses testing with Bonferroni correction further strengthens the reliability of the results.

(Remarks on code availability)

General comments

These general comments are valid for all sections of the study. While the English is generally good, the sentences tend to be lengthy with some grammar mistakes and quite a lot of repetition of terms. The text must be simplified and, above all, synthesized, made more direct, and easier to read. For example, breaking down complex and long sentences into shorter ones and avoiding redundant phrases would improve readability.

I believe it would be beneficial to have the paper reviewed by someone who was not involved in its writing, as there is a lot of repetition that hinders the text flow. Simplification is possible in most sections, so aim for clear, concise communication by avoiding long sentences.

Additionally, instead of having the last paragraph in the main text discuss limitations, please write a dedicated section for limitations. This will provide a clearer and more structured discussion of the study's constraints and how they might impact the results and conclusions.

At the start (L87/88), the authors question "how policy and industry is how to effectively and efficiently support these policy goals of strongly reducing pesticide use, while minimizing crop losses to pests." It would complement and strengthen the paper to add a section addressing the policy aspect. This section could specify potential current and future approaches to address this complex issue.

The titles of all subsections should be shorter and more punchy to capture attention effectively. As they stand, their importance is lost in the wiggle of words. For example, "Large data gaps on the potential effects of transforming global agricultural pest management - despite global reduction targets" (L72/73) could be: "Data gaps in global pest management transformation despite reduction targets" or something along these lines.

I have provided a non-exhaustive list of suggestions. I recommend that the authors apply them throughout the text:

- Although the work is about pest management, please avoid overusing this term as it makes the text heavy to read. Use the IPM acronym consistently for Integrated Pest Management.
- The phrase "expected effects" is used repeatedly; consider varying your language.
- Ensure consistency in terminology and punctuation.
- Streamline repetitive phrases describing the survey information.
- Ensure subject-verb agreement in sentences about missing information.
- Avoid using a wide spread of parentheses, e.g., (over), (reducing), etc.

- Remove unnecessary hyphens for better readability.
- Consider using acronyms or varying language to avoid repetition.

Specific comments

Abstract

The abstract could benefit from a clearer structure, with a more distinct separation between the background, methods, results, and conclusions. While the abstract mentions "leverage points" and "combinations of actions", specific examples are missing. In addition, some sentences are lengthy and complex, making them difficult to follow. Breaking them down into shorter, clearer sentences would improve readability. For example, in L37-39 could be changed to: "Our study assesses the expected effects across five domains: economic, human health, food security, social, and environmental. We examine these effects in major agricultural production regions worldwide"

L35 (and 99): suggestion to modify to "Here, we provide the first global assessment..."

L38 (and 102): suggestion to modify to "...five domains (economic, human health, food security, social, and environmental) in major agricultural production regions."

L42: I think that instead of "... a global transformation to sustainable pest management ...", the text would flow better with "... transforming agricultural pest management could be an important nexus for addressing multiple sustainability challenge"

L37/39: The phrase "key disciplines and regions worldwide" is somewhat redundant with "main agricultural production regions worldwide" mentioned later. Rephrase one of the two.

L46: The phrase "controlling for important production system- and participant characteristics" could be clearer if rephrased, as the hyphenation is strange. Actually throughout the text the authors use hyphenation (too) often. Please revise

Main text

L60-61: "Effective management of agricultural pests is thus essential for food security and farmers' livelihoods." This point is repeated in the context of future importance without adding new information. Consider merging or refining for brevity.

L65: "Pesticides often provide affordable and efficient pest management solutions, which can be applied in a wide range of production contexts." The comma before "which" is unnecessary.

L67-68. Change to ""Additionally, heavy pesticide use can diminish long-term agricultural productivity."

L69-71: "For example, due to negative effects on ecosystems services essential for agriculture, such as biocontrol potential (13-15), pollination (16) and soil functions (17), as well as increasing resistances of pests to widely used active substances (18)." This sentence is a fragment and should be restructured to complete the thought.

L75:78. Too long, it should be split into shorter and clearer sentences to improve readability

L80: "Not clear, what those "other means" are?."

L81-84: "However, global efforts to reduce and substitute pesticide use remain fragmented, lacking a coherent strategy to mitigate pesticide pollution and adverse effects (26). There is also little evidence of significant progress toward achieving pesticide risk and use reduction targets (e.g., 27, 28)".

L87-88: This sentence could be more concise for better readability

L88:91. Too long, it should be split into shorter and clearer sentences to improve readability

L107-109: This sentence could be more concise for better readability. For example: "We first identify suitable indicators to assess the effects and trade-offs of reducing pesticide use and transforming global pest management systems." Use this as a guideline for other similar sentences throughout the text to align with the high writing standards of this journal.

L112-116: This sentence could be more concise for better readability

L116-120: (and other cases) no need to enumerate. "We identified 24 indicators across five domains: environment (ENV),"

L121:125. Too long, it should be split into shorter and clearer sentences to improve readability. Both scenarios must be clearly defined.

L130: potentials of transforming to "potential for transforming"

L134: "across different indicators" could be simplified to "across indicators.". idem for other cases

L145-148: This sentence could be more concise for better readability

L148-149: where?

L151-153: please simplify to "We lack global estimates and regional heterogeneity data for all 24 key indicators of sustainable pest management (see Table 1)."

L202-204: Please support it with references

L2015-2019. Not clear. I do not see the connection with the previous sentence.

L307: such as?

Material and Methods

The methods described in the Materials and Methods section are generally solid and sound. The study design is comprehensive, data collection procedures are robust, data analysis techniques are appropriate, and validation methods are thorough. However it should be simplified and made clearer, and placing additional details in the supplementary material. The first part, where the surveys are described, should be straightforward and concise. A diagram would help the reader understand the process better. As it stands, it does not flow well.

Specifically:

L696-697: I am still wondering what is the "other survey and external data"

L710-...: The process of defining the baseline scenario could be described in more detail to understand how the current situations of pest management in different regions were identified and characterized.

L710-...: avoid using so often the word "elicit*". Idem for "concepts, approaches, technologies,"

L754-757. The main points are unclear. Could you clarify the primary message and its significance?

L771-... The study effectively identifies relevant dimensions for assessing transformations in agricultural production systems by reviewing high-level studies and literature on related topics. However, providing a brief summary of each dimension's relevance and its contribution to the overall assessment could enhance clarity.

In addition, although the selected indicators (societal and policy relevance, relevance across diverse production systems, representativeness of potential effects, and distinctiveness per category) are well-defined and appropriate. I expected to see example of other indicators that were considered but not selected, and explaining why. This could provide additional context and validation for the chosen indicators.

L836-... The creation of the online survey based on the assessment framework is well-executed and methodologically sound. The survey design follows best practices, is structured to gather comprehensive data, and includes thorough validation and pre-testing processes. However, additional details on the development of survey questions, interpretation of responses, and specific feedback from pre-testing would enhance the methodology's clarity.

The survey is divided into five main parts: expert information, perceived importance of pest management, current state of sustainable pest management, potential effects of transformation, and potential tools and policies for transition. Adding a brief rationale for each part and its contribution to the overall research goals would also improve clarity. Providing more detail on how the self-elicited expertise scores were validated and used in the analysis would also strengthen this section.

L921-... I commend the authors for their extensive work in selecting the target population. Focusing on scientific experts ensures that respondents are knowledgeable and capable of evaluating current and future scenarios, as well as potential trade-offs. However, including specific criteria for selecting experts and ensuring balanced representation from each region and discipline would be advisable. As well as providing more information on how non-academic experts were selected and how their responses compared to those of academic experts.

L944-... Finally the external data. First point, I think this section should be moved up in the article, as external data is referenced several times earlier without clear direction, leaving the reader uncertain. This section could also be expanded. Providing more information on the sources and reliability of the external data, and any potential limitations or biases. Additionally, clarifying the process for handling cases where aggregated data were unavailable or partially missing, and how this might impact the analysis.

The authors have selected several relevant external variables (e.g., HDI, GDP per capita, potential pest damages, attainable crop yields, agricultural productivity, current pesticide pollution). However, a discussion on the rationale behind selecting these specific variables and their relation to the expected effects of sustainable pest management would improve the understanding of the analysis.

Data analysis

The data analysis methods described are statistically relevant and sound. The use of descriptive analyses, Kendall rank correlation, and multiple linear regression with appropriate control variables ensures a robust approach to understanding the potential effects of sustainable pest management. The inclusion of multiple hypotheses testing with Bonferroni correction further strengthens the reliability of the results.

Reviewer #2

(Remarks to the Author)

General comments:

The important question of whether strong restrictions on synthetic pesticides would improve or worsen pest control along a long list of relevant outcomes has not been asked. Instead, the authors essentially asked if an improved pest control management labelled "Sustainable Pest Management" (defined as not or only marginally including synthetic pesticides, and as being superior to current management along the major dimensions) would improve pest control. The answer they received is that SPM would be better than synthetic pesticides. The question is tautological, the results are trivial. The assessment framework and drafting reveals in multiple instances ignorance of scientific evidence or basic logic. This includes, for example, misinterpretations of the causal roles of biodiversity and ecosystem services, pesticide residues in food, non-target adverse effects, land sharing vs. sparing, confounding of partial and total/net effects, and more. Some of the cited references do not support what the manuscript claims (checked on a random basis, not exhaustively).

Specific comments:

36: How reliable are those expert views? How reliable are their impact predictions? It could have been good to include some expert quality checks (asking basic knowledge questions on pest management for which only one correct answer exists).

46: Aren't higher yields better for the environment? Isn't higher production and protection from pests better for human health? The framing seems to be biased on the benefits side- the expectation for ambitious restrictions on widespread pesticide use would be to impose a cost on humanity, not provide benefits. And it misses the important point that in many developed countries, pesticide use does not pose relevant health risks because of food safety and occupational health standards.

47: Be more concrete, what is this different pathway?

50: "delivering"- developing? researching? innovating?

51: "political boundary conditions"- Unclear what is meant. Subsidies? But aren't subsidies costly and prone to mismanagement?

56: Weeds should be explicitly mentioned here to avoid the potential misunderstanding that it's only about animal pests and diseases. According to your first cited source, potential yield losses are much higher from weeds than from animal pests and pathogens (despite weeds having a wider range of effective management options).

57: It would be important to make clear here whether you are considering pre-harvest losses only or also post-harvest (post-harvest losses are particularly important in developing countries, and can be alleviated with higher pesticide use to the extent that storage pests matter).

58: These are quantity losses, for quality losses, any estimates?

59: Where does the 29% come from? The abstract of the cited paper mentions 26% as the lower bound (for soybeans)

61: Higher yields, to which pesticides contribute, are also essential to keep the agricultural area smaller than it otherwise would be (<https://ourworldindata.org/yields-vs-land-use-how-has-the-world-produced-enough-food-for-a-growing-population>)- with positive impacts on the size of the natural environment and other potential uses of land. If you like biodiversity, it should also be mentioned as a potential benefit of pesticide use through the yield-land-sparing mechanism (as natural areas tend to have higher biodiversity than agricultural areas).

67: The cited evidence is too weak to support the statement "pesticides adversely affect ecosystem functioning [and] biodiversity". Generally, evidence is missing to support those statements because the accounting of the species potentially affected (positively or negatively) by pesticide use is hopelessly incomplete- citing adverse effects on an extremely small share of those species does not constitute evidence that biodiversity is "adversely affected". More importantly, biodiversity

has no causal relationship with ecosystem functioning or ecosystem services (see Maier, 2023: Chasing Biodiversity Off the Scientific and Conservation Tracks. doi.org/10.3998/ptpbio.4337; Schoolmaster Jr., et al, 2020: A graphical causal model for resolving species identity effects and biodiversity-ecosystem function correlations. doi.org/10.1002/ecy.3070). The statement in the manuscript linking pesticide use, ecosystem functioning and biodiversity is therefore unsupported, highly misleading and should be removed.

To the extent that pesticides increase effective yield, they can also positively contribute to the sparing of natural areas from agricultural land expansion.

67: The first cited source (9) refers to human health outcomes. However, the source states: "studies with more refined measures of agricultural pesticide exposure and/or other outcomes have generally reported null or inconsistent effects of exposure on birth defects, low birth weight, and gestational length", citing a handful of studies. The results from the study (9) also provide only limited evidence of a general harmful effect on actual pesticide use (statistically significant effect only for top 1% of exposure quantities, i.e. extremely high and very rare applied quantities)- the cited evidence thus does not support the broad-brush statement in the manuscript that "pesticides adversely affect human health", this needs to be much qualified or "human health" must be removed. The statement in the manuscript is also providing undue fuel to the widespread misconception that pesticide residues in food represent significant health risks (they do not).

68: This is a highly misleading sentence that should be removed or substantially qualified.

Biocontrol can act as a complement or substitute to chemical and mechanical pest control. While it may be true that pesticide use potentially reduces biocontrol (e.g. by reducing populations of non-target species that are helpful in pest control), it does not follow that overall agricultural production potentials are reduced. If pesticide use improves pest control through its direct effects on target-organisms more than it worsens pest control through its indirect effects on non-target organisms, then the net impact of pesticide use on pest control is still positive.

Similar considerations apply to soil functions. The cited reduction in P uptake as a result of fungicide applications could also be balanced with studies showing that pesticide use increases water or N use efficiency.

The threat to agricultural production from pesticide use reducing pollination is much overstated here in light of the available evidence and facts. The cited source (16) itself states: "The few available true field studies assessing the effects of field-realistic exposure on pollinators provide conflicting evidence of their effect" and "Substantial gaps in knowledge remain regarding the effects of pesticides on pollinators". Importantly, and similar to the biocontrol case, even if pesticides did meaningfully reduce pollination, their net impact on agricultural production might still be positive, namely if the pest control afforded by them brings greater yield gains than the losses through reduced pollination. Some other relevant facts: Sixty percent of global agricultural production comes from crops that do not depend on animal pollination (doi.org/10.1098/rspb.2006.3721). For crops depending on animal pollination, far less than 100% of the yield is due to animal pollination, e.g. 25% of the soybean and rapeseed yield is due to pollination, for pulses and tomatoes only 5% (doi:10.2760/619793). Furthermore, the domesticated honeybee provides primary pollination services to crops highly dependent on animal pollination (e.g. melons, pumpkins, some exotic fruits/nuts), and global honeybee numbers are increasing (doi.org/10.1007/s13592-020-00788-9). The domesticated honeybee is a managed species to whom pesticide use poses no outsized threat. Most wild bee species, which are less amenable to direct management, make no significant contribution to crop production (doi.org/10.1038/ncomms8414).

74: The policy targets are not necessarily rational, you have to consider decades of environmentalist campaigning, rather than (scientifically validated) "issues".

77: Please note that in May 2024 the Commission withdrew the proposal. (<https://www.europarl.europa.eu/legislative-train/spotlight-JD22/file-sustainable-use-of-pesticides-%E2%80%93-revision-of-the-eu-rules>)

78: More precisely, since when and in which world regions and how intense have these promotion efforts been?

83: This seems counterintuitive. Haven't older more toxic pesticides have been removed from the market over the last decades, at least in many developed countries, due to more stringent approval standards (e.g. in the EU)? And isn't the lack of detailed pesticide use and risk data (e.g. in many European countries) not the main point here, rather than the conclusion that the available data shows no reductions in overall risk? What is the best data source underlying the statement that no sizeable achievements in overall pesticide risk have been made in Europe or North America, for example? And I miss the point that it's less the applied quantity but rather the combined unmanaged quantity-toxicity ("overall risk"?) what matters - although policy might indeed mis-focus on quantities.

86: The harmonised risk indicator 1 published by the European Commission shows a substantial decrease during 2011-2021 for the EU (https://food.ec.europa.eu/plants/pesticides/sustainable-use-pesticides/harmonised-risk-indicators/trends-eu_en). For Germany, SYNOPSIS shows a downward trend in risk. The available data might be insufficient for strong conclusions, but they point towards risk reductions rather than no reductions. So a bit more nuance would be necessary here.

92: There are several studies assessing the impact of the F2F strategy (https://agriculture.ec.europa.eu/document/download/69adfd09-3836-4a9e-b673-4435a56ec4cd_en), and also studies on organic farming expansion (e.g. doi.org/10.1038/s41467-019-12622-7)

102: You missed climate change mitigation and adaptation, an important omission.

117-118: Several overlaps exist among these domains

123: "sustainable"- This is inappropriate framing, as pest management without synthetic pesticide use might not be sustainable, at least in the short-medium term. The term, which is found across the key survey questions, is loaded and might bias answers towards favouring "sustainable PM". The definition provided in Section C of the questionnaire is not neutral ("sustainable PM cornerstone of sustainable agriculture", "meet society's current and future needs", "overreliance on synthetic pesticides"). This is a huge red flag.

The question might be ill-defined and misunderstood, casting doubts on the interpretation of the answers. Is this about restrictions on pesticide use? Or about the replacement with alternatives that are stipulated to be equally effective (then the answers are trivial)? What is the time-frame for the impact to materialise?

A more neutral and precise definition and question would have included "policy-imposed restrictions on the use of synthetic pesticides" and their extent (50%? 90%?) (clearly stating whether and what accompanying measures are implemented) and its impact at a specified point in time (2030? 2050?).

129: Best might still not be reliable enough, depending on the extent to which respondents' replies are accurate.

133: "Direction"- Effect size is of great importance too, why did you not ask about it?

133: "at global level"- How are trade interactions accounted for? If one region goes ahead with unsustainable pesticide reductions, others may pick up the slack.

141: Some more detailed information on the respondent characteristics (education levels, disciplines, institutions) would be useful here for the reader. Are these mostly leading scientists in relevant fields, professors at the best global universities?

160: But is it in practice? What exactly is the policy scenario here?

167 "Domain 1"- Where is climate change? Through supporting higher yields, pesticides can reduce agricultural land expansion/deforestation and GHG emissions.

167 "Domain 3"- The positive results of the expected direction of the global effect cast substantial doubt on the survey design (question framing) and/or the reliability of the expressed expert views.

The phrasing of the questions in this section are highly loaded in combination with the provided definition: the definition says that SPM is a cornerstone and meets current and future needs in terms of food and nutrition security, and the question then asks whether SPM is more or less suited to provide food in sufficient quantities etc. I do not believe that the design of this survey passed appropriate quality control before it was launched. Any textbook on survey design would advise against it

167: "FS3, +++"- This result makes very clear that the expressed views are not reliable indicators of the actual impact. According to the scientific evidence, actual levels of pesticide residues pose no significant threat to food safety. So the belief that a replacement of synthetic pesticide use with other pest control measures would substantially (+++, in relation to other expected effects) improve feed and food safety is simply not aligned with the science.

The first cited key reference (51) does not provide evidence on pesticide residues as a significant health risk, it cites another study (of dubious quality) which seems to refer to poisonings not from food residues but rather direct exposure. I strongly encourage the authors to omit this reference.

The other cited key reference (52) is not focused on pesticides but on chemicals more broadly. Of 470 (!) pesticides tested, only one (chlorpyrifos in smoked fish, unauthorised) could pose a human health risk. This reference is also clearly insufficient to serve as a model for the "expected effect" of a pesticide restriction policy.

167: "Domain 4"- The positive expected effects in this domain are also highly questionable as to their accuracy. (unless the question is understood as the tautology that SPM is defined as better for everything compared to pesticides)

167: "Domain 5"- I am not going to check every reference but generally again when you define SPM as "ensuring economic and social equality" (while not doing the same for pesticide use) then don't be surprised if the answers say that SPM has a positive effect on equality.

167: "SOC1, 22"- The first cited reference (22) does not examine the impact of SPM versus pesticides on equality.

192: Again the authors confound a negative effect of pesticide use on ecosystem services with a (unsupported, speculative) negative effect of pesticide use on agricultural productivity.

205: This is an incorrect statement, as long you do not assume that SPM is equivalent or superior to pesticides on most dimensions (but then it's a tautological statement).

211-218: More plausible explanations are that the experts have no clue what they are talking about (so they are not real experts), or they are answering tautological questions (where under the provided definition only one answer is logically possible). The first explanation is plausible if you ask people who are not real experts in the given domain but in another. Having the expectation of a negative effect for "at least one indicator" across one or two(?) domains, and each indicator having (also) negative responses, are red herrings, as the relevant result is the mean expected negative effect for each indicator.

The last sentence on "well-documented adverse impacts" is again confounding the impact on a sub-outcome (e.g. pollination) with the net impact on the relevant total outcome (e.g. productivity), and it is the latter that the survey questions

are about, not the former.

239: "especially in these regions..." But you should again explain why. I do not see an extensive comparison with "findings from the literature", or another plausible explanations. Have you considered whether developed country experts might have more accurate beliefs than developing country experts, due to different cultures or education?

243: "benefits in the human health domain..."- What could account for this?

248: "We consistently find..."- What could be possible explanations?

259: If you have reliable experts at hand you should not expect to have respondent-specific characteristics driving results (only random variation around the mean)? Or is this a sociological study about belief systems across cultures and disciplines?

268: This "relative deviation" does not have an initiative interpretation, nor is clear why it is included here. Consider modifying the figure or better explaining it in the footnote, as each figure should be easy to understand on its own.

276: 12 is a questionable sample size

277: "relative to expected effects on global levels" Again this variable is not easy to understand. The logical thing would be to show the absolute results by region (where are these results?), not in relation to the global means.

286: What is the reason for controlling for key characteristics of the respondents? I would make that clear to the reader

321: "less beneficial"- This is counterintuitive

325: "We interpret these findings..."- Why this interpretation? It seems a bit far-fetched.

332-336: Not obvious that these are the most plausible explanations. You didn't ask the reasons for the experts opinions so this is speculation.

355-357: What does this tell us about the accuracy of experts' beliefs?

363-365: Well, or maybe one discipline is more right than another discipline?

370: Tautology again

372: "potential costs"- But your experts say the transformation would be beneficial across all dimensions including the economic and social. These costs are factored in your H ("SPM more affordable")

376: Full elimination of synthetic pesticide use?

379: I do not find this phrasing among the predefined measures in the questionnaire. Align.

387: But this also contradicts the notion that restricting pesticide use would lead to better outcomes. It underlines that the main questions of the survey are hypothetical, imagining a SPM that is nicer than synthetic pesticides yet does not exist.

428-429: But your experts say the opposite, namely that the transformation is great for equality! The transformation has no negative impacts if we take your results at face value.

430: Your results say nothing of pesticide leakage.

434: "important insights"- If the experts' beliefs are correct and you ask them the right questions, yes.

437-438: There are some regional assessments, for example for the EU's F2F strategy, which included reducing pesticide use. Or organic farming expansion scenarios. These assessments (indicating a negative impact on production and environment, possibly contradicting your experts) are themselves incomplete and flawed but should be mentioned in your manuscript.

699: Why? Post-harvest losses are very important.

(Remarks on code availability)

Reviewer #3

(Remarks to the Author)

This work presents some interesting findings, but the current manuscript does not discuss important contextual, procedural, and definitional limitations of the work. These need to be addressed and the potential of biases that are introduced need to

be discussed before the work is published. Here are some of the issues:

The definition of “Sustainable Pest Management” as “no or minimal use of pesticides” can create a number of biases in the responses and results. The experts could well have interpreted this to be a definition that essentially limits “alternative pest management” to “no pesticide use”, which is even stricter than organic agriculture (where biologically-based pesticides are readily accepted and used). The use of the term “pesticides”, with no attempt to differentiate among different groups or risks is overly broad. Far more interesting would have been to deepen the conversation and response to a risk-reduction approach to pest management.

An important source of potential bias is the group of experts. To provide transparency about their interests and allegiances, data on their potential conflicts of interest should have been collected and reported, as is the norm in scientific publications. What has been the funding sources for their research? What organizations currently or have recently employed them? This information is fundamental in deciding how unbiased this group of experts is in their responses to the potential transition to using fewer high-risk products that are still owned and produced by powerful corporations in the plant protection arena. The analogy to the tobacco researchers who received massive funding from the tobacco industry 30-50 years ago is appropriate here. Without clearly dispelling potential sources of bias, the results of the surveys of this group of experts is questionable and potentially unreliable.

The role of the pesticide industry should be explicitly discussed as an impediment to the transition of more sustainable pest management. The huge industry lobbies, advocates, funds research, coerces, and uses other means to promote their interests in maintaining the status quo to sell more of their products. They have successfully imposed dominant narratives of the “need” for pesticides. This powerful narrative is typically unquestioned by many experts in the field. Another weakness of the survey was to not differentiate among different production systems within regions. Smallholder maize farmers in sub-Saharan Africa face a completely different socio-economic context than large-scale commercial maize farmers in southern Africa.

Another approach to answering the question about the transition from “status-quo” crop protection to alternatives, would be to use an empirical methodology. This could be done by looking at “status-quo” crop protection methods and results and compare them to farmers who use organic pest management methods, so that they can receive organic certification for their crops. The worldwide production of organic crops is now large enough that many such directly comparable (same crops, same regions, etc.) could be found and compared. The evidence would be far more conclusive than the responses from a potentially biased group of responders to a blunt survey.

(Remarks on code availability)

Version 1:

Reviewer comments:

Reviewer #1

(Remarks to the Author)

Dear Authors,

Thank you for addressing my comments and suggestions in your revised manuscript titled “Expected Effects of Transforming Agricultural Pest Management across Global Scales.” I appreciate the effort you have put into improving the clarity, structure, and overall quality of the manuscript.

The revisions have, IMHO, significantly strengthened the paper, and I am pleased with how you have addressed the points I raised. Your detailed responses demonstrate a thorough consideration of the feedback, and the manuscript now reads more cohesively.

I have raised a few additional comments and suggestions in my latest review, such as refining (still) long sentences and addressing other minor points noted in your response letter. However, these are minor adjustments.

Congratulations on producing such an impactful and comprehensive piece of work. Your study provides valuable insights into a critical global challenge and will be a significant contribution to the field.

I wish you all the best with the final stages of publication and your future research endeavours.
With my best regards

General comments

As mentioned above, the revisions have addressed most of the previously raised concerns, and the manuscript is now a much stronger contribution to the field. However, I would like to provide some additional comments and suggestions to enhance the clarity and depth of the manuscript further.

1. Long Sentences and Readability. While readability has improved, some sentences remain dense and could benefit from

being broken down. For example:

"Combining agronomic developments for improved and diversified production systems with new opportunities from digital innovations and genomic breeding techniques is seemingly a promising avenue to increase productivity while significantly reducing pesticide use and risk".

2.The terms "current status" and "current situation" appear interchangeably throughout the manuscript. It would be beneficial to homogenize this terminology for consistency.

3.The legend for Figure 3 requires further explanation to ensure clarity.

4.The policy discussion section would benefit from more actionable and specific recommendations. For instance, the discussion could highlight concrete policy steps or tools to facilitate the transition to sustainable pest management.

5.L410-413. The sentence "In general, alternative pest management requires combinations of management actions implemented at various scales and adapted to the production context, while pesticides are often more broadly applicable (e.g., 30, 32, 36 for a discussion)" is unclear and could be rephrased for better understanding.

6.While the section "Holistic Policy Approaches Required" identifies alternative solutions as a priority, it does not specify examples or evidence-based approaches for alternatives. For instance, it would be helpful to include examples of proven or promising non-pesticide strategies that could be scaled globally.

7."Locally Adapted Bundles of Support Measures": This phrase is quite broad. Do you have examples of successful case studies or pilot programs? This would make the argument more robust and relatable.

8.The section refers "Holistic Policy Approaches Required" to "our results" multiple times without explicitly connecting these to specific findings. For example, which results demonstrate the insufficiency of technical feasibility for alternatives?

9.Still on the section "Holistic Policy Approaches Required". I think it would be beneficial to reflect on specific policy challenges faced by low-income (limited regulatory capacity) versus high-income regions.

Reviewer #2

(Remarks to the Author)

Dear authors,

I remain unconvinced by your survey design. The answers you received from the survey respondents were already baked into the definition of the target scenario. To get reliable results, a good survey should have avoided the type of biased framing you chose.

In addition, your text on pesticide effects on biodiversity, ecosystem services, agricultural production and human health is still riddled with errors and obfuscations.

Best regards

2.1

This is not the definition used in the manuscript, nor in the survey. The manuscript states: „pest management systems with no or minimal use of pesticides, using (a combination of) ... alternative pest management practices...their implementation leading to a substantial reduction of current pesticide use levels“.

The survey questionnaire's definition under C also includes: „SPM cornerstone of sustainable agriculture", "meet society's current and future needs", „ensuring food security“ etc., "overreliance on synthetic pesticides" / „minimal use“.

The crucial point is that the definition has the superiority of the SPM baked into it, so the answer can only be that it is superior. The authors cannot redress this major mistake made during the survey design. They do not respond to it here but the proper way to deal with it is to admit the study is gravely flawed and not seek publication.

2.11

The authors ignore the positive impact of pesticide use (through higher yields) on biodiversity.

2.12

Pesticides can also positively affect biodiversity and ecosystem functioning (through the land-sparing mechanism).

Negative effects on selected individual species, but the evidence does not support a negative effect on biodiversity (the number of different species).

The manuscript does not acknowledge the fact that currently pesticides have a positive effect on biodiversity via the land-sparing mechanism.

Adversely affecting ecosystem services is different from adversely affecting ecosystem services through adversely affecting biodiversity. The sentence in the revised manuscript „However, pesticides can adversely affect biodiversity and ecosystem functioning” obfuscates that distinction. Taking into account evidence for demonstrated harm of pesticide use on certain animals, the revised manuscript sentence also still confuses individual species with biodiversity. And the sentence fails to acknowledge positive impacts of pesticides on biodiversity via the land-sparing mechanism.

The sentence is thus still very bad and should not be published as it stands.

You should also be careful to distinguish “ecosystem function” from “ecosystem services”, they are not the same thing.

2.13

„Can“ is weak and obfuscating. It would be better to be more precise (health effects not causally proven and very small in Jones 2020, no effects at a broad range of normal pesticide use quantities in Larsen et al 2017, old and no longer used substances in Fletcher & Nogh. 2024; Frank 2024 +7.9% effect size in the context of the low base rate of ~7/1000 births extremely small).

If there are negative health effects from pesticide use, they appear to be very small and occur only in extreme or historical use cases. Furthermore, positive effects of pesticide use (via improved diet and nutrition, in poorer farming contexts also through higher household income) should be mentioned alongside (<https://doi.org/10.1016/j.cropro.2007.03.022>).

2.14

„this“ can be read as linking back to biodiversity. But the link of biodiversity and agricultural production via natural enemies, pollination and soil function is not established, it is individual species that are driving these effects, not the diversity of different species present. And again, ecosystem services do not automatically follow from ecosystem function (imagine an ecosystem that is very functional at harbouring -if you want also a very diverse set of- crop pests and diseases- that is a clear disservice to agriculture).

2.15

I still have not seen good evidence showing that pesticide use decreases yields by killing off natural enemies / pollinators / soil functions at sufficient scale to counterbalance the positive yield impact of pest control/nitrogen/water use efficiency afforded by pesticides. This should be acknowledged in the manuscript.

2.17

The legislative proposal was withdrawn and whether another attempt at legislation will be made is highly doubtful. It would be appropriate to mention the withdrawal or simply not mention the 50% goals anymore.

The CBD is concerned with risk, not quantities.

2.21

Peer-review in the sense of a publication in a scientific journal should not be the only criterion when deciding whether to mention or not to mention a study. Many studies published in journals are terrible and important data and evidence can be found outside these journals. In this case I would argue that these F2F study results are important and should be mentioned in the manuscript. If you state limitations of these studies, you should do the same for many of the studies you cite when you talk negatively about pesticides (see comments 2.12-2.15).

2.24

I remain unconvinced. The survey framing is strongly biased and the answers already baked into the definition.

Version 2:

Reviewer comments:

Reviewer #4

(Remarks to the Author)

Dear Authors,

The Reviewer expresses its great appreciation, congratulation that you targeted such an important topic, with overarching approaches, developing questionnaire, collecting over 500 responses globally (based on selected global regions).

Extremely difficult task, enormous differences globally. But the scientific (and broader) community needs your planned paper.

First of all, the Reviewer is less competent for the data analysis, methods but he should like to support your Manuscript (improvement) from the viewpoint of readers (be it scientists, policy makers, students, environmentalists, etc).

Reviewer aims at contributing to better common understanding of the survey, its results and subsequent interpretation.

L 52: "insect pests".... Please change for "animal pests" otherwise you disregard slugs, mites, birds, vertebrates, etc. UN FAO IPPC and EU definitions say "pests are all harmful organisms".

L 60 (and later in the entire text: "pesticides" as a definition is fully misunderstood or differently used globally. Reviewer understands Pesticides (to control pests, be it human disease vectors, urban rats, crop pests) and be it synthetic chemical plant protection product or natural plant extracts, plant oils, sulfur, microbes, entomopathogenic fungi, etc.

Reviewer suggests to DEFINE right in the beginning what "pesticides" stand for in your Manuscript, Assume "chemical/synthetic" active substances. Or?

L63: Ecosystem functioning seems to be a bit cloudy, maybe ecological functions and subsequent ecosystem services?

L 66: Reviewer strongly disagrees with "resistance development" as a specific risk. Resistance development in populations of target organisms is a "natural process", response of target populations to specific/multiple selection pressure (be it pesticide, agronomic practice, behavior, etc).

L 68-70: Increasing pest pressure under climate change is likely one sided approach. Some pests occur, others disappear, their natural enemies may develop their increased population, thus it is more complex.

L 75: "more hazardous" seems to be not scientific. Pls look at UN FAO and other definitions.

L77: "For a long time"... is a bit questionable. Maybe until recent years (2025)...

L 127 "minimal use": Reviewer suggest to avoid non-scientific "minimal"? use.

L 132 "global and societal": Rev. understands global as spatial scale, societal as context environment. Might be rephrased.

L 170-172: Missing where, when?

L 243: Reviewer calls the attention of the authors, that globally wrong fake news (input-intensive/extensive" phrasings are widespread. I strongly suggest to separate "material and energy input" from knowledge input.

L410: pls correct for NGT (New Genomic Technics), see relevant EU documents.

L 428: "Pesticides Free"??? Means free of natural products, microbials, etc? At least, pls correct the popular non-scientific and misleading wording.

L 462-465: Sorry, very misleading non-factual, non scientific sentence, pls. correct.

L 561-562: Pls rephrase (as per "pesticide" definition). Therefor NO??? or MINIMAL??? should be rephrased. (What is minimal?)

L 575: "Pests and Pathogens", pls rewrite. ("Pests, incl. plant pathogens") I assume authors did not want to include animal and human pathogens?

L 578: "Natural pest control" in non-scientific. The examples are fine but might be non-chemical pest control (Conservation, augmentative bioctrl, use of natural active substances, low risk products, etc. should be referred to. Good luck for publishing your great outputs.

Reviewer #5

(Remarks to the Author)

Expected Effects of a Global Transformation of Agricultural Pest Management

Comments

1. Lines 56-57: "for the five major food crops worldwide". Which are these five major food crops?

1. Lines 60-61: "2.6 million tons in 2020" The latest FAO data can be used

2. Lines 71-72: "the Post-2020 Global Biodiversity Framework of the United Nations sets a reduction target of 50% for overall pesticide risks until 2030". Is this target achievable or is it meant for only academic discussions? It has been five plus decades since integrated pest management was globally adopted as the cardinal principle of plant protection strategy, yet pesticide use in agriculture has surged, and the same has been highlighted by the authors. The authors have used "sustainable pest management" terminology and not "integrated pest management". IPM is based on the principle of "sustainable pest management"? The authors acknowledge this in the "Supplementary Information" at point 4. Although IPM endeavoured to promote sustainable pest management, thereby reducing synthetic pesticide use, it would have been better to analyse the obstacles to integrated pest management and the global transformation of IPM for sustainable agricultural production.

3. As happened with IPM, there are tens, if not hundreds, of definitions, and the same will be the fate of the concept "sustainable pest management". Authors could have focused on IPM to conduct a survey of experts AND FARMERS on "Transformation of Agricultural Pest Management". Sustainable pest management has been operationalised as "pest management systems with

4. no or minimal use of pesticides, using (a combination of) ... alternative pest management practices".

5. What are alternative adoptable pest management practices available with and disseminated by survey institutes that should have been part of the survey? Thereafter, a farmers' survey should have been conducted to assess the adoptability and sustainability of these practices at the farm level across different farming systems. Without involving farmers reached by the sampled institutes and 517 experts, the exercise is top-down. What are alternative pest management technologies for farmers in different global hotspots? The practices they adopt or reject, and pesticide use, would have made this survey impactful. What attitude farmers have towards alternative pest management tactics in different regions across different crops for reducing pesticide use and making pest management sustainable could have been another study variable?

6. 517 experts are from the international organisations (Supplementary Table 5). What have been the outcomes of programmes implemented by these organisations in "reducing pesticide use"? Experts from National Agricultural Research Systems of countries having large cultivated land should also have been part of the sampling plan.

7. This article is simply an academic exercise. We cannot test hypotheses solely based on the survey of 517 experts.

Experts' and farmers' data triangulation would have added to the robustness of the results for drawing inferences.

8. Data on "pesticide use and pests" should have been added from the selected countries of the 517 experts, which would add value to the article.

9. Highest potential of sustainable pest management in low-income regions: Why "Experts on highly productive and input-intensive production systems in Europe and North America envision lower potential benefits due to sustainable pest management -across nearly all indicators?" Many times, experts are far removed from the farms/ farm households. Experts mostly do not collect primary data themselves in the developing countries, so how can they conclude "..... potential win-win scenarios due to the reduction of pesticide use and the transformation of pest management" AND WHY EXPERTS FROM INDUSTRIALIZED AGRICULTURE SEE LOWER POTENTIAL. What about the input-intensive production systems in many Green Revolution areas of developing countries, for example, Indian Punjab and Haryana, which are highly productive?

10. Lines 162-170: In the results section, the authors are again explaining the rationale for the study. The results section should exclusively focus on survey results.

11. Lines 398-399: "identifying and providing access to effective and cost-efficient substitutes for pesticides". It is a generic statement. Experts should have been asked to list the i) effective, ii) adoptable, iii) farmers' compatible, and iv) cost-effective non-pesticidal pest management practices available.

12. Many statements in the survey text are double barrel statement/questions. For example: "What extent are sustainable

pest management.....as defined above already....in your area of expertise..”

How can a expert answer if one, two or three agronomic solutions are adopted?

Affordable and efficient. These are two constructs? Again, double barrel .

Environmental and health effects!

Version 3:

Reviewer comments:

Reviewer #4

(Remarks to the Author)

Dear Authors,

Thank you for addressing my remarks, questions in my first Review step. I feel you adequately finetuned all those lines, sentences, words I questioned With that readers will be able to better and understand (and understand equally, unambiguously) the context of your work and its key messages. Since I provided details, gighlighted the excellence, great value of your manuscript, I will not repeat it.

One point, what you mentioned several times in the corrected manuscript and in your response to the Reviwers, is indeed, your present manuscript is the first step that wiil and should be followed by regionalized, participatory (involvement of farmers, farming communities) further surveys, analysis.

Reviewer #5

See Attachment

(Remarks to the Author)

Response to review manuscript NCOMMS-24-32226: “Expected Effects of Transforming Agricultural Pest Management across Global Scales”

We would like to thank the anonymous reviewers for providing insightful comments and suggestions for improving and balancing the manuscript, adding important explanations, discussions and examples. Please find detailed point-by-point responses to each in the respective sections below. We wish to thank reviewers for the opportunity to elaborate our arguments and improve the manuscript as a whole.

Reviewer #1

Remarks to authors of reviewer 1

Dear Authors,

I have had the pleasure of reviewing your manuscript titled "Expected Effects of Transforming Agricultural Pest Management across Global Scales" submitted to Nature Communications. Your work provides a comprehensive global assessment based on survey evidence from 517 experts across key disciplines and regions worldwide. The study evaluates the heterogeneity of expected impacts across five domains: economic, human health, food security, social, and environmental.

This study is novel in its holistic approach and global scope, employing rigorous and well-executed statistical methods. This makes me believe that this is an important paper for the scientific community dealing with transforming agricultural pest management systems and mitigating the burden of pesticides on sustainable agriculture, the environment, and social health.

The results suggest that indicators related to human health and environmental quality have higher expected benefits than those related to short-term economic gains and food security. These interesting results emphasize the need for a holistic and multidisciplinary approach to transform them into actionable and tailored regional strategies, especially for policymakers.

Hence, I recommend this manuscript for publication in Nature Communications, but I do so subject to major revisions to enhance readability and focus. As it stands, it is dense, which makes reading it sometimes difficult and unclear. The manuscript would benefit from a more streamlined presentation to enhance readability, which includes reducing redundancy and focusing on the most critical findings and implications. Additionally, simplifying the methodology section by summarizing the key steps and referring to supplementary materials for detailed descriptions is advisable. These general recommendations are elaborated in more detail in the specific comments provided to the authors.

I look forward to seeing the revised version of your manuscript.
Thank you very much.

Response to remarks to authors.

Thank you very much for the compliments, the rigorous assessment and the constructive comments on our manuscript. Following your recommendations, we have now improved the readability: i) we have implemented your detailed suggestion on parts of the text, and further ii) revised and streamlined the rest of the manuscript, following your remarks.

We have further simplified the methodology section, by summarizing key steps in an overview graph, as suggested.

General comments reviewer 1

Comment 1.1

These general comments are valid for all sections of the study. While the English is generally good, the sentences tend to be lengthy with some grammar mistakes and quite a lot of repetition of terms. The text must be simplified and, above all, synthesized, made more direct, and easier to read. For example, breaking down complex and long sentences into shorter ones and avoiding redundant phrases would improve readability.

Response to 1.1

Thanks, we have now revised the complete text, following your detailed comments below, but also improving language and readability throughout the rest of the manuscript.

To support us in improving the readability of the text we have further used the support of an external language editor, as suggested by you.

Comment 1.2

I believe it would be beneficial to have the paper reviewed by someone who was not involved in its writing, as there is a lot of repetition that hinders the text flow. Simplification is possible in most sections, so aim for clear, concise communication by avoiding long sentences.

Response to 1.2

Thanks, we have followed your advice by simplifying the text. To support us in improving the readability of the text we have further used the support of an external language editor, as suggested by you.

Comment 1.3

Additionally, instead of having the last paragraph in the main text discuss limitations, please write a dedicated section for limitations. This will provide a clearer and more structured discussion of the study's constraints and how they might impact the results and conclusions.

Response to 1.3

Thank you, this is a good idea. We have now extended our section on limitations and added an own heading to separate it from the rest of the text (ll. 466-494).

Comment 1.4

At the start (L87/88), the authors question “how policy and industry is how to effectively and efficiently support these policy goals of strongly reducing pesticide use, while minimizing crop losses to pests.” It would complement and strengthen the paper to add a section addressing the policy aspect. This section could specify potential current and future approaches to address this complex issue.

Response to 1.4

Thank you. We have followed your advice and have added to the discussion in the paragraph on policies and added a subheading to highlight its importance (ll. 403-423).

Comment 1.5

The titles of all subsections should be shorter and more punchy to capture attention effectively. As they stand, their importance is lost in the wiggle of words. For example, “Large data gaps on the potential effects of transforming global agricultural pest management - despite global reduction targets” (L72/73) could be: “Data gaps in global pest management transformation despite reduction targets” or something along these lines.

Response to 1.5

Thanks, we have followed your recommendation and revised all titles of subsections to make them shorter and punchier.

Comment 1.6

I have provided a non-exhaustive list of suggestions. I recommend that the authors apply them throughout the text:

- Although the work is about pest management, please avoid overusing this term as it makes the text heavy to read. Use the IPM acronym consistently for Integrated Pest Management.
- The phrase "expected effects" is used repeatedly; consider varying your language.
- Ensure consistency in terminology and punctuation.
- Streamline repetitive phrases describing the survey information.
- Ensure subject-verb agreement in sentences about missing information.
- Avoid using a wide spread of parentheses, e.g., (over), (reducing), etc.
- Remove unnecessary hyphens for better readability.
- Consider using acronyms or varying language to avoid repetition.

Response to 1.6

Thank you for the detailed feedback.

We have implemented most of your suggestions throughout the text.

Regarding acronyms, we agree that acronyms can help to make the text more concise. However, in our case, we are addressing a broad audience which might not be familiar with all abbreviations, and therefore opted to reduce the use of acronyms.

Specific comments Reviewer 1

Note: some responses below refer to several comments simultaneously (as indicated).

Comment 1.7

Abstract

The abstract could benefit from a clearer structure, with a more distinct separation between the background, methods, results, and conclusions. While the abstract mentions "leverage points" and "combinations of actions", specific examples are missing. In addition, some sentences are lengthy and complex, making them difficult to follow. Breaking them down into shorter, clearer sentences would improve readability. For example, in L37-39 could be changed to: "Our study assesses the expected effects across five domains: economic, human health, food security, social, and environmental. We examine these effects in major agricultural production regions worldwide"

L35 (and 99): suggestion to modify to "Here, we provide the first global assessment..."

L38 (and 102): suggestion to modify to "...five domains (economic, human health, food security, social, and environmental) in major agricultural production regions."

L42: I think that instead of "... a global transformation to sustainable pest management ...", the text would flow better with "... transforming agricultural pest management could be an important nexus for addressing multiple sustainability challenge"

L37/39: The phrase "key disciplines and regions worldwide" is somewhat redundant with "main agricultural production regions worldwide" mentioned later. Rephrase one of the two.

L46: The phrase "controlling for important production system- and participant characteristics" could

be clearer if rephrased, as the hyphenation is strange. Actually throughout the text the authors use hyphenation (too) often. Please revise.

Response to 1.7

Thanks, we improved the abstract following your suggestions (ll. 31-50).

Comment 1.8

Main text

L60-61: "Effective management of agricultural pests is thus essential for food security and farmers' livelihoods." This point is repeated in the context of future importance without adding new information. Consider merging or refining for brevity.

L65: "Pesticides often provide affordable and efficient pest management solutions, which can be applied in a wide range of production contexts." The comma before "which" is unnecessary.

L67-68. Change to ""Additionally, heavy pesticide use can diminish long-term agricultural productivity."

L69-71: "For example, due to negative effects on ecosystems services essential for agriculture, such as biocontrol potential (13-15), pollination (16) and soil functions (17), as well as increasing resistances of pests to widely used active substances (18)." This sentence is a fragment and should be restructured to complete the thought.

Response to 1.8

Thanks, we have rephrased the text in line with your recommendations (ll. 53-71).

Comment 1.9

L75:78. Too long, it should be split into shorter and clearer sentences to improve readability

L80: "Not clear, what those "other means" are?."

L81-84: "However, global efforts to reduce and substitute pesticide use remain fragmented, lacking a coherent strategy to mitigate pesticide pollution and adverse effects (26). There is also little evidence of significant progress toward achieving pesticide risk and use reduction targets (e.g., 27, 28)".

Response to 1.9

Thank you for the recommendation. We have adjusted the text according to your recommendations (ll. 83-93).

Comment 1.10

L87-88: This sentence could be more concise for better readability

L88:91. Too long, it should be split into shorter and clearer sentences to improve readability.

Response to 1.10

Thanks, we have reformulated this part: "A central question for research, policy and industry is how to effectively and efficiently support pesticide reduction, while minimizing crop loss due to pests. In order to establish priorities and to design effective and efficient reduction strategies an understanding of the current states, potential effects and trade-offs of a transformation of pest management is required." (ll. 90-93).

Comment 1.11

L107-109: This sentence could be more concise for better readability. For example: "We first identify suitable indicators to assess the effects and trade-offs of reducing pesticide use and transforming global pest management systems." Use this as a guideline for other similar sentences throughout the text to align with the high writing standards of this journal.

L112-116: This sentence could be more concise for better readability.

Response to 1.11

Thank you, we have now rephrased and made the sentences more concise:

- "We first identify suitable indicators to assess the effects and trade-offs of reducing pesticide use and transforming global pest management systems. We selected indicators on the basis of the broad and multidisciplinary literature in the field." (ll. 115-118).
- "Importantly, we also included indicator domains and single indicators in which the literature has previously set clear expectations regarding the direction of effects (e.g., 2, 40, 45). This is to produce a holistic overview and compare expectations from the literature (which are typically based on case studies) to expert opinions at the global level and across indicators, regions and agricultural production systems." (120-124).

Comment 1.12

L116-120: (and other cases) no need to enumerate. "We identified 24 indicators across five domains: environment (ENV),...."

Response to 1.12

Thanks, we have now deleted the enumeration from the text.

Comment 1.13

L121:125. Too long, it should be split into shorter and clearer sentences to improve readability. Both scenarios must be clearly defined.

Response to 1.13

Thanks, we have now improved readability and clearly defined both scenarios: "We then conducted a global online survey (in English, Chinese, Spanish, Portuguese, French and German, covering most global experts) to obtain expert assessments of expected changes in indicators from a baseline scenario to a *sustainable pest management* scenario. The baseline was defined as the current status of pest management in each production system and region, and the target scenario was defined as pest management with no or minimal use of pesticides using combinations of alternative practices in pest management (see Materials and Methods section)." (ll. 129-134).

Comment 1.14

L130: potentials of transforming to "potential for transforming"

L134: "across different indicators" could be simplified to "across indicators.". idem for other cases

L145-148: This sentence could be more concise for better readability

L148-149: where?

L151-153: please simplify to “We lack global estimates and regional heterogeneity data for all 24 key indicators of sustainable pest management (see Table 1).”

Response to 1.14

Thank you for these remarks – we have followed them and improved the text accordingly:

- “We provide a detailed overview of the survey design, validation, and implementation as well as the cleaning and preparation of the survey data in the Materials and Methods section. A detailed overview of the respondent sampling and characteristics is provided in the Supplementary Materials Section A2. The survey data, as well as the complete code for data analysis in R, are freely available.” (ll. 154-158).
- “We lack global estimates and regional heterogeneity data for all 24 key indicators considered in our study (see Table 1).” (ll.160-161).
- We provided the data and codes to replicate results with the review files. Once the paper is accepted, we will further provide a link to the data and codes, which will be stored in an openly accessible repository.

Comment 1.15

L202-204: Please support it with references

Response to 1.15

Thanks, we have now specified that to our knowledge no studies on this topic are available: “Thus far, however, the literature fails to provide quantitative estimates of this variation, to our knowledge.” (ll. 213-214).

Comment 1.16

L2015-2019. Not clear. I do not see the connection with the previous sentence.

Response to 1.16

Thanks, we have now deleted this part since it was already mentioned one paragraph earlier.

Comment 1.17

L307: such as?

Response to 1.17

Thanks, the example follows the first introductory sentence.

Comment 1.18

Material and Methods

The methods described in the Materials and Methods section are generally solid and sound. The study design is comprehensive, data collection procedures are robust, data analysis techniques are appropriate, and validation methods are thorough. However it should be simplified and made clearer, and placing additional details in the supplementary material. The first part, where the surveys are described, should be straightforward and concise. A diagram would help the reader understand the process better. As it stands, it does not flow well.

Response to 1.18

Thank you for the compliment and the constructive remarks. We have followed all of your detailed suggestions (see below) and improved the readability of the methods section. For a better overview, we have further added an overview diagram (Fig. 3), as suggested.

Comment 1.19

Specifically:

L696-697: I am still wondering what is the “other survey and external data”

Response to 1.19

Thanks, this was indeed confusing. We have now specified: “Fifth, we combined survey data with various external data sources.” (ll. 812-813).

Comment 1.20

L710-...: The process of defining the baseline scenario could be described in more detail to understand how the current situations of pest management in different regions were identified and characterized.

Response to 1.20

Thank you for the suggestion. We have now rephrased part of this section to make it clearer: “We defined the baseline scenario as the current situation of pest management in each production system. These current situations can be highly diverse depending on the region. In order to control for this heterogeneity in our analyses, we thus collected key characteristics of production and pest management systems in the survey.” (ll. 836-839).

Comment 1.21

L710-...: avoid using so often the word “elicit*”. Idem for “concepts, approaches, technologies,”

Response to 1.21

Thanks, we followed your recommendation and rephrased throughout.

Comment 1.22

L754-757. The main points are unclear. Could you clarify the primary message and its significance?

Response to 1.22

Thank you, we have now added a statement to clarify and highlight the significance of the statement: “A variety of different measures exists, but their use is very context dependent and currently not widely adopted, despite their recognition.” (ll. 861-862).

Comment 1.23

L771-... The study effectively identifies relevant dimensions for assessing transformations in agricultural production systems by reviewing high-level studies and literature on related topics. However, providing a brief summary of each dimension's relevance and its contribution to the overall assessment could enhance clarity.

Response to 1.23

Thank you for this comment. We have now added explanations on the relevance of each of the dimensions and their contribution to the overall assessment. We now write: “First, the environment, as pesticides are an important source of environmental pollution. Second, human health, as pesticides have been shown to have adverse effects on agricultural and non-agricultural populations. Third, food

and nutrition security, due to the key role of crop protection for food production. Fourth, economic prosperity and livelihoods of farmers, due to the important role of crop protection for economically viable farming, and fifth, social security and equality due to heterogeneity in pesticide exposure and distribution of costs and benefits.” (ll. 896-902).

Comment 1.24

In addition, although the selected indicators (societal and policy relevance, relevance across diverse production systems, representativeness of potential effects, and distinctiveness per category) are well-defined and appropriate. I expected to see example of other indicators that were considered but not selected, and explaining why. This could provide additional context and validation for the chosen indicators.

Response to 1.24

Thanks, we have now more explicitly explained how we come to the set of indicators. More specifically, we write: “We focus on direct effects of changes in pest management, where literature indicates a potentially significant relation to key societal challenges, for example those covered in the United Nations Sustainable Development Goals (see Table S6). We acknowledge that changes in pest management could have even broader, indirect effects, for example if land use decisions of farmers are affected. This is an important topic for future research but was not in the scope of our analysis.” (ll. 936-941).

Comment 1.25

L836-... The creation of the online survey based on the assessment framework is well-executed and methodologically sound. The survey design follows best practices, is structured to gather comprehensive data, and includes thorough validation and pre-testing processes. However, additional details on the development of survey questions, interpretation of responses, and specific feedback from pre-testing would enhance the methodology's clarity.

The survey is divided into five main parts: expert information, perceived importance of pest management, current state of sustainable pest management, potential effects of transformation, and potential tools and policies for transition. Adding a brief rationale for each part and its contribution to the overall research goals would also improve clarity.

Response to 1.25

Thank you, in order to improve clarity for readers, we have now re-arranged this section and added details (ll. 972-1033). More specifically, we now explain the general intuition behind the design of our survey questions, based on literature. We then explain the structure of the survey and reasoning for each question, and we provide guidance on the interpretation of potential critical questions. To make this clear, we now also provide a brief statement on the rationale of the section and its contribution directly underneath each survey part headline.

This is followed by information on pre-testing and implementation.

Comment 1.26

Providing more detail on how the self-elicited expertise scores were validated and used in the analysis would also strengthen this section.

Response to 1.26

Thanks, we have now added further information on how the self-elicited expertise scores were used in the analysis: “This information was needed to exclude respondents from domains where they have no expertise and conduct robustness checks for domain-specific expertise, since our survey covered

different domains and experts with different disciplinary backgrounds. It allowed us to control for potential biases related to different levels of expertise in the regression analyses.” (ll. 980-984).

We further went back to the list of respondents and collected data on institutions, tenure and scientific output. We have now added a new section on the expertise of respondents in the manuscript (ll. 1333-1355).

Comment 1.27

L921-... I commend the authors for their extensive work in selecting the target population. Focusing on scientific experts ensures that respondents are knowledgeable and capable of evaluating current and future scenarios, as well as potential trade-offs. However, including specific criteria for selecting experts and ensuring balanced representation from each region and discipline would be advisable. As well as providing more information on how non-academic experts were selected and how their responses compared to those of academic experts.

Response to 1.27

Thank you, we agree that participant’s characteristics are important to consider in a survey-based approach. We therefore elicit and control for key characteristics of participants in our regression analyses, such as occupation and discipline, scope of expertise, opinions on indicator relevance, and self-elicited domain expertise.

We have now added a new section with additional data on the expertise of respondents (ll. 1333-355)

We further highlight the results of scientific experts compared to non-academic experts: “Furthermore, each indicator received negative responses that ranged from 2% (HH1) to 28% (ECON2) out of all responses per indicator (Fig. S3) and mean responses remain stable across different groups of respondents (e.g., by discipline or sampling strategy and between academic and non-academic experts Fig. S5-S7).” (ll. 224-227).

Comment 1.28

L944... Finally the external data. First point, I think this section should be moved up in the article, as external data is referenced several times earlier without clear direction, leaving the reader uncertain. This section could also be expanded. Providing more information on the sources and reliability of the external data, and any potential limitations or biases. Additionally, clarifying the process for handling cases where aggregated data were unavailable or partially missing, and how this might impact the analysis.

The authors have selected several relevant external variables (e.g., HDI, GDP per capita, potential pest damages, attainable crop yields, agricultural productivity, current pesticide pollution). However, a discussion on the rationale behind selecting these specific variables and their relation to the expected effects of sustainable pest management would improve the understanding of the analysis.

Response to 1.28

Thank you. For the flow of the materials and methods section, we believe that the external data and merging it with survey data has to come after the description of the survey and data collection. But we definitely see your point and have now referenced the external data better throughout the manuscript. We have further extended the section to provide a better overview to readers.

We provide more information on the use of the external data in literature, clarified how we handled cases of unavailable or missing aggregate data and how this might impact analyses. Further, we have now added a paragraph on our choice of external variables (ll. 1085-1112).

Finally, in section 6 of the materials and methods section, where we also discuss the choice of variables and estimation strategies, we have now introduced additional sub-headings to improve the visibility to readers (ll.1117-1212).

Comment 1.29

Data analysis

The data analysis methods described are statistically relevant and sound. The use of descriptive analyses, Kendall rank correlation, and multiple linear regression with appropriate control variables ensures a robust approach to understanding the potential effects of sustainable pest management. The inclusion of multiple hypotheses testing with Bonferroni correction further strengthens the reliability of the results.

Response to 1.29

Thank you!

Reviewer #2

General comments reviewer 2:

Comment 2.1

The important question of whether strong restrictions on synthetic pesticides would improve or worsen pest control along a long list of relevant outcomes has not been asked. Instead, the authors essentially asked if an improved pest control management labelled "Sustainable Pest Management" (defined as not or only marginally including synthetic pesticides, and as being superior to current management along the major dimensions) would improve pest control. The answer they received is that SPM would be better than synthetic pesticides. The question is tautological, the results are trivial.

The assessment framework and drafting reveals in multiple instances ignorance of scientific evidence or basic logic. This includes, for example, misinterpretations of the causal roles of biodiversity and ecosystem services, pesticide residues in food, non-target adverse effects, land sharing vs. sparing, confounding of partial and total/net effects, and more. Some of the cited references do not support what the manuscript claims (checked on a random basis, not exhaustively).

Response to 2.1

Thank you for the intensive examination of our study, which gives us the possibility to further clarify and improve the manuscript. Please find detailed responses to your remarks below.

Note that our definition of Sustainable Pest Management did not exclude the use of pesticides but their use in more sustainable ways than what is observed today, which, in some regions, would imply a clear reduction. We stress that SPM is not simply about the reduction of chemical pesticides but about developing smarter ways of pest control, where chemical means are combined/integrated with other approaches (e.g., biological, genetic, agronomic, mechanical, etc.).

Specific comments reviewer 1

Comment 2.2

36: How reliable are those expert views? How reliable are their impact predictions? It could have been good to include some expert quality checks (asking basic knowledge questions on pest management for which only one correct answer exists).

Response to 2.2

Thank you for your question.

Generally, expert surveys are regularly used in the field to assess important societal questions, where data is limited. They have, for example, previously been used to estimate global yield losses from pests (Savary et al., 2019), obstacles to integrated pest management in developing countries (Parsa et al., 2014), or the state of global plant health (Savary and GPHA, 2023). We are not aware of any similar studies that included a knowledge test and can see several problems doing this. Our expert survey covers a very large sample (substantially larger than recently studies published in adjacent fields in high-ranking journals) of mainly academic experts across several scientific fields (Agronomy & Pest Management Sciences, Ecotoxicology, Ecology, Bio-technology, Agricultural Economics & Sociology) and across five continents, covering 69 countries, key hotspots of global food production and representing a combined share of 88%, 85% and 90% of global cereal, fruit and vegetable production.

We have now additionally collected information on respondents' scientific expertise and independence. We find that the large majority of respondents are employed at independent research

institutions, and are experienced, tenured researchers with a high scientific output in the field (ll. 1333-1354).

We adopt a broad and diverse approach to expert quality checks in our study. More specifically, we include three types of expert quality checks:

First, in selecting experts from academia, who are knowledgeable and have published on the topic (see above for an overview of academic experience). Second by selecting experts from a wide range of disciplines and specifically asking experts about their expertise in each field of questions, and third by explicitly accounting for expert heterogeneity in our results. More specifically, we check robustness of results regarding discipline, scope of expertise, and expertise in the field of question. For example, following your comments, we now show that dropping certain groups of experts, disciplines, or weighting results by expertise does not change the main results (ll. 1327-1332). We further explicitly control for the different characteristics of respondents in our regression analysis.

Comment 2.3

46: Aren't higher yields better for the environment? Isn't higher production and protection from pests better for human health? The framing seems to be biased on the benefits side- the expectation for ambitious restrictions on widespread pesticide use would be to impose a cost on humanity, not provide benefits. And it misses the important point that in many developed countries, pesticide use does not pose relevant health risks because of food safety and occupational health standards.

Response to 2.3

Thank you. We specifically agree with important parts of your comment.

Pesticides are important to protect yields and contribute to the production of sufficient quantities for food security. These effects also differ regionally. We agree that producing the same quantities without pesticides could have other negative environmental effects (more land required) if no other pest control options exist. Further, we do not define SPM as a scenario that strictly operates without pesticides but that uses and combines various pest control strategies (including some that need further research) - the availability and efficiency of substitutes will again differ by crop and region.

There is broad evidence that pesticides have potential negative effects for human health and the environment (see the responses to your comments below). These effects are more relevant in some regions than in others due to various factors (e.g., different regulations/standards), which is why we consider and compare different regions in our study.

We account for such potential tradeoffs and heterogeneity in our analysis, for example by comparing results across crop types and regions. We have now clarified this throughout the manuscript.

Comment 2.4

47: Be more concrete, what is this different pathway?

Response to 2.4

Thank you, we have now replaced “pathway” by “strategy” and hope this better conveys our intended message.

Comment 2.5

50: "delivering"- developing? researching? innovating?

Response to 2.5

Thanks, we have now replaced “delivering” by “developing”.

Comment 2.6

51: "political boundary conditions"- Unclear what is meant. Subsidies? But aren't subsidies costly and prone to mismanagement?

Response to 2.6

Thank you, we agree that this can include a broad range of measures, but since space is limited in the abstract, we refer to the sections six and seven of the manuscript (ll. 389 - 465) for more details.

Political boundary conditions may include subsidies as you mentioned, but also information measures regulations and frameworks, for example on the authorization of pesticides or legislation on biotechnology (gene editing), which is closely related to crop protection (see Möhring et al., 2020 for an overview).

Comment 2.7

56: Weeds should be explicitly mentioned here to avoid the potential misunderstanding that it's only about animal pests and diseases. According to your first cited source, potential yield losses are much higher from weeds than from animal pests and pathogens (despite weeds having a wider range of effective management options).

Response to 2.7

Thank you, we now explicitly state "insect pests, plant diseases, and weeds".

Comment 2.8

57: It would be important to make clear here whether you are considering pre-harvest losses only or also post-harvest (post-harvest losses are particularly important in developing countries, and can be alleviated with higher pesticide use to the extent that storage pests matter).

Response to 2.8

Thanks, we state this later on in the manuscript, but see your point that this matters from the beginning. We have now added "[...] with a focus on pre-harvest losses" (l. 109) already in the introduction, where we introduce the research design.

Comment 2.9

58: These are quantity losses, for quality losses, any estimates?

Response to 2.9

Thanks, this is an important point. To our knowledge there is no global assessment of quality losses due to insect pests, plant diseases, and weeds, but only case study evidence (e.g., Savary et al., 2012, Zakowski and Mace, 2022).

We have now added: "Further, quality losses can be important (3), but no global assessments exist." (ll. 58-59).

Comment 2.10

59: Where does the 29% come from? The abstract of the cited paper mentions 26% as the lower bound (for soybeans)

Response to 2.10

Thank you for your remark. We took the upper bound estimation for each crop to summarize results. We agree that this does not fit with how the sentence is phrased and have changed "29%" to "26%".

Comment 2.11

61: Higher yields, to which pesticides contribute, are also essential to keep the agricultural area smaller than it otherwise would be (<https://ourworldindata.org/yields-vs-land-use-how-has-the-world-produced-enough-food-for-a-growing-population>)- with positive impacts on the size of the natural environment and other potential uses of land. If you like biodiversity, it should also be mentioned as a potential benefit of pesticide use through the yield-land-sparing mechanism (as natural areas tend to have higher biodiversity than agricultural areas).

Response to 2.11

Thanks, we acknowledge the role of pesticides as effective, cheap and easy to apply solutions for crop protection in our manuscript (also see lines 65-66).

To emphasize the points mentioned by you, we have now added “Effective management of agricultural pests is thus essential for farmers’ livelihoods and for producing sufficient and diverse food for a growing population on a limited amount of land.” (ll. 59-60)

Please also see our response to comment 2.3.

Comment 2.12

67: The cited evidence is too weak to support the statement "pesticides adversely affect ecosystem functioning [and] biodiversity". Generally, evidence is missing to support those statements because the accounting of the species potentially affected (positively or negatively) by pesticide use is hopelessly incomplete- citing adverse effects on an extremely small share of those species does not constitute evidence that biodiversity is "adversely affected". More importantly, biodiversity has no causal relationship with ecosystem functioning or ecosystem services (see Maier, 2023: Chasing Biodiversity Off the Scientific and Conservation Tracks. doi.org/10.3998/ptpbio.4337; Schoolmaster Jr., et al, 2020: A graphical causal model for resolving species identity effects and biodiversity-ecosystem function correlations. doi.org/10.1002/ecy.3070). The statement in the manuscript linking pesticide use, ecosystem functioning and biodiversity is therefore unsupported, highly misleading and should be removed.

To the extent that pesticides increase effective yield, they can also positively contribute to the sparing of natural areas from agricultural land expansion.

Response to 2.12

We have modified the sentence to "pesticides **can** adversely affect biodiversity and ecosystem functioning" (ll. 66-67).

Sure, we do not disagree that evidence is missing; only a small fraction of living species is thought to be described (Costello et al. 2013). However, existing evidence illustrates that bats (e.g., Frank et al. 2024), amphibians (e.g., Baker et al. 2013), birds (e.g., Li et al. 2020), and invertebrates (e.g., Gandara et al., 2024, Sánchez-Bayo, 2014) are harmed by pesticides directly and indirectly through changes in traits such as diet, disease susceptibility, and growth. Do we know that every species will be negatively affected, of course not. Will it depend on the pesticides used, and the environmental conditions, sure. Will all studies uniformly find a negative effect, no. However, that does not negate the large body of evidence suggesting that pesticides have a negative effect.

The land-sparing effect of pest control (higher effective yields) and its important role for natural biodiversity is well acknowledged, but we stress that our SPM scenario is not about zero pest control, but about more sustainable forms of pest control.

Regarding biodiversity ecosystem function (BEF) - it was not our intention to rehash the 50-year-old discussion of BEF relationships. This sentence was noting that pesticides [can] adversely affect ecosystem function (as well as biodiversity, as well as human health). Evidence suggests many

ecosystem functions such as pest control (Frank et al. 2024, Gross and Rosenheim, 2011), pollination (Stanley et al., 2015) and nutrient and carbon cycling (eg Rumschlag et al. 2021, Sim et al. 2022) are negatively impacted by pesticides. We were not arguing whether it is due to "biodiversity" or due to functional trait identity. We agree, BEF is a topic worth the serious debate it has received in the past decades, but it is also a debate outside of the scope of our analysis.

Comment 2.13

67: The first cited source (9) refers to human health outcomes. However, the source states: "studies with more refined measures of agricultural pesticide exposure and/or other outcomes have generally reported null or inconsistent effects of exposure on birth defects, low birth weight, and gestational length", citing a handful of studies. The results from the study (9) also provide only limited evidence of a general harmful effect on actual pesticide use (statistically significant effect only for top 1% of exposure quantities, i.e. extremely high and very rare applied quantities)- the cited evidence thus does not support the broad-brush statement in the manuscript that "pesticides adversely affect human health", this needs to be much qualified or "human health" must be removed. The statement in the manuscript is also providing undue fuel to the widespread misconception that pesticide residues in food represent significant health risks (they do not).

Response to 2.13

Thank you. We have now qualified the statement to "[...] pesticides can have adverse health effects, especially on farmers and bystanders (12, 20-22) [...]" (ll. 69-71).

Analogous to our response to comment 2.12, we maintain that human health effects of pesticides are broadly established in the literature, although they are of course highly dependent on the context. We agree that the health risks of pesticide residues in food are often overrated in rich-world countries, where strict standards apply. This is different in many low- and middle-income country settings. Beyond the food pathway, other mechanisms of exposure exist.

To support our statement on links between pesticides and human health, we now additionally refer to three recent studies that show causal links, using data on observed health outcomes and pesticide use (in real-world settings) (Jones et al., 2020, Frank et al., 2024, Fletcher and Noghanibehambari, 2024).

For a more detailed discussion on the health effects of pesticides and some of the underlying physiological mechanisms we refer to, for example, Landrigan et al., (2018), INSERM (2022), Karalexi et al. (2021) and Giulioni et al., (2022).

Responding to your concern about effects on consumers, we now additionally add a clarifying sentence in the part of the manuscript where health effects are discussed in detail: "Pesticide residues in food are found to be mostly below maximum residue levels in regions with strict standards (e.g., EU countries), but can be important concerns in countries with laxer standards and controls (e.g., Handford et al., 2015, Donkor et al., 2016)." (ll. 196-198) (also see the response to comment 2.31).

Comment 2.14

68: This is a highly misleading sentence that should be removed or substantially qualified. Biocontrol can act as a complement or substitute to chemical and mechanical pest control. While it may be true that pesticide use potentially reduces biocontrol (e.g. by reducing populations of non-target species that are helpful in pest control), it does not follow that overall agricultural production potentials are reduced. If pesticide use improves pest control through its direct effects on target-organisms more than it worsens pest control through its indirect effects on non-target organisms, then the net impact of pesticide use on pest control is still positive.

Similar considerations apply to soil functions. The cited reduction in P uptake as a result of fungicide applications could also be balanced with studies showing that pesticide use increases water or N use efficiency.

Response to 2.14

Thank you, we have now revised the sentence and write: “This includes ecosystem services that are important for agricultural production, such as natural pest control (13-15), insect-pollination (16-18) and soil functions (19).” (ll. 67-69).

Also see the response to comment 2.12.

Comment 2.15

The threat to agricultural production from pesticide use reducing pollination is much overstated here in light of the available evidence and facts. The cited source (16) itself states: "The few available true field studies assessing the effects of field-realistic exposure on pollinators provide conflicting evidence of their effect" and "Substantial gaps in knowledge remain regarding the effects of pesticides on pollinators". Importantly, and similar to the biocontrol case, even if pesticides did meaningfully reduce pollination, their net impact on agricultural production might still be positive, namely if the pest control afforded by them brings greater yield gains than the losses through reduced pollination. Some other relevant facts: Sixty percent of global agricultural production comes from crops that do not depend on animal pollination (doi.org/10.1098/rspb.2006.3721). For crops depending on animal pollination, far less than 100% of the yield is due to animal pollination, e.g. 25% of the soybean and rapeseed yield is due to pollination, for pulses and tomatoes only 5% ([doi:10.2760/619793](https://doi.org/10.2760/619793)). Furthermore, the domesticated honeybee provides primary pollination services to crops highly dependent on animal pollination (e.g. melons, pumpkins, some exotic fruits/nuts), and global honeybee numbers are increasing (doi.org/10.1007/s13592-020-00788-9). The domesticated honeybee is a managed species to whom pesticide use poses no outsized threat. Most wild bee species, which are less amenable to direct management, make no significant contribution to crop production (doi.org/10.1038/ncomms8414).

Response to 2.15

Thank you. We have adjusted our statement (see response to comment 2.12). We now state in this section that i) pollination is important for agricultural production and ii) that it [can] be negatively affected by pesticides. Both points are firmly established in the scientific literature (see below).

In order to account for the diverse discussion on the topic you have pointed out, we now also reference two of the studies indicated by you in the manuscript (Klein et al., 2007, Wood et al., 2020).

To point i.): Pollination is important for agriculture. It is true that important parts of human calory intake do not rely on pollination for food production. However, this does not mean that pollination is not important for the production of the diverse food needed for a healthy human diet. For example, the first paper cited by you states that “Thinking beyond caloric intake, however, our results support the opinion of Steffan-Dewenter *et al.* (2005) that our diet would be greatly impoverished, both nutritionally and culturally, if pollination services further decline.” (Klein et al., 2007, p.307). Further, pollination is of economic importance for agriculture. For example, the third paper cited by you states: “On average, wild bee communities contributed \$3,251 ha⁻¹ to production of the examined crops (s.e.= \$547, range \$7–14,252), about the same as the contribution of managed honey bees (mean±s.e.= \$2,913±574, range \$0–18,679).” (Kleijn et al., 2019, p.2).

To point ii): There is broad scientific evidence of negative effects of pesticides on pollinators (e.g., Goulson et al., 2015, Sánchez-Bayo et al., 2016, Nicholson et al., 2023, Basu et al., 2024 for an overview). The studies referenced by you state that further research is needed to understand the effects of pesticides on bees and their heterogeneity across pesticides, species, management types, and landscapes and they derive different conclusions for the management of pollination and the conservation of bees. We welcome this differentiated discussion and therefore include two of the

references mentioned by you - but a broader discussion on the topic is not in the scope of our article (also see response to comment 2.12).

Comment 2.16

74: The policy targets are not necessarily rational, you have to consider decades of environmentalist campaigning, rather than (scientifically validated) "issues".

Response to 2.16

Thanks, as it stands, reduction targets are adopted on global levels, for important regions and on national levels, which is indicative of a societal consensus that has been reached in these democratic systems. We here do not judge how governments reached consensus on these decisions, but rather ask the important question of what potential consequences could be - and how these targets could best be reached while considering a wide range of indicators, including production and economic viability of farming.

Comment 2.17

77: Please note that in May 2024 the Commission withdrew the proposal.

(<https://www.europarl.europa.eu/legislative-train/spotlight-JD22/file-sustainable-use-of-pesticides-%E2%80%93-revision-of-the-eu-rules>)

Response to 2.17

Thanks, this is an important point. It shows that data and scientific evidence on the topic are required to support policies that consider pesticide risks, production needs, and economic viability simultaneously. Our survey is an important step for future research on this topic, since large-scale data are missing. Following your comment, we now emphasized this point in the discussion.

It is further worth mentioning that From-Farm-to-Fork still stands as a strategic goal (as part of the EU Green Deal) and that Regulation 1107/2009 has also been reviewed and will stand as a matter of fact. It will have a significant impact on crop protection in the EU in the future due the expected reduction in the number of a.i. 's authorised (e.g., the future of all candidates for substitution is very uncertain).

Finally, the EU countries are also members of the Convention for Biological Biodiversity, which has set similar reduction targets on global levels (Convention for Biological Biodiversity, 2022). Also here, (policy) support and implementation have not been assessed yet – for which our study can be an important starting point.

Comment 2.18

78: More precisely, since when and in which world regions and how intense have these promotion efforts been?

Response to 2.18

Thanks, we collected some indications regarding your question below:

- The history of Integrated Pest Management can be traced back to the late 1800s (Kogan, 1998).
- In modern times, the research institute for organic farming was founded in 1973 (<https://www.fibl.org/en/about-us>), which aims to promote organic farming globally.
- Integrated Pest Management was promoted starting from the 1960s globally in various regions (see Parsa et al., 2014 for an overview), and for example since the 1970s in the USA on a federal level (<https://www.usda.gov/sites/default/files/documents/20201110-FIPMCC-History.pdf>).

We have now additionally referenced Kogan (1998) in the manuscript to give readers an overview on the topic.

Comment 2.19

83: This seems counterintuitive. Haven't older more toxic pesticides have been removed from the market over the last decades, at least in many developed countries, due to more stringent approval standards (e.g. in the EU)? And isn't the lack of detailed pesticide use and risk data (e.g. in many European countries) not the main point here, rather than the conclusion that the available data shows no reductions in overall risk? What is the best data source underlying the statement that no sizeable achievements in overall pesticide risk have been made in Europe or North America, for example? And I miss the point that it's less the applied quantity but rather the combined unmanaged quantity-toxicity ("overall risk") what matters - although policy might indeed mis-focus on quantities.

Response to 2.19

Thanks for the questions – we here respond to comments 2.19 and 2.20.

We agree with your remark that our statement is not true for all countries and regions and have now adjusted it “[...], with exceptions such as Denmark (Nielsen et al., 2023).” [ll. 86-87].

We have now clarified in the manuscript that “Both initiatives set reduction targets until 2030 and emphasize the reduction of environmental and health risks from pesticide use (see 24, 26-27 for extensive discussions on this question).” (ll. 77-79)

We indeed lack good data on pesticide use and risks over time and space (e.g., Mesnage et al., 2021). However, case study evidence for several important agricultural regions (in terms of pesticide use and production) shows high levels or even increasing trends in toxicity and risks of used pesticides (see ll. 87-89).

Thank you for mentioning the very nice comparison of several indicators by the JKI. Looking at their comparison, we cannot observe a clear downwards trend (<https://sf.julius-kuehn.de/pesticide-dbx/synops>), nor is the slight reduction in the last year of the sample close to fulfilling the intended reduction targets.

The Harmonized Risk Indicator of the European Union has intentionally not been mentioned by us, since it can be misleading. It has for example been strongly criticized for actually not reflecting risk reductions or policy effects by the European Court of Auditors (see European Court of Auditors, 2020). More specifically, because of the classification of a.i. 's into four groups depending on their legislative status, the HRII indicator reflects changes in the legislative status of pesticides in the EU and not necessarily the adoption of sustainable crop protection methods.

Comment 2.20

86: The harmonised risk indicator 1 published by the European Commission shows a substantial decrease during 2011-2021 for the EU (https://food.ec.europa.eu/plants/pesticides/sustainable-use-pesticides/harmonised-risk-indicators/trends-eu_en). For Germany, SYNOPSIS shows a downward trend in risk. The available data might be insufficient for strong conclusions, but they point towards risk reductions rather than no reductions. So a bit more nuance would be necessary here.

Response to 2.20

Please see the response to comment 2.19.

Comment 2.21

92: There are several studies assessing the impact of the F2F strategy (https://agriculture.ec.europa.eu/document/download/69adfd09-3836-4a9e-b673-4435a56ec4cd_en), and also studies on organic farming expansion (e.g. doi.org/10.1038/s41467-019-12622-7)

Response to 2.21

Thank you for your comment. We know these studies on F2F impacts, but have specifically not mentioned them, since they are not peer-reviewed. Reflecting the importance of this cautious approach, Wageningen University has recently put out a statement that there were “apparent conflicts of interest” taking place in the creation of their study [one of the studies indicated in your reference] (<https://www.wur.nl/en/newsarticle/dialogue-about-the-pitfalls-of-contract-research.htm>). Further, some of the mentioned studies are only based on estimates of a small number of experts or results from a limited number of field experiments under controlled conditions, not accounting for potential improvements in efficiency of crop health management, behavior of farmers, or the availability of novel substitutes or means to improve pesticide use efficiency. Finally, they have been criticized for their narrow scope, for example not considering potential environmental spillover effects of pesticide reduction on agricultural production (Schneider et al., 2023).

We have now added an explicit statement reflecting these points: “Some studies exist on a European level (see 43 for an overview). However, they are not peer-reviewed. Further, some of these studies have been criticized because they are based on estimates of a small number of experts or results from a limited number of field experiments under controlled conditions and have narrow scope. They, for example, do not consider potential environmental spillover effects of pesticide reduction on agricultural production, do not account for potential improvements in efficiency of crop health management, farmer’s decision-making, the availability of novel substitutes or means to improve pesticide use efficiency (e.g., 43).” (ll. 96-103)

The study you have mentioned on organic farming by Smith et al. (2019) is a good example for the literature gap we are aiming to bridge with this study. Their study focuses on one region (England and Wales) and GHG impacts of conversion to organic farming (including global spillover projections). For our context, studies focusing on a certain pest, pesticide, crop protection practice, indicator or region exist and are important to understand local conditions and underlying processes in detail, as we acknowledge in the manuscript. We here simultaneously account for all global regions, a broad range of potential practices and production systems, as well as a holistic set of sustainability indicators, which is novel, to our knowledge.

Comment 2.22

102: You missed climate change mitigation and adaptation, an important omission.

Response to 2.22

Thanks, in our manuscript we assess direct expected effects from changing crop protection practices. Since this was not sufficiently clear in the previous version, we have now clarified this throughout the manuscript:

- “We compare heterogeneity and assess the drivers of expected direct effects of a transformation of pest management [...]” (ll. 109-110)
- “We focused on indicators, which are largely related to direct effects, for example, not considering effects from potential land use changes, e.g., on GHG emissions.” (ll.910-911).

Although we note the clear relation of climate change leading to changes in pest pressure (in the first paragraph of the manuscript), we are not aware of a large literature of the effects of (changing) crop protection practices on climate change. A recent study outlines that pesticides only make up for around 3.1% of global cropland GHG emissions (Wyckhuys et al., 2022).

We acknowledge that yield drops could and likely would lead to additional land-use change with important climate effects, but our results suggest that more sustainable pest control may be possible without large yield drops and additional land requirements, for example through integrated pest management. We further acknowledged in the limitations that a global analysis of land-use change scenarios is out of the scope of our analysis (also see response to comment 2.27).

Nonetheless, we have additionally added in our manuscript that: “Future assessments should take potential changes in land use and resulting direct and indirect effects, for example on greenhouse gas emissions into account.” (ll. 487-489).

We have further added in the Materials and Methods section: “We focus on direct effects of changes in pest management, where literature indicates a potentially significant relation to key societal challenges, for example those covered in the United Nations Sustainable Development Goals (see Table S6). We acknowledge that changes in pest management could have even broader, indirect effects, for example if land use decisions of farmers are affected. This is an important topic for future research but was not in the scope of our analysis.” (ll. 936-941).

Comment 2.23

117-118: Several overlaps exist among these domains.

Response to 2.23

Thanks, this is correct and intended in our approach.

We explain this in detail in the materials and methods section: “Note that to create a holistic overview of potential effects and trade-offs across all indicator domains, we also included indicators that highlight a similar challenge from a different, domain-specific angle. For example, HH3 (health effect on consumers) and FS3 (safe food), as well as HH4 (healthy food choices) and FS2 (healthy diets). We believe that this approach does not affect our analysis of results, as we i) interpret results by indicator category, and ii) use multiple, separate regression analysis, and iii) check for multiple hypothesis testing in our analysis.” (ll. 931-936).

Comment 2.24

123: "sustainable"- This is inappropriate framing, as pest management without synthetic pesticide use might not be sustainable, at least in the short-medium term. The term, which is found across the key survey questions, is loaded and might bias answers towards favouring "sustainable PM". The definition provided in Section C of the questionnaire is not neutral ("sustainable PM cornerstone of sustainable agriculture", "meet society's current and future needs", "overreliance on synthetic pesticides"). This is a huge red flag.

The question might be ill-defined and misunderstood, casting doubts on the interpretation of the answers. Is this about restrictions on pesticide use? Or about the replacement with alternatives that are stipulated to be equally effective (then the answers are trivial)? What is the time-frame for the impact to materialise?

A more neutral and precise definition and question would have included "policy-imposed restrictions on the use of synthetic pesticides" and their extent (50%? 90%?) (clearly stating whether and what accompanying measures are implemented) and its impact at a specified point in time (2030? 2050?).

Response to 2.24

General response:

Thanks, the main goal of the study is to support the strategic decision-making of policymakers and food-value chain actors by providing data on potential outcomes of a transformation of pest management, following goals for pesticide reduction that have been set on global and regional levels.

The framing and definition of the scenarios in the survey are therefore, in content and language, closely aligned to the policy goals, notably the From-Farm-to-Fork strategy and the Post-2020 Kunming-Montreal Biodiversity Framework (European Union, 2020; Convention for Biological Diversity, 2022). We explain how we chose these definitions in detail in the materials and methods section (ll. 822-879).

Detailed responses:

Out of these explanations we would like to highlight two important points in response to your comment:

i) Thank you for your question regarding substitutes. We have chosen a definition that is closely aligned with the principles of integrated pest management (e.g., Barzman, 2021), which is widely known: “may contain a minimal use of synthetic pesticides but mainly consist of single alternatives or a mix of alternatives: Such as biological solutions (e.g., biocontrol, bio-pesticides), agronomic solutions (e.g., adapted crop rotations, field hygiene) technical solutions (e.g., mechanical weed control, smart farming), breeding solutions (e.g., resistant and adapted varieties) and system redesign (e.g., systems that favor natural solutions for pest control).” This also aligns with recent definitions of Sustainable Pest Management in literature (Finger et al., 2024). Additionally, we conducted several robustness checks of the survey design *ex-ante* and *ex-post* of the survey execution (Materials and Methods, ll. 947-957). In pre-tests of the survey and post-survey interviews, respondents confirmed that their understanding aligned with the above-described, intended definition. Further, our main results aligned with expectations from literature.

Global data on the availability and performance of substitutes for pesticides is currently not available. We therefore asked experts to estimate its current level in their expertise region and controlled for it, alongside a range of other important socio-economic and environmental variables in our regression analyses. We further included “the availability of substitutes” when asking respondents to rate the importance of different measures for achieving a full transformation to sustainable pest management. It received the highest score out of all alternatives.

ii) Thank you for your question regarding the definition: We explicitly used a subjective definition of pesticide reduction (“no or minimal use of pesticides” and “substantial reduction”), and not a fixed, numeric goal (XY% reduction), as regional reduction goals and especially reference scenarios might differ across production systems and global regions (see ll.843-847). Using a fixed percentage reduction scenario with different reference scenarios would thus have led to less rather than more comparability of results.

We therefore opted for a more absolute definition of a reference scenario and at the same time controlled for current implementation levels in the statistical analyses. We further formulated a goal that closely sticks to the paradigm of integrated pest management (which is widely known) and reflects the spirit of the policy targets (Integrated Pest Management is for example a part of the Sustainable Use Regulation of Pesticides of the European Union, Directive 2009/128/EC) (ll. 735-739).

Exactly because perfect substitutes for chemical pesticides may not (yet) exist for all situations, there are important tradeoffs, which we evaluate through our expert survey to the extent possible.

Comment 2.25

129: Best might still not be reliable enough, depending on the extent to which respondents' replies are accurate.

Response to 2.25

Please see the response to Comment 2.2.

Comment 2.26

133: "Direction"- Effect size is of great importance too, why did you not ask about it?

Response to 2.26

We asked respondents about responses on a scale from -10 to +10. However, we believe that other methodological approaches (which are currently not available in the domain, due to methodological and data gaps) would be needed to capture exact effect sizes for different indicators and regions (ll. 489-494).

In designing our study, we therefore cautiously restrained the interpretation of our results to the direction of effects, differences in levels between indicators and regions, as well as potential drivers (using regression analyses). We also state this explicitly in the limitations section of our manuscript: "Given the broad geographic and interdisciplinary scope of our work, our results are not intended to quantify absolute effect sizes for given indicators and regions and can therefore only be a starting point for identifying focal points for more regional- and local- in-depth analysis." (ll. 483-485).

Comment 2.27

133: "at global level"- How are trade interactions accounted for? If one region goes ahead with unsustainable pesticide reductions, others may pick up the slack.

Response to 2.27

Thank you. This is an important point, which we also highlight in the introduction of the manuscript and discuss in the conclusion section (ll. 487-494).

Accounting for such trade effects is not in the scope of our analysis. It would require economy- or sector-wide models that cover crop protection in detail, which currently do not exist - but for which our survey could provide first entry points and data.

We had previously already highlighted this limitation and the importance of modeling for assessing future scenarios for crop protection, but now make this more explicit in response to your comment: "Further, future research should complement our results from expert assessments with results from statistical estimates based on observed data or simulation models, for example on changes of farmer decision-making, global prices and land use decisions. These have been successfully combined in adjacent fields, e.g., for impacts of climate change (79). But currently suitable methods are missing in the field of sustainable pest management and their implementation across scales would be hampered by limited data availability." (ll. 489-494).

Comment 2.28

141: Some more detailed information on the respondent characteristics (education levels, disciplines, institutions) would be useful here for the reader. Are these mostly leading scientists in relevant fields, professors at the best global universities?

Response to 2.28

Thanks, following your comment, we went back to the list of participants and collected data on tenure, scientific experience and output, and potential conflicts of interest and funding sources.

We have anonymized the data and now added descriptive statistics on this information in the manuscript.

Our data shows that the large majority of respondents are employed at independent research institutions, and are experienced, tenured researchers with a high scientific output in the field (ll. 1333-1354).

Comment 2.29

160: But is it in practice? What exactly is the policy scenario here?

Response to 2.29

We closely stick to the definition of Integrated Pest Management here, which emphasizes the use of substitutes.

Please see also the response to comment 2.24.

Comment 2.30

167 "Domain 1"- Where is climate change? Through supporting higher yields, pesticides can reduce agricultural land expansion/deforestation and GHG emissions.

Response to 2.30

Thank you for your remark. We focus on direct effects of pest management in our assessment. We have now clarified this throughout the manuscript.

Please also see the response to comment 2.22.

Comment 2.31

167 "Domain 3"- The positive results of the expected direction of the global effect cast substantial doubt on the survey design (question framing) and/or the reliability of the expressed expert views. The phrasing of the questions in this section are highly loaded in combination with the provided definition: the definition says that SPM is a cornerstone and meets current and future needs in terms of food and nutrition security, and the question then asks whether SPM is more or less suited to provide food in sufficient quantities etc. I do not believe that the design of this survey passed appropriate quality control before it was launched. Any textbook on survey design would advise against it

167: "FS3, +++"- This result makes very clear that the expressed views are not reliable indicators of the actual impact. According to the scientific evidence, actual levels of pesticide residues pose no significant threat to food safety. So the belief that a replacement of synthetic pesticide use with other pest control measures would substantially (+++, in relation to other expected effects) improve feed and food safety is simply not aligned with the science.

The first cited key reference (51) does not provide evidence on pesticide residues as a significant health risk, it cites another study (of dubious quality) which seems to refer to poisonings not from food residues but rather direct exposure. I strongly encourage the authors to omit this reference.

The other cited key reference (52) is not focused on pesticides but on chemicals more broadly. Of 470 (!) pesticides tested, only one (chlorpyrifos in smoked fish, unauthorised) could pose a human health risk. This reference is also clearly insufficient to serve as a model for the "expected effect" of a pesticide restriction policy.

General response to comments 2.31-2.36

This general response applies to comments 2.31-2.36. Please further find detailed responses below the respective comments.

Thanks, regarding your remarks on the design of our study, we would like to emphasize that we have conducted several validations of the study design and the results *ex-ante* and *ex-post* of the survey execution, which we now describe in detail in the Materials and Methods section (ll. 947-957, please also see responses to comments 2.2 and 2.24).

We would like to specifically highlight that for the majority of the 24 indicators, results were in line with expectations from literature (see ll. 186-227 for an extensive discussion). We further show that

even for the few indicators where results did not align with general expectations from literature, literature provides potential indications and explanations.

Importantly, for all of these indicators we find strong heterogeneity in responses over production systems and regions – and the direction of heterogeneous responses again aligns with expectations from literature (ll. 229-252). We further find that correlations between indicator responses and country-level indicators of human development and progress on related SDGs, again, consistently align with expectations from literature (ll. 253-269).

Finally, we assess a broad range of potential socio-economic and environmental drivers and again find that results align with expectations from literature on the direction of relations (ll. 327-388).

To complement these analyses, we have now added an additional analysis, comparing the heterogeneity of responses across indicators and regions (see Supplementary Materials, Section A3). This analysis shows that especially for the most productive regions (Europe, North America and Oceania) and the domains of Economics and Food Security respondents have diverging opinions on expected effects. This highlights the need to clearly communicate this heterogeneity of opinions (we do so at several instances) and investigate potential drivers (which we do in the regression analyses). We now report our findings in the manuscript (see Supplementary Materials, Section A3) and refer to them in the main text: “Finally, out of all indicators and regions, respondents’ expectations were most heterogeneous for Europe, North America and Oceania and the domains of “Economics and Farmer Livelihood” and “Food and Nutrition Security” (see Supplementary Materials, Section A3). This indicates that experts had diverging opinions on expected effects especially for these indicators and regions. It highlights the importance of looking into potential drivers of expected effects.” (ll. 270-274).

Importantly, defining a production system always remains subjective to some degree (especially when no official definition exists) and choosing a different definition would (and should) affect responses (see response to comment 2.24 for a detailed discussion). This is an inherent limitation of expert assessments compared to statistical approaches or simulation models (which do not exist in this field on global levels). We address this limitation of expert assessments by using a large and diverse sample and a broad range of different control variables in the empirical analyses. We now explicitly acknowledge and describe this in the limitations section of the manuscript: “An inherent limitation of expert assessments is that results are dependent on the chosen definitions in the survey. We here addressed this by an external and internal validation of our study design before its execution, by choosing a large and diverse sample (different geographic regions, professions and disciplines), controlling for important respondent characteristics in our regression analysis, and a general focus on heterogeneity and drivers instead of levels of indicators (see Fig. 3 for an overview).” (ll. 467-472).

We have further carefully accounted for this limitation of expert assessment in the study design. We do not interpret results on the absolute levels of indicators, but mainly focus on the heterogeneity across indicators, regions and potential drivers in our analysis. We, for example, state in the manuscript: “Given the broad geographic and interdisciplinary scope of our work, our results are not intended to quantify absolute effect sizes for given indicators and regions and can therefore only be a starting point for identifying focal points for more regional- and local- in-depth analysis.” (ll. 483-485, also see the response to comment 2.26).

Given the scope of our research goals and the above-mentioned broad empirical evidence on results of analyses aligning with literature, we are therefore confident in our study design.

In response to your comment, we have now also improved readability and organization of the Materials and Methods section and provided an overview graph (Fig. 3). We have further better referenced the Materials and Methods in the main body of the text to provide readers a better overview of the methodological approaches used and the robustness checks performed.

We have further substantially extended our section on the limitations of the study and expert assessments in general (ll. 467-494).

Finally, we have checked all references in Table 1, deleted those indicated by you and added new references for several indicators. Please find detailed responses to your comments, addressing these changes, below.

Detailed response to 2.31

Thanks for your remark. We acknowledge that the indicated references were not sufficient and follow your recommendation of removing them. We have now replaced them by Handford et al., 2015, Donkor et al., 2016 and Syed et al., 2014.

Please note that although assessments by large food safety organizations, for example from the EU and USA, regularly show a low number of food samples violating MRLs, global regulations are not consistent (e.g., Verger and Boobis, 2013, Handford et al., 2015) and there is broad evidence in the literature that contamination with pesticides can lead to food safety issues, especially in countries of the Global South (Donkor et al., 2016, Syed et al., 2014, Kazar Soydan et al., 2021, Lozowicka et al., 2015 Gerage et al., 2013, Ssemugabo et al., 2022, Philippe et al., 2021). Additionally, in these countries data and assessments are often scarcer, and enforcement is challenging (e.g., Handford et al., 2015, Donkor et al., 2016). We now explicitly acknowledge this global heterogeneity in the manuscript: "Pesticide residues in food are found to be mostly below maximum residue levels in regions with strict standards (e.g., EU countries), but can be important concerns in countries with laxer standards and controls (e.g., Handford et al., 2015, Donkor et al., 2016)." (ll. 196-198) (also see response to comment 2.13).

Further note that the patterns we find in the survey data align with the above-described expectations from literature. We find that expected positive effects in the human health domain are higher for countries where current regulations are weaker and discuss one example (ll.248-252).

Comment 2.32

167: "Domain 4"- The positive expected effects in this domain are also highly questionable as to their accuracy. (unless the question is understood as the tautology that SPM is defined as better for everything compared to pesticides)

Detailed response to 2.32

Thanks, we have now checked and extended the references for Domain 4, for example, accounting for studies on the economics of pesticide reduction in agriculture (Ziehmann et al., 2024, Finger et al., 2024) and economic spillover effects of pesticides (Skevas et al., 2013).

Comment 2.33

167: "Domain 5"- I am not going to check every reference but generally again when you define SPM as "ensuring economic and social equality" (while not doing the same for pesticide use) then don't be surprised if the answers say that SPM has a positive effect on equality.

167: "SOC1, 22"- The first cited reference (22) does not examine the impact of SPM versus pesticides on equality.

Detailed response to 2.33

Thanks, please again note that our definition of Sustainable Pest management does not exclude the use of pesticides, but closely aligns to definitions of Integrated Pest Management (also see our response to comment 2.24).

You refer to Parsa et al. (2014) (former reference 22), who analyze global obstacles for Integrated Pest management. They state: “Statistical analyses conducted on the responses showed significant differences between ratings of participants originating from high-income countries and those from developing countries, particularly for ratings on difficulties (Fig. 2). As a group, developing-country participants rated the statement “IPM requires collective action within a farming community” (IPM-3) as the most important obstacle.” (pp. 3891-3892). They go on to discuss: “Indeed, some pest management decisions are subject to a collective action dilemma (38), whereby the payoffs from adopting a technology depend on whether others adopt it too (39, 40). For example, smallholder farmers in Peru are encouraged to plow their previous-season potato fields to kill overwintering weevils before they colonize newly planted fields, but this practice is ineffective if their neighbors do not also plow their fields (41).” (p. 3892). This statement directly relates to indicator SOC1 on “equal distribution of all costs and benefits from pest management [...] between actors in a region.”

We have now checked and extended the references for Domain 5, adding reference to Rasmussen et al. (2018).

We have further checked all other references indicated in the Table and extended them.

Comment 2.34

192: Again the authors confound a negative effect of pesticide use on ecosystem services with a (unsupported, speculative) negative effect of pesticide use on agricultural productivity.

Detailed response to 2.34

Thanks, we now rephrase and additionally refer to Skevas et al. (2013):

“Although pesticides are typically cheaper and easier to apply compared with alternatives, they can negatively affect agricultural productivity, especially if incorrectly applied (e.g., Skevas et al., 2013). They can, for example, reduce ecosystem services (e.g., 13, 16, 17).” (ll. 201-203).

Please also see responses to comments 2.14 and 2.15.

Comment 2.35

205: This is an incorrect statement, as long you do not assume that SPM is equivalent or superior to pesticides on most dimensions (but then it's a tautological statement).

Detailed response to 2.35

Please see responses to comments 2.31 - 2.34.

Comment 2.36

211-218: More plausible explanations are that the experts have no clue what they are talking about (so they are not real experts), or they are answering tautological questions (where under the provided definition only one answer is logically possible). The first explanation is plausible if you ask people who are not real experts in the given domain but in another.

Having the expectation of a negative effect for "at least one indicator" across one or two(?) domains, and each indicator having (also) negative responses, are red herrings, as the relevant result is the mean expected negative effect for each indicator.

The last sentence on "well-documented adverse impacts" is again confounding the impact on a sub-outcome (e.g. pollination) with the net impact on the relevant total outcome (e.g. productivity), and it is the latter that the survey questions are about, not the former.

Detailed response to 2.36

Thanks, we have now collected additional data on survey participants and provide information on tenure, scientific expertise and output. Our data shows that the large majority of respondents are employed at independent research institutions, and are experienced, tenured researchers with a high scientific output in the field (ll. 1333-1354) (also see the response to comment 2.28).

We have also conducted several robustness checks on results across different groups of participants (for example by discipline, and academic vs. non-academic), and found results to be robust (ll. 1327-1332). In response to your comment, we additionally highlighted this in the manuscript now: “[...] and mean responses remain stable across different groups of respondents (e.g., by discipline or sampling strategy and between academic and non-academic experts Fig. S5-S7).” (ll. 225-227).

Additionally, in response to your comment we now deleted the statement on “adverse impacts”, since it was already mentioned in the paragraph before (also see response to comment 2.34).

Please also see responses to comments 2.14-2.15 and 2.31-2.34.

Comment 2.37

239: "especially in these regions..." But you should again explain why. I do not see an extensive comparison with "findings from the literature", or another plausible explanations. Have you considered whether developed country experts might have more accurate beliefs than developing country experts, due to different cultures or education?

Response to 2.37

Thanks, we sample experts with the same criteria for each region (see the response to comment 2.2). We oppose the view that experts in developing countries are less educated or that their culture affects the quality of their expert judgment. In our analysis, we control for regions of expertise and not for regions of origin. We believe that local experts in all regions have important perspectives to contribute.

Please also see the responses to comments 2.28 and 2.36.

Comment 2.38

243: "benefits in the human health domain..."- What could account for this?

Response to 2.38

Thanks, for your remark. The phrasing might have been unclear previously – it refers to the previous sentence and reference 69 (in the previous manuscript) which emphasizes the tighter pesticide regulations in Europe. We have now rephrased to make this clearer.

Comment 2.39

248: "We consistently find..."- What could be possible explanations?

Response to 2.39

General response:

Thanks, for the question. First note that our study presents the first global assessments and that this part of our analysis (section of the manuscript) is descriptive, i.e., we present and discuss potential drivers in the following section, based on our regression results. We can thus only hypothesize on potential reasons based on previous literature that is restricted to case studies for certain countries or regions. Further note that global agriculture is very heterogeneous and underlying processes might therefore also differ. Distinguishing these mechanisms on a more local level will therefore require other methodological approaches (ll. 483-194).

Detailed responses:

i) Correlation to lower incomes and HDIs: Considerable heterogeneity exists. While farmers in some poorer regions do not use sufficient crop protection at all (for example due to economic reasons, lack of access, or education), farmers in other regions overuse or misapply pesticides (for example due to a lack of training on crop protection). Related to that, Altieri (1994), Pretty et al. (2003) and Wang et al. (2024) discuss as potential reasons for a high success of ecological and integrated pest management practices in the global south the (lacking) availability of suitable chemical options, resource scarcity of farmers and cheaper alternative solutions, but also pesticide over-use and synergistic and ecologically beneficial effects of more diverse and pesticide reduced production systems.

i) Correlation to SDGs: We check correlation to SDGs as one of the described robustness checks. Results are consistent with general expectations on decreasing marginal benefits of sustainable pest management – where SDG levels are low, a transformation to sustainable pest management is related to higher benefits in respective indicator categories.

We have now added this discussion in the manuscript (ll. 265-29).

Comment 2.40

259: If you have reliable experts at hand you should not expect to have respondent-specific characteristics driving results (only random variation around the mean)? Or is this a sociological study about belief systems across cultures and disciplines?

Response to 2.40

Thanks. It is important to highlight that this is a survey-based study, eliciting expectations from experts, and not a (field) experimental study.

Expert surveys are regularly used in the field to assess important societal questions, where data is limited. They have, for example, previously been used to estimate global yield losses from pests (Savary et al., 2019), obstacles to integrated pest management in developing countries (Parsa et al., 2014), or the state of global plant health (Savary and GPHA, 2023) (also see the response to comment 2.2).

Controlling for respondent characteristics when doing a survey is a classic scientific practice and broadly established. It allows controlling for specific biases that may arise from different knowledge and expertise among participants. For example, discourses in different fields might differ and affect expectations of respondents. Using our approach, we can control for such differences and assess their influence on results instead of assuming, for example, that one discipline is “more right” about a certain topic than another. This approach thus reduces potential biases.

Moreover, heterogeneity in responses and its drivers is one of our research questions, and therefore of interest to our study.

Our approach allows us to exploit the heterogeneity in responses across different fields and provide a broad assessment of the research questions: We i) use established sampling procedures and standards (also see response to comment 2.2.), ii) have sampled a large and diverse number of experts (in terms of expertise and discipline) – which is required given the broad nature of the assessment, iii) and have controlled for differences in individual participant-specific attributes using statistical analyses (see the response to comment 2.31 for details).

To make this clearer we have now re-arranged the Materials and Methods section and provide an additional overview graph indicating the steps taken, sources and reasoning as well as conducted robustness checks (Fig. 3).

Comment 2.41

268: This "relative deviation" does not have an initiative interpretation, nor is clear why it is included here. Consider modifying the figure or better explaining it in the footnote, as each figure should be easy to understand on its own.

Response to 2.41

Thanks, we have now improved and simplified the description of the Figure.

Please also see the response to comment 2.26.

Comment 2.42

276: 12 is a questionable sample size

Response to 2.42

Our goal was to achieve over 30 responses for all regions and disciplines. We achieved this goal for all regions and disciplines, except Oceania (see section A2 for an extended discussion).

On statistical grounds, there is however no reason to exclude results from Oceania, since we report confidence intervals, accounting for sample sizes. Additionally, we again take a prudent approach here and very carefully interpret results for Oceania, considering the smaller sample size.

Comment 2.43

277: "relative to expected effects on global levels" Again this variable is not easy to understand. The logical thing would be to show the absolute results by region (where are these results?), not in relation to the global means.

Response to 2.43

We clearly state our research goals in the beginning: "We designed the survey to assess the expected direction of the effects at the global level (negative or positive), heterogeneity of effects across indicators, heterogeneity of effects across regions worldwide and the potential drivers of the expected effects. For analyzing drivers, we combine the survey data with external data (see Materials and Methods section 5)." (ll. 142-145).

We follow this structure in our study, reporting and discussing i) the direction of effects (Table 1), ii) heterogeneity across indicators (Figure 1a), iii) heterogeneity across regions (Figure 1b), and iv) potential drivers (Figure 2).

We deliberately take a prudent approach to interpreting levels of indicators and single responses (also see the response to comment 2.26). We report single responses in detail in Fig. S3-S6.

Comment 2.44

286: What is the reason for controlling for key characteristics of the respondents? I would make that clear to the reader

Response to 2.44

Thanks, we follow your suggestion and now write: "The survey data allowed us to control for key characteristics of the respondents in the analysis and mitigate potential biases, for example from different knowledge and expertise of participants." (ll. 305-307).

Also see the response to comment 2.40.

Comment 2.45

321: "less beneficial" - This is counterintuitive

325: "We interpret these findings..."- Why this interpretation? It seems a bit far-fetched.

Response to 2.45

Thanks, the first phrase is actually connected to the latter. We now rephrase to make this clearer in the manuscript: "We interpret these findings on biodiversity loss, consumer confidence and inequalities as an indication of the legacy effects of pesticide contamination." (ll. 346-348).

We further add additional references to support (Burian et al., 2024, Dakos et al., 2014).

Comment 2.46

332-336: Not obvious that these are the most plausible explanations. You didn't ask the reasons for the experts opinions so this is speculation.

Response to 2.46

Thanks, we now rephrase to make it clearer to readers that this is our interpretation of the data: "We interpret these results to suggest [...]" (l. 353).

We have added further references to support (Pretty et al., 2012, Pretty, 2018, Ratnadass, 2020).

Comment 2.47

355-357: What does this tell us about the accuracy of experts' beliefs?

Response to 2.47

Please see the response to comment 2.40.

Comment 2.48

363-365: Well, or maybe one discipline is more right than another discipline?

Response to 2.48

Please see the response to comment 2.40.

Comment 2.49

370: Tautology again

Response to 2.49

Please see the response to comments 2.31-2.36.

Comment 2.50

372: "potential costs"- But your experts say the transformation would be beneficial across all dimensions including the economic and social. These costs are factored in your H ("SPM more affordable")

Response to 2.50

Please see the response to comments 2.31-2.36.

Comment 2.51

376: Full elimination of synthetic pesticide use?

Response to 2.51

Please see the response to comment 2.24

Comment 2.52

379: I do not find this phrasing among the predefined measures in the questionnaire. Align.

Response to 2.52

Thanks. The definition in our questionnaire is very similar: “Identifying and providing access to effective and cost-efficient substitutes for pesticides.”

We have now used the exact same phrase in the manuscript to make this clearer.

Comment 2.53

387: But this also contradicts the notion that restricting pesticide use would lead to better outcomes. It underlines that the main questions of the survey are hypothetical, imagining a SPM that is nicer than synthetic pesticides yet does not exist.

Response to 2.53

Exactly, the goal of our survey was to assess what potential effects a reduction in pesticide use and transformation of pest management, in line with global and regional pesticide reduction goals could have. The survey is about expectations of experts on the effects of having a system, as defined in our survey, in place. This does not imply that a transformation to such a system is easy or free of costs.

In this second part we therefore assess potential policy pathways to reach these goals (ll. 390-423).

We now explicitly state this in the text to make it clearer to readers: “Despite our results of multiple expected benefits from a transformation to sustainable pest management, the pathway to such a transformation is not easy or free of costs. In the second part of the analysis, we therefore ask what is required to transform global pest management systems.” (ll. 390-392).

Comment 2.54

428-429: But your experts say the opposite, namely that the transformation is great for equality! The transformation has no negative impacts if we take your results at face value.

Response to 2.54

Thanks, we have now rephrased to make the statement more accurate: “Our finding of lower distributional and social benefits from transforming pest management in systems with high attainable yields and high pest pressure indicates that the benefits of a transformation are distributed heterogeneously across regions worldwide. This emphasizes the importance of a coherent global strategy. Future regional and international agreements should address the question of pesticide leakage in an interconnected world (e.g., 35). More specifically, we need to avoid environmental and social impacts, for example from Europe, being shifted to other parts of the world via changes in trade-flows (e.g., 104).” (ll. 449-456).

Comment 2.55

430: Your results say nothing of pesticide leakage.

Response to 2.55

Thanks, we have now rephrased to make our statement clearer. Please see the response to comment 2.54.

Comment 2.56

434: "important insights"- If the experts' beliefs are correct and you ask them the right questions, yes.

Response to 2.56

Please see the response to comment 2.31-2.36.

Comment 2.57

437-438: There are some regional assessments, for example for the EU's F2F strategy, which included reducing pesticide use. Or organic farming expansion scenarios. These assessments (indicating a negative impact on production and environment, possibly contradicting your experts) are themselves incomplete and flawed but should be mentioned in your manuscript.

Response to 2.57

Please see the response to comment 2.21.

Comment 2.58

699: Why? Post-harvest losses are very important.

Response to 2.58

Thanks, we acknowledge that post-harvest losses are important (that is why we mention them here). However, post-harvest losses and national biosecurity measures require different types of management measures and involve different stakeholders, requiring an additional survey and a different study set-up. We thus narrowed the scope to pest management in agricultural production in our study.

References

- Altieri, MA . 1994. Biodiversity and pest management in agroecosystems. New York: Haworth Press.
- Baker, N. J., Bancroft, B. A., & Garcia, T. S. (2013). A meta-analysis of the effects of pesticides and fertilizers on survival and growth of amphibians. *Science of the total environment*, 449, 150-156.
- Barzman, M., Bärberi, P., Birch, A. N. E., Boonekamp, P., Dachbrodt-Saaydeh, S., Graf, B., ... & Sattin, M. (2015). Eight principles of integrated pest management. *Agronomy for sustainable development*, 35, 1199-1215.
- Basu, P., Ngo, H. T., Aizen, M. A., Garibaldi, L. A., Gemmill-Herren, B., Imperatriz-Fonseca, V., ... & Vanbergen, A. J. (2024). Pesticide impacts on insect pollinators: Current knowledge and future research challenges. *Science of The Total Environment*, 954, 176656.
- Burian, A., Kremen, C., Wu, J. S. T., Beckmann, M., Bulling, M., Garibaldi, L. A., ... & Seppelt, R. (2024). Biodiversity–production feedback effects lead to intensification traps in agricultural landscapes. *Nature Ecology & Evolution*, 8(4), 752-760.
- Convention on Biological Diversity (2022). Kunming-Montreal Global Biodiversity Framework. Online: <https://www.cbd.int/doc/decisions/cop-15/cop-15-dec-04-en.pdf> [last accessed October, 28th, 2024].
- Costello, M. J., May, R. M., & Stork, N. E. (2013). Can we name Earth's species before they go extinct?. *Science*, 339(6118), 413-416.
- Dakos, V., & Bascompte, J. (2014). Critical slowing down as early warning for the onset of collapse in mutualistic communities. *Proceedings of the National Academy of Sciences*, 111(49), 17546-17551.
- Donkor, A., Osei-Fosu, P., Dubey, B., Kingsford-Adaboh, R., Ziwu, C., & Asante, I. (2016). Pesticide residues in fruits and vegetables in Ghana: a review. *Environmental Science and Pollution Research*, 23, 18966-18987.
- European Court of Auditors (2020). Special Report 05/2020: Sustainable use of plant protection products: limited progress in measuring and reducing risks. Online: <https://eca.europa.eu/de/publications?did=53001> [last accessed October, 25th, 2024].
- European Union (2020). Farm to Fork strategy. Online: https://food.ec.europa.eu/document/download/472acca8-7f7b-4171-98b0-ed76720d68d3_en?filename=f2f_action-plan_2020_strategy-info_en.pdf [last accessed October, 28th, 2024].
- Finger, R., Sok, J., Ahovi, E., Akter, S., Bremmer, J., Dachbrodt-Saaydeh, S., ... & Möhring, N. (2024). Towards sustainable crop protection in agriculture: A framework for research and policy. *Agricultural Systems*, 219, 104037.
- Fletcher, J., & NoghaniBehambari, H. (2024). The siren song of cicadas: early-life pesticide exposure and later-life male mortality. *Journal of Environmental Economics and Management*, 123, 102903.
- Frank, E. G. (2024). The economic impacts of ecosystem disruptions: Costs from substituting biological pest control. *Science*, 385(6713), eadg0344
- Gandara, L., Jacoby, R., Laurent, F., Spatuzzi, M., Vlachopoulos, N., Borst, N. O., ... & Crocker, J. (2024). Pervasive sublethal effects of agrochemicals on insects at environmentally relevant concentrations. *Science*, 386(6720), 446-453.
- Gerage, J. M., Meira, A. P. G., & da Silva, M. V. (2017). Food and nutrition security: pesticide residues in food. *Nutrire*, 42, 1-9.

- Giulioni C, Maurizi V, Castellani D, et al. The environmental and occupational influence of pesticides on male fertility: A systematic review of human studies. *Andrology*. 2022;10(7):1250-1271.
- Goulson, D., Nicholls, E., Botías, C., & Rotheray, E. L. (2015). Bee declines driven by combined stress from parasites, pesticides, and lack of flowers. *Science*, 347(6229), 1255957.
- Gross, K., & Rosenheim, J. A. (2011). Quantifying secondary pest outbreaks in cotton and their monetary cost with causal-inference statistics. *Ecological Applications*, 21(7), 2770-2780.
- Handford, C. E., Elliott, C. T., & Campbell, K. (2015). A review of the global pesticide legislation and the scale of challenge in reaching the global harmonization of food safety standards. *Integrated environmental assessment and management*, 11(4), 525-536.
- INSERM Collective Expertise Centre. Effects of pesticides on health: New data. Montrouge (FR): EDP Sciences; 2022. <https://www.inserm.fr/expertise-collective/pesticides-et-sante-nouvelles-donnees-2021/>
- Jones, B. A. (2020). Invasive species control, agricultural pesticide use, and infant health outcomes. *Land Economics*, 96(2), 149-170.
- Karalexi, M. A., Tagkas, C. F., Markozannes, G., Tseretopoulou, X., Hernández, A. F., Schüz, J., ... & Ntzani, E. E. (2021). Exposure to pesticides and childhood leukemia risk: A systematic review and meta-analysis. *Environmental pollution*, 285, 117376.
- Kazar Soydan, D., Turgut, N., Yalçın, M., Turgut, C., & Karakuş, P. B. K. (2021). Evaluation of pesticide residues in fruits and vegetables from the Aegean region of Turkey and assessment of risk to consumers. *Environmental Science and Pollution Research*, 28, 27511-27519.
- Klein, A. M., Vaissière, B. E., Cane, J. H., Steffan-Dewenter, I., Cunningham, S. A., Kremen, C., & Tscharntke, T. (2007). Importance of pollinators in changing landscapes for world crops. *Proceedings of the royal society B: biological sciences*, 274(1608), 303-313.
- Kleijn, D., Winfree, R., Bartomeus, I., Carvalheiro, L. G., Henry, M., Isaacs, R., ... & Potts, S. G. (2015). Delivery of crop pollination services is an insufficient argument for wild pollinator conservation. *Nature communications*, 6(1), 7414.
- Kogan, M. (1998). Integrated pest management: historical perspectives and contemporary developments. *Annual review of entomology*, 43(1), 243-270.
- Landrigan, P. J., Fuller, R., Acosta, N. J., Adeyi, O., Arnold, R., Baldé, A. B., ... & Zhong, M. (2018). The Lancet Commission on pollution and health. *The lancet*, 391(10119), 462-512.
- Li, Y., Miao, R., & Khanna, M. (2020). Neonicotinoids and decline in bird biodiversity in the United States. *Nature Sustainability*, 3(12), 1027-1035.
- Lozowicka, B., Abzeitova, E., Sagitov, A., Kaczynski, P., Toleubayev, K., & Li, A. (2015). Studies of pesticide residues in tomatoes and cucumbers from Kazakhstan and the associated health risks. *Environmental monitoring and assessment*, 187, 1-19.
- Mesnage, R., Straw, E. A., Antoniou, M. N., Benbrook, C., Brown, M. J., Chauzat, M. P., ... & Zioga, E. (2021). Improving pesticide-use data for the EU. *Nature ecology & evolution*, 5(12), 1560-1560.
- Möhring, N., Ingold, K., Kudsk, P., Martin-Laurent, F., Niggli, U., Siegrist, M., ... & Finger, R. (2020). Pathways for advancing pesticide policies. *Nature food*, 1(9), 535-540.

- Nicholson, C. C., Knapp, J., Kiljanek, T., Albrecht, M., Chauzat, M. P., Costa, C., ... & Rundlöf, M. (2024). Pesticide use negatively affects bumble bees across European landscapes. *Nature*, *628*(8007), 355-358.
- Nielsen, H. Ø., Konrad, M. T. H., Pedersen, A. B., & Gyldenkærne, S. (2023). Ex-post evaluation of the Danish pesticide tax: a novel and effective tax design. *Land Use Policy*, *126*, 106549.
- Parsa, S., Morse, S., Bonifacio, A., Chancellor, T. C., Condori, B., Crespo-Pérez, V., ... & Dangles, O. (2014). Obstacles to integrated pest management adoption in developing countries. *Proceedings of the National Academy of Sciences*, *111*(10), 3889-3894.
- Philippe, V., Neveen, A., Marwa, A., & Basel, A. Y. A. (2021). Occurrence of pesticide residues in fruits and vegetables for the Eastern Mediterranean Region and potential impact on public health. *Food Control*, *119*, 107457.
- Pretty, J. N., Williams, S., & Toulmin, C. (2012). *Sustainable intensification: increasing productivity in African food and agricultural systems*. Routledge.
- Pretty, J. (2018). Intensification for redesigned and sustainable agricultural systems. *Science*, *362*(6417), eaav0294.
- Rasmussen, L. V., Coolsaet, B., Martin, A., Mertz, O., Pascual, U., Corbera, E., ... & Ryan, C. M. (2018). Social-ecological outcomes of agricultural intensification. *Nature Sustainability*, *1*(6), 275-282.
- Rumschlag, S. L., Casamatta, D. A., Mahon, M. B., Hoverman, J. T., Raffel, T. R., Carrick, H. J., ... & Rohr, J. R. (2022). Pesticides alter ecosystem respiration via phytoplankton abundance and community structure: Effects on the carbon cycle?. *Global Change Biology*, *28*(3), 1091-1102.
- Ratnadass, A. (2020). Crop protection for agricultural intensification systems in Sub-Saharan Africa. *Sustainable Agriculture Reviews* *39*, 1-34.
- Sánchez-Bayo, F. (2014). The trouble with neonicotinoids. *Science*, *346*(6211), 806-807.
- Sánchez-Bayo, F., Goulson, D., Pennacchio, F., Nazzi, F., Goka, K., & Desneux, N. (2016). Are bee diseases linked to pesticides?—A brief review. *Environment international*, *89*, 7-11.
- Savary, S., Ficke, A., Aubertot, J. N., & Hollier, C. (2012). Crop losses due to diseases and their implications for global food production losses and food security. *Food security*, *4*(4), 519-537.
- Savary, S., Willocquet, L., Pethybridge, S. J., Esker, P., McRoberts, N., & Nelson, A. (2019). The global burden of pathogens and pests on major food crops. *Nature ecology & evolution*, *3*(3), 430-439.
- Savary, S., & Global Plant Health Assessment Project (GPHA). (2023). A global assessment of the state of plant health. *Plant Disease*, *107*(12), 3649-3665.
- Schneider, K., Barreiro-Hurle, J., & Rodriguez-Cerezo, E. (2023). Pesticide reduction amidst food and feed security concerns in Europe. *Nature Food*, *4*(9), 746-750.
- Skevas, T., Stefanou, S. E., & Lansink, A. O. (2013). Do farmers internalise environmental spillovers of pesticides in production?. *Journal of Agricultural Economics*, *64*(3), 624-640.
- Sim, J. X., Doolette, C. L., Vasileiadis, S., Drigo, B., Wyrsh, E. R., Djordjevic, S. P., ... & Lombi, E. (2022). Pesticide effects on nitrogen cycle related microbial functions and community composition. *Science of the Total Environment*, *807*, 150734.

- Smith, L. G., Kirk, G. J., Jones, P. J., & Williams, A. G. (2019). The greenhouse gas impacts of converting food production in England and Wales to organic methods. *Nature communications*, *10*(1), 4641.
- Ssemugabo, C., Bradman, A., Ssempebwa, J. C., Sillé, F., & Guwatudde, D. (2022). An assessment of health risks posed by consumption of pesticide residues in fruits and vegetables among residents in the Kampala Metropolitan Area in Uganda. *International journal of food contamination*, *9*(1), 4.
- Stanley, D. A., Garratt, M. P., Wickens, J. B., Wickens, V. J., Potts, S. G., & Raine, N. E. (2015). Neonicotinoid pesticide exposure impairs crop pollination services provided by bumblebees. *Nature*, *528*(7583), 548-550.
- Syed, J. H., Alamdar, A., Mohammad, A., Ahad, K., Shabir, Z., Ahmed, H., ... & Eqani, S. A. M. A. S. (2014). Pesticide residues in fruits and vegetables from Pakistan: a review of the occurrence and associated human health risks. *Environmental Science and Pollution Research*, *21*, 13367-13393.
- Verger, P. J., & Boobis, A. R. (2013). Reevaluate pesticides for food security and safety. *Science*, *341*(6147), 717-718.
- Wang, Y. H., Wee, S. L., De Faveri, S. G., Gagic, V., Hossain, S., Cheng, D. F., ... & Krutmuang, P. (2024). Advancements in Integrated Pest Management strategies for *Bactrocera dorsalis* in Asia: current status, insights, and future prospects. *Entomologia Generalis*.
- Wood, T. J., Michez, D., Paxton, R. J., Drossart, M., Neumann, P., Gerard, M., ... & Vereecken, N. J. (2020). Managed honey bees as a radar for wild bee decline?. *Apidologie*, *51*, 1100-1116.
- Wyckhuys, K. A., Furlong, M. J., Zhang, W., & Gc, Y. D. (2022). Carbon benefits of enlisting nature for crop protection. *Nature Food*, *3*(5), 299-301.
- Zakowski, E. and Mace, K., 2022. Cosmetic pesticide use: quantifying use and its policy implications in California, USA. *International Journal of Agricultural Sustainability*, *20*(4), pp.423-437.
- Ziehm, E., Möhring, N., & Finger, R. (2024). Economics of herbicide-free crop production. *Applied Economic Perspectives and Policy*.

Reviewer #3

General comments reviewer 3

Comment 3.1

This work presents some interesting findings, but the current manuscript does not discuss important contextual, procedural, and definitional limitations of the work. These need to be addressed and the potential of biases that are introduced need to be discussed before the work is published. Here are some of the issues.

Response to 3.1

Thank you for your constructive comments that helped us to better highlight contributions and limitations of the study. We are positive that we were able to address all of your points in the revised version of the manuscript.

Comment 3.2

The definition of “Sustainable Pest Management” as “no or minimal use of pesticides” can create a number of biases in the responses and results. The experts could well have interpreted this to be a definition that essentially limits “alternative pest management” to “no pesticide use”, which is even stricter than organic agriculture (where biologically-based pesticides are readily accepted and used. The use of the term “pesticides”, with no attempt to differentiate among different groups or risks is overly broad. Far more interesting would have been to deepen the conversation and response to a risk-reduction approach to pest management.

Response to 3.2

Thank you for your remark.

Following your comments and comments of R1, we have now improved readability and organization of the Materials and Methods section and better referenced it in the main body of the text to provide readers a better overview of the methodological approach used and the robustness checks performed.

We have further substantially extended our section on the limitations of the study, explicitly mentioning that it does not refer to risk reduction: “Additionally, we define the target as a system with minimal or no pesticide use, which closely aligns to the well-known concept of Integrated Pest Management. However, we do not explicitly refer to “pesticide risk” reduction, since this term is often not clearly defined and assessments in terms of pesticide risk reduction are less common in the field, and therefore harder to judge for experts from various disciplines.” (ll. 472-476).

We have chosen a definition that is closely aligned with the principles of integrated crop protection (e.g., Barzman, 2021, Deguine 2023), which are widely known: “[...] may contain a minimal use of synthetic pesticides but mainly consist of single alternatives or a mix of alternatives: Such as biological solutions (e.g., biocontrol, bio-pesticides), agronomic solutions (e.g., adapted crop rotations, field hygiene) technical solutions (e.g., mechanical weed control, smart farming), breeding solutions (e.g., resistant and adapted varieties) and system redesign (e.g., systems that favor natural solutions for pest control).”

We explicitly included “biocontrol” and “bio-pesticides” in the list of substitutes we gave respondents as part of the definition of “Sustainable Pest Management” in the survey.

Further, negative effects on the environment have also been found for pesticides that are used in organic agriculture (e.g., Lamichhane et al., 2018 for an overview), which is why we have not used terms like “synthetic pesticides” in our definition.

We conducted several robustness checks of the survey design *ex-ante* and *ex-post* of the survey execution (ll. 947-957). In pre-tests of the survey and post-survey interviews, respondents confirmed that their understanding aligned with Integrated Pest Management principles (as was our above-described intention).

Importantly, the choice of a definition or description of a production system always remains subjective to some degree (especially when no official definition exists) and has to cover heterogeneous production systems in a global survey. This is an inherent limitation of expert assessments compared to statistical approaches or simulation models (which do not exist in this field on global levels). We now explicitly acknowledge and describe this in the limitations section of the manuscript: “An inherent limitation of expert assessments is that results are dependent on the chosen definitions in the survey. We here addressed this by an external and internal validation of our study design before its execution, by choosing a large and diverse sample (different geographic regions, professions and disciplines), controlling for important respondent characteristics in our regression analysis, and a general focus on heterogeneity and drivers instead of levels of indicators (see Fig. 3 for an overview).” (ll. 467-472).

We agree that a risk reduction should be the ultimate goal of sustainable pest management approaches. However, we deliberately did “[...] not refer to “pesticide risk” reduction, since this term is often defined differently, for example, using different indicators (24), or has different connotations. Further, assessments in terms of pesticide risk reduction are less common in the field, and therefore harder to judge for experts from various disciplines (e.g., 36)” (ll. 848-851).

Comment 3.3

An important source of potential bias is the group of experts. To provide transparency about their interests and allegiances, data on their potential conflicts of interest should have been collected and reported, as is the norm in scientific publications. What has been the funding sources for their research? What organizations currently or have recently employed them? This information is fundamental in deciding how unbiased this group of experts is in their responses to the potential transition to using fewer high-risk products that are still owned and produced by powerful corporations in the plant protection arena. The analogy to the tobacco researchers who received massive funding from the tobacco industry 30-50 years ago is appropriate here. Without clearly dispelling potential sources of bias, the results of the surveys of this group of experts is questionable and potentially unreliable.

Response to 3.3

Thank you for this important remark.

Biases may arise in all directions in a field that is debated controversially. We account for this in our research design by sampling a diverse set of experts from different disciplines and for different regions globally, providing maximum transparency about our sampling research design, data and codes, and by then controlling for important expert characteristics in the analysis. We, for example, find in our regression analysis, that the discourse in different disciplines seems to affect expectations of experts, and control for these effects in our analysis.

We would be happy to share the list of respondents with you and the other reviewers, however this is not possible due to data privacy rules. To address your concern, we therefore chose an approach that provides additional transparency about scientific expertise, tenure, institutions, as well as funding and potential conflicts of interest without compromising data privacy of respondents:

1. We went back to the list of survey participants (identified through the network of co-authors, or as corresponding authors of relevant papers, corresponding to 76.4% of all respondents) and

gathered data on tenure, scientific expertise and experience (output), and funding and potential conflicts of interest.

2. We find that 91.1% are employed at public research institutions or universities.
3. Respondents are on average experienced, tenured researchers: 53% are professors or senior researchers, 42% are researchers and 5% are junior researchers. They have 3543 citations and an H-index of 22.3 on average.
4. To address concerns about conflicts of interest and funding sources we took the following approach: For those respondents which were identified as corresponding authors of publications in the field, we went through these publications and identified any potential conflicts of interest and funding sources reported. We found that the large majority either did not mention any specific outside funding (35%) or had funding from public institutions (50%). 13% received funding from foundations and 2% from companies. Out of all papers not publicly funded, only one paper declared a conflict of interest (working in industry and a public research institute at the same time), but authors stated that the independence of the research results was assured in a funding arrangement.

We now additionally report this check in Appendix section A2 (ll. 1333-1354).

Importantly, our main results remain stable when excluding different groups of experts (e.g., by discipline or by sampling method, see Fig. S5, Fig. S7).

Finally, despite our best efforts, we cannot rule out remaining biases completely, and now explicitly state this in the limitations section of the manuscript: “Still, despite our best efforts, we cannot rule out remaining biases completely.” (ll. 476-477).

Comment 3.4

The role of the pesticide industry should be explicitly discussed as an impediment to the transition of more sustainable pest management. The huge industry lobbies, advocates, funds research, coerces, and uses other means to promote their interests in maintaining the status quo to sell more of their products. They have successfully imposed dominant narratives of the “need” for pesticides. This powerful narrative is typically unquestioned by many experts in the field.

Response to 3.4

Thank you for your comment. We have now integrated it in the discussion and explicitly mention the important role of food-value chain actors and industry in the transition to sustainable pest management.

We now write: “Advancing on this question requires a holistic and place-based approach that not only accounts for farmers, but also for the important, systemic role of food-value chain actors and industry in the global transition to sustainable pest management (36, 100-102).” (ll. 420-423).

Please also see our response to comment 3.3.

Comment 3.5

Another weakness of the survey was to not differentiate among different production systems within regions. Smallholder maize farmers in sub-Saharan Africa face a completely different socio-economic context than large-scale commercial maize farmers in southern Africa.

Response to 3.5

Thanks, this is a valid point. We actually had to balance including more details on the agricultural system of expertise with the level of detail and the length of the survey when designing it - since our goal was to reach a large and diverse sample of experts and cover global scales.

We agree that our survey can only give a first indication on global and regional effects. We thus acknowledge that our study can “[...] only be a starting point for identifying focal points for more regional- and local- in-depth analysis.” (ll. 484-485).

We make this more explicit in the limitations section of our manuscript now, using your example: “Future research should consider other important characteristics of production systems, such as farm and farmer characteristics, which we could not include in our global survey. For example, smallholder maize farmers in sub-Saharan Africa face a completely different socio-economic context than large-scale commercial maize farmers in Southern Africa.” (ll. 479-482).

Additionally, an important indication of such heterogeneity in the assessed production systems in a region would be a higher coefficient of variation in responses (compared to other regions). We have therefore added an additional analysis and computed coefficients of variation for indicator responses and compared them across regions (see Supplementary Materials, Section A3).

We find that out of all indicators and regions, respondents’ expectations were most heterogeneous for Europe, North America and Oceania and the domains of “Economics and Farmer Livelihood” and “Food and Nutrition Security”.

We now report our findings in the manuscript (see Supplementary Materials, Section A3) and refer to them in the main text: “Finally, out of all indicators and regions, respondents’ expectations were most heterogeneous for Europe, North America and Oceania and the domains of “Economics and Farmer Livelihood” and “Food and Nutrition Security” (see Supplementary Materials, Section A3). This indicates that experts had diverging opinions on expected effects especially for these indicators and regions. This result highlights the importance of looking into potential drivers of expected effects.” (ll. 270-274).

Comment 3.6

Another approach to answering the question about the transition from “status-quo” crop protection to alternatives, would be to use an empirical methodology. This could be done by looking at “status-quo” crop protection methods and results and compare them to farmers who use organic pest management methods, so that they can receive organic certification for their crops. The worldwide production of organic crops is now large enough that many such directly comparable (same crops, same regions, etc.) could be found and compared. The evidence would be far more conclusive than the responses from a potentially biased group of responders to a blunt survey.

Response to 3.6

Thanks for this comment. Studies comparing organic and conventional yields (Seufert and Ramankutty, 2017), economic outcomes (Crowder and Reaganold, 2015) and adoption (Möhring et al., 2024) exist, and can give some first indications. However, there are important differences between conversion to organic production and pesticide reduction with implications for the assessed indicators, such as restrictions on the use of synthetic fertilizer and important changes in farm management, such as closed nutrient cycles (Möhring et al., 2024). Further, pesticides used in organic agriculture can be harmful to the environment and are therefore under pressure to be reduced as well (e.g., Lamichhane et al., 2018 for an overview).

The currently emerging pesticide-free production systems in Europe would provide better empirical evidence of potential effects of pesticide reduction – but are so far only used in few countries and crops, with no global data available (Finger and Möhring, 2024).

We agree that a statistical approach or a simulation model would be interesting for assessing potential effects. However, both a consistent empirical approach and simulation models for estimating yield,

economic and environmental effects of reducing pesticide reduction are currently missing on regional and global levels. Importantly, even if such an approach would exist, data on global scales to estimate such models is scarce (also see Finger et al., 2024). This study is a first step in this direction, providing insights and data for future research efforts.

We now make this clearer in the manuscript and include the above argumentation in the revised limitations and future research section. (ll. 467-494).

References

- Crowder, D. W., & Reganold, J. P. (2015). Financial competitiveness of organic agriculture on a global scale. *Proceedings of the National Academy of Sciences*, *112*(24), 7611-7616.
- Finger, R., Sok, J., Ahovi, E., Akter, S., Bremmer, J., Dachbrodt-Saaydeh, S., ... & Möhring, N. (2024). Towards sustainable crop protection in agriculture: A framework for research and policy. *Agricultural Systems*, *219*, 104037.
- Finger, R., & Möhring, N. (2024). The emergence of pesticide-free crop production systems in Europe. *Nature Plants*, *10*(3), 360-366.
- Lamichhane, J. R., Osdaghi, E., Behlau, F., Köhl, J., Jones, J. B., & Aubertot, J. N. (2018). Thirteen decades of antimicrobial copper compounds applied in agriculture. A review. *Agronomy for sustainable development*, *38*(3), 28.
- Möhring, N., Muller, A., & Schaub, S. (2024). Farmers' adoption of organic agriculture—a systematic global literature review. *European Review of Agricultural Economics*, jbae025.
- Schreinemachers, P., Grovermann, C., Praneetvatakul, S., Heng, P., Nguyen, T. T. L., Buntong, B., ... & Pinn, T. (2020). How much is too much? Quantifying pesticide overuse in vegetable production in Southeast Asia. *Journal of Cleaner Production*, *244*, 118738.
- Seufert, V., & Ramankutty, N. (2017). Many shades of gray—The context-dependent performance of organic agriculture. *Science advances*, *3*(3), e1602638.
- Staudacher, P., Brugger, C., Winkler, M. S., Stamm, C., Farnham, A., Mubeezi, R., ... & Günther, I. (2021). What agro-input dealers know, sell and say to smallholder farmers about pesticides: a mystery shopping and KAP analysis in Uganda. *Environmental Health*, *20*, 1-19.
- Wyckhuys, K. A., Gu, B., Fekih, I. B., Finger, R., Kenis, M., Lu, Y., ... & Hadi, B. A. (2024). Restoring functional integrity of the global production ecosystem through biological control. *Journal of Environmental Management*, *370*, 122446.

Response to review manuscript NCOMMS-24-32226A: “Expected Effects of a Global Transformation of Agricultural Pest Management”

We would like to thank the anonymous reviewers for providing insightful comments and suggestions for improving and balancing the manuscript, adding important explanations, discussions and examples. Please find detailed point-by-point responses to each comment in the respective sections below. We wish to thank the reviewers for the opportunity to elaborate our arguments and improve the manuscript as a whole.

Reviewer #1

Remarks to authors of reviewer 1

Dear Authors,

Thank you for addressing my comments and suggestions in your revised manuscript titled “Expected Effects of Transforming Agricultural Pest Management across Global Scales.” I appreciate the effort you have put into improving the clarity, structure, and overall quality of the manuscript.

The revisions have, IMHO, significantly strengthened the paper, and I am pleased with how you have addressed the points I raised. Your detailed responses demonstrate a thorough consideration of the feedback, and the manuscript now reads more cohesively.

I have raised a few additional comments and suggestions in my latest review, such as refining (still) long sentences and addressing other minor points noted in your response letter. However, these are minor adjustments.

Congratulations on producing such an impactful and comprehensive piece of work. Your study provides valuable insights into a critical global challenge and will be a significant contribution to the field.

I wish you all the best with the final stages of publication and your future research endeavours.

With my best regards

Response to remarks to authors.

Thank you very much for the compliments, the rigorous assessment and the constructive comments on our manuscript, also in the previous round of revisions. We have implemented your remaining comments. Please find a detailed point-by-point response below.

General comments reviewer 1

Comment 1.1

Long Sentences and Readability. While readability has improved, some sentences remain dense and could benefit from being broken down. For example:

"Combining agronomic developments for improved and diversified production systems with new opportunities from digital innovations and genomic breeding techniques is seemingly a promising avenue to increase productivity while significantly reducing pesticide use and risk".

Response to 1.1

Thanks, we have now gone through the complete text again, following your comment and simplified sentences throughout.

Comment 1.2

The terms "current status" and "current situation" appear interchangeably throughout the manuscript. It would be beneficial to homogenize this terminology for consistency.

Response to 1.2

Thanks, we have homogenized this throughout and now use current status.

Comment 1.3

The legend for Figure 3 requires further explanation to ensure clarity.

Response to 1.3

Thanks, we have edited the legend of Figure 3 and now explain the content of the Figure in it.

Comment 1.4

The policy discussion section would benefit from more actionable and specific recommendations. For instance, the discussion could highlight concrete policy steps or tools to facilitate the transition to sustainable pest management.

Response to 1.4

Thank you. We now expand on this in our discussion and write: "Wuepper et al. (105) highlighted the global importance of policies for the reduction of pesticide risk, and Möhring et al. (25) described pathways for supporting this transformation in the context of European agriculture and showed that a policy mix is needed. This can, for example, consist of providing information and facilitating access to machinery by farmers. But in places where big trade-offs remain this will not be sufficient and a transformation will likely additionally require the development of new farming systems and technologies combined with economic instruments to overcome barriers (25). The recent emergence of pesticide-free production systems in Europe shows that the combination of integrated pest management approaches, together with a mix of private and public funding can enable large-scale pesticide-free systems that offer a "third way" between conventional and organic agriculture (106). However, different approaches will be required for globally heterogeneous regions and agricultural production systems." (ll. 422-434).

Comment 1.5

L410-413. The sentence "In general, alternative pest management requires combinations of management actions implemented at various scales and adapted to the production context, while pesticides are often more broadly applicable (e.g., 30, 32, 36 for a discussion)" is unclear and could be rephrased for better understanding.

Response to 1.5

Thanks, we have now split this sentence up and rephrased: "Pesticides are usually easily and broadly applicable in different contexts. Substituting pesticides often requires combinations of several alternative management actions (e.g., adjustment of the crop rotation and crop variety together with mechanical pest control), at various scales (e.g., the field and landscape scale), and adapted to the

production context (e.g., growing conditions and pest pressure; see 26, 31, 33 for a discussion).” (ll. 413-417).

Comment 1.6

While the section “Holistic Policy Approaches Required” identifies alternative solutions as a priority, it does not specify examples or evidence-based approaches for alternatives. For instance, it would be helpful to include examples of proven or promising non-pesticide strategies that could be scaled globally.

Response to 1.6

Thanks, we now give such examples in the text. First, we rephrase and extend a previous sentence writing: “Agronomic developments for improved and diversified production systems that prevent pests can be combined with new opportunities from digital innovations, genomic breeding techniques and the development of novel biopesticides (100-102). This is a promising avenue, which has already shown the possibility to increase productivity while significantly reducing pesticide use and risk (103-104).” (ll. 408-412).

Then we add: “The recent emergence of pesticide-free production systems in Europe shows that the combination of integrated pest management approaches, together with a mix of private and public funding can enable large-scale pesticide-free systems that offer a “third way” between conventional and organic agriculture (106). However, different approaches will be required for globally heterogeneous regions and agricultural production systems.” (ll. 428-434).

Comment 1.7

"Locally Adapted Bundles of Support Measures": This phrase is quite broad. Do you have examples of successful case studies or pilot programs? This would make the argument more robust and relatable.

Response to 1.7

Thanks, we have now added the example of pesticide-free production systems in Europe. Please see the response to comment 1.6.

Comment 1.8

The section refers “Holistic Policy Approaches Required” to "our results" multiple times without explicitly connecting these to specific findings. For example, which results demonstrate the insufficiency of technical feasibility for alternatives?

Response to 1.8

Thanks. We now refer to results more explicitly in this section.

Comment 1.9

Still on the section “Holistic Policy Approaches Required”. I think it would be beneficial to reflect on specific policy challenges faced by low-income (limited regulatory capacity) versus high-income regions.

Response to 1.9

Thank you for the recommendation. We think that is an important and valuable point but refrain from just differentiating low- and high-income countries, as differences will probably be more nuanced. We therefore now more generally write: “For example, natural conditions, the strength of institutions for implementing and enforcing rules, the capacity of governments to support farmers and their knowledge base, access to markets and technology, and the economic capacity of farmers to purchase

inputs and implement practices differ strongly across regions and will determine appropriate strategies.” (ll. 434-437).

Reviewer #2

General comments reviewer 2:

Comment 2.1

This is not the definition used in the manuscript, nor in the survey. The manuscript states: „pest management systems with no or minimal use of pesticides, using (a combination of) ... alternative pest management practices...their implementation leading to a substantial reduction of current pesticide use levels“.

The survey questionnaire's definition under C also includes: „SPM cornerstone of sustainable agriculture", "meet society's current and future needs", „ensuring food security“ etc., "overreliance on synthetic pesticides" / „minimal use“.

The crucial point is that the definition has the superiority of the SPM baked into it, so the answer can only be that it is superior. The authors cannot redress this major mistake made during the survey design. They do not respond to it here but the proper way to deal with it is to admit the study is gravely flawed and not seek publication.

Refers to the previous round of review, also see previous comment and response 2.1.

Response to 2.1

Thanks, in order to address your concern, we have i) added additional quantitative evidence on concerns of respondents and how we have addressed them in the paper, and ii) have made additional changes in the manuscript text directly responding to your comment.

First, we now provide additional quantitative evidence to respond to your comment in the paper. More specifically, we have now added a section on concerns of survey respondents and how we addressed them in our study to the Supplementary Information (Supplementary Note 4 “Consideration of respondents’ concerns”).

Your comment suggests that the framing of the system of sustainable pest control is biased and proleptic. In order to check if your concern was shared among the respondents of the survey, which mostly consist of senior researchers from a broad range of disciplines and regions, we have looked in detail into comments made by respondents at the end of the survey. The comment function at the end of the survey provided the opportunity to respondents to signal any problems/comments with the survey and contact the authors. We assume that respondents with strong concerns would have used this opportunity after taking the time to fill out the whole survey. Indeed, overall, 84 out of 517 respondents (16.2%) gave comments, with a broad distribution across disciplines and expert regions.

Importantly, these 84 comments show that your concern was not perceived as a major concern for the majority of respondents (only shared partly by one person or 0.2% of all respondents). More respondents even expressed concerns that the definition was too general and will be interpreted very differently by different respondents (12 persons/ 2.3% of all respondents), or that our definition was too close to the definition of Integrated Pest Management (6 persons/ 1.2% of all respondents). This shows that, as expected in an interdisciplinary and global study, respondents from different scientific fields and global regions had different concerns and perceptions of the study and its design, which were sometimes even opposing.

We have now summarized all of these concerns in the Supplementary Information (Supplementary Note 4 “Consideration of respondents’ concerns”) and show how we addressed them in our analysis, robustness checks and discussions.

Second, we take your concern very seriously and took the following additional steps to respond to it:

- To provide more transparency, we have now additionally included the full survey text in the supplementary materials (Supplementary Note 5).
- We have now additionally summarized respondents' comments and how we addressed them (Supplementary Note 4).
- We have now additionally added in the paragraph on limitations that "Biases may occur depending on how individuals understand and perceive the questions." (ll. 489-490).
- We have now additionally discussed potential biases arising from the used definition and how we address these concerns in the limitations section (ll. 490-502) and in the new section on the consideration of respondents' concerns (Supplementary Note 5, ll. 223-241).

Third, we would like to specifically highlight that **we clearly find and discuss trade-offs** especially for the most productive regions (Europe, North America and Oceania) and the domains of Economics and Food Security. **This shows that respondents had diverging opinions on expected effects** (see Supplementary Note 3). This empirical finding contrasts your suggestion that the framing of the system only allows for a positive response. This finding aligns with our above finding that comments and concerns of experts from different scientific disciplines and global regions may oppose at times. It highlights the need to clearly communicate this heterogeneity of opinions (we do so at several instances) and investigate potential drivers (which we do in the regression analyses). We report our findings of heterogeneous expert opinions in the manuscript (see Supplementary Note 3) and refer to them in the main text: "Finally, out of all indicators and regions, respondents' expectations were most heterogeneous for Europe, North America and Oceania and the domains of "Economics and Farmer Livelihood" and "Food and Nutrition Security" (see Supplementary Notes 3). This indicates that experts had diverging opinions on expected effects especially for these indicators and regions and aligns with our expectation of heterogeneous effects in these domains. This result highlights the importance of looking into potential drivers of expected effects." (ll. 270-275).

Finally, we empirically find that our results and the identified heterogeneity in results closely aligns with expectations from literature. We document and discuss this extensively throughout the manuscript text: We find that for the majority of the 24 indicators, results were in line with expectations from literature (see ll. 186-227 for an extensive discussion). We further show that even for the few indicators where results did not align with general expectations from literature, literature provides potential indications and explanations. Importantly, for all of these indicators we find strong heterogeneity in responses over production systems and regions – and the direction of heterogeneous responses again aligns with expectations from literature (ll. 232-252). We further find that correlations between indicator responses and country-level indicators of human development and progress on related SDGs again consistently align with expectations from literature (ll. 253-269). Finally, we assess a broad range of potential socio-economic and environmental drivers and again find that results align with expectations from literature on the direction of relations (ll. 328-389).

Comment 2.11

The authors ignore the positive impact of pesticide use (through higher yields) on biodiversity.

Refers to the previous round of review, also see previous comment and response 2.11.

Response to 2.11

Thanks, in response to your comment we have made several changes in the text that we explain below.

We acknowledge the role of pesticides as effective, cheap and easy to apply solutions for crop protection in our manuscript: "Pesticides typically provide affordable and efficient pest management

solutions that can be applied in a wide range of production contexts (5).” (ll. 61-62). We further highlight its important role for food production on a limited amount of global land: “Effective management of agricultural pests is thus essential for farmers’ livelihoods and for producing sufficient and diverse food for a growing population on a limited amount of land.” (ll. 58-59).

We acknowledge the land-sparing effect of effective pest control (lower damage and higher yields) and its important role for natural biodiversity.

We want to stress again that our SPM scenario is not about zero pest control, but about more sustainable forms of pest control. So how we can maintain controlling pests, while reducing potential adverse effects of pesticides.

In response to your comment, we have now further emphasized these points in the manuscript.

First, we write: “Importantly, we chose a definition that allows for pesticide use due to its importance for effective pest control in certain contexts (see Methods section for a detailed explanation how the survey was designed and Supplementary Note 5 for the full survey text).” (ll. 128-131).

Second, we add: “This aspect is crucial, because pesticides are often effectively controlling pests and a reduction of pesticides must be accompanied by the implementation of alternative measures of pest management to protect crops.” (ll. 166-168).

Third, following your comments, we now clarify in the paper that cropping systems shall be evaluated by efficiency metrics such as yield per unit of input (e.g., energy, water) and by impacts on ecosystems and ecosystem services and not per-se by the type of production, e.g., whether and how much pesticides are used (ll. 440-443). We also cite Cassman and Grassini (2020) accordingly.

Fourth, following your argumentation, we now clarify that leakage shall be avoided. That is, if lower pesticide footprints in one part of the world imply lower productivity to be compensated by higher imports, this may increase the intensity and footprint of production and land use in other parts of the world. Such mechanisms may lead to simply shifting environmental and human health impacts from one country to another and shall be avoided. (ll. 473-480).

→ Thus, we promote the idea of SPM on global levels, not only for specific countries or regions.

Comment 2.12

Pesticides can also positively affect biodiversity and ecosystem functioning (through the land-sparing mechanism).

Negative effects on selected individual species, but the evidence does not support a negative effect on biodiversity (the number of different species).

The manuscript does not acknowledge the fact that currently pesticides have a positive effect on biodiversity via the land-sparing mechanism.

Adversely affecting ecosystem services is different from adversely affecting biodiversity through adversely affecting biodiversity. The sentence in the revised manuscript „However, pesticides can adversely affect biodiversity and ecosystem functioning” obfuscates that distinction. Taking into account evidence for demonstrated harm of pesticide use on certain animals, the revised manuscript sentence also still confuses individual species with biodiversity. And the sentence fails to acknowledge positive impacts of pesticides on biodiversity via the land-sparing mechanism.

The sentence is thus still very bad and should not be published as it stands.

You should also be careful to distinguish “ecosystem function” from “ecosystem services”, they are not the same thing.

Refers to the previous round of review, also see previous comment and response 2.12.

Response to 2.12

Thanks, we have now taken your concern regarding the land-sharing and land-sparing effects of effective pest control into account and added additional statements in the revised manuscript (see the response to comment 2.11).

Regarding your other comments, only a small fraction of living species is thought to be described (Costello et al. 2013). But certain negative effects of pesticides have been identified on a broad range of species.

Note that several studies in high-level journals (including this journal) have been recently published showing negative effects of pesticides on a wide range of non-target species, adding to the cited evidence in the manuscript. See for example:

- Wan, N. F., Fu, L., Dainese, M., Kiær, L. P., Hu, Y. Q., Xin, F., ... & Scherber, C. (2025). Pesticides have negative effects on non-target organisms. *Nature Communications*, 16(1), 1360.
- Gandara, L., Jacoby, R., Laurent, F., Spatuzzi, M., Vlachopoulos, N., Borst, N. O., ... & Crocker, J. (2024). Pervasive sublethal effects of agrochemicals on insects at environmentally relevant concentrations. *Science*, 386(6720), 446-453.

Further note that evidence suggests many ecosystem functions such as pest control (Frank et al. 2024, Gross and Rosenheim, 2011), pollination (Stanley et al., 2015) and nutrient and carbon cycling (e.g., Rumschlag et al., 2021, Sim et al., 2022) are negatively impacted by pesticides. We are here not arguing whether it is “due to biodiversity” or “due to functional trait identity”.

Comment 2.13

„Can“ is weak and obfuscating. It would be better to be more precise (health effects not causally proven and very small in Jones 2020, no effects at a broad range of normal pesticide use quantities in Larsen et al 2017, old and no longer used substances in Fletcher & Nogh. 2024; Frank 2024 +7.9% effect size in the context of the low base rate of ~7/1000 births extremely small).

If there are negative health effects from pesticide use, they appear to be very small and occur only in extreme or historical use cases. Furthermore, positive effects of pesticide use (via improved diet and nutrition, in poorer farming contexts also through higher household income) should be mentioned alongside (<https://doi.org/10.1016/j.cropro.2007.03.022>).

Refers to the previous round of review, see also previous comment and response 2.13.

Response to 2.13

Thanks. We totally agree that there are important benefits of pesticides and that they make an important contribution to crop protection. We make this clear in the manuscript from the very beginning of the paper (first paragraph) and now further emphasize this point (see response to 2.11). Throughout the text, we argue for a balanced and careful approach to reducing pesticide use and highlight benefits of pesticides.

To further emphasize this, we now additionally cite the paper you referenced above (ll. 61-62).

All of the studies cited by us here on health effects use causal identification strategies, are published in top field- or interdisciplinary journals, and mostly find relatively large effect sizes on sensitive parts of the population, like infants. Given the difficulty of causally identifying such health effects in a

heterogeneous environment and populations we think that this is sufficient evidence to make such a statement. Further note that we have here only cited a handful of studies since space is limited. For a broader overview, a more detailed discussion on the health effects of pesticides and some of the underlying physiological mechanisms also see, Landrigan et al., (2018), INSERM (2022), Karalexi et al. (2021) and Giulioni et al., (2022).

Comment 2.14

„this“ can be read as linking back to biodiversity. But the link of biodiversity and agricultural production via natural enemies, pollination and soil function is not established, it is individual species that are driving these effects, not the diversity of different species present. And again, ecosystem services do not automatically follow from ecosystem function (imagine an ecosystem that is very functional at harbouring -if you want also a very diverse set of- crop pests and diseases- that is a clear disservice to agriculture).

Refers to the previous round of review, also see previous comment and response 2.14.

Response to 2.14

Thank you, we have now revised the sentence and write: “Ecosystem services such as natural pest control (13-15), insect-pollination (16-18) and soil functions (19) contribute to agricultural production.” (ll. 63-65).

Comment 2.15

I still have not seen good evidence showing that pesticide use decreases yields by killing off natural enemies / pollinators / soil functions at sufficient scale to counterbalance the positive yield impact of pest control/nitrogen/water use efficiency afforded by pesticides. This should be acknowledged in the manuscript.

Refers to the previous round of review, also see previous response and comment 2.15.

Response to 2.15

Thanks. Please see the responses to 2.11 and 2.13 for adjustments that we have made.

Comment 2.17

The legislative proposal was withdrawn and whether another attempt at legislation will be made is highly doubtful. It would be appropriate to mention the withdrawal or simply not mention the 50% goals anymore.

The CBD is concerned with risk, not quantities.

Refers to the previous round of review, also see previous comment and response 2.17.

Response to 2.17

Thanks, please note that the Farm-to-Fork strategy still stands, and that quantitative reduction targets in a range of 30-50% have further been implemented on national levels in the EU in the mandated National Action Plans for pesticide reduction. We have now added a reference to these National Action Plans.

Please also note that the Farm-to-Fork strategy envisions a reduction of i) pesticide quantity, ii) most toxic substances, and iii) pesticide risk.

We support the focus on pesticide risks as a reduction target, not quantities, in the manuscript.

As the Farm-to-Fork policy might soon be replaced by another policy we now refer more broadly to the efforts of the EU member states without referring to the concrete reduction targets (that vary across National Action Plans).

We now write: “In response to health and environmental concerns, substantial policy targets for pesticide reduction have been set on policy agendas globally. For example, the Post-2020 Global Biodiversity Framework of the United Nation sets a reduction target of 50% for overall pesticide risks until 2030 (25). The European Union member states further aim to reduce the use and risk of chemical pesticides and the use of more hazardous pesticides (26). Both emphasize the reduction of environmental and health risks from pesticide use (see 25, 27-28 for extensive discussions on this question).” (ll. 71-76).

Comment 2.21

Peer-review in the sense of a publication in a scientific journal should not be the only criterion when deciding whether to mention or not to mention a study. Many studies published in journals are terrible and important data and evidence can be found outside these journals. In this case I would argue that these F2F study results are important and should be mentioned in the manuscript. If you state limitations of these studies, you should do the same for many of the studies you cite when you talk negatively about pesticides (see comments 2.12-2.15).

Refers to the previous round of review, see also previous comment and response 2.21.

Response to 2.21

Thanks.

We were not only talking about “limitations”, but serious violations of research ethics, i.e., serious conflicts of interest that were not declared in the case of one of the studies (<https://www.wur.nl/en/newsarticle/dialogue-about-the-pitfalls-of-contract-research.htm>).

This is an important reason why peer-review by respected journals is an important safeguard. Not every scientist may agree with all published methods/data/conclusions all the time, but good journals at least ensure compliance with basic research ethics. This is hard to control in case of “white papers” or institutional studies. We therefore take a very prudent approach to such unpublished studies.

In response to your comment, we have now added references to these studies in the text (except the study with undeclared conflicts of interest) (ll. 92-93).

Comment 2.24

I remain unconvinced. The survey framing is strongly biased and the answers already baked into the definition.

Refers to the previous round of review (also see comment and response 2.24 in previous round of review).

Response to 2.24

Please see our response to comment 2.1.

References

- Cassman, K.G., Grassini, P. A global perspective on sustainable intensification research. *Nat Sustain* **3**, 262–268 (2020). <https://doi.org/10.1038/s41893-020-0507-8>
- Frank, E. G. (2024). The economic impacts of ecosystem disruptions: Costs from substituting biological pest control. *Science*, *385*(6713), eadg0344
- Gandara, L., Jacoby, R., Laurent, F., Spatuzzi, M., Vlachopoulos, N., Borst, N. O., ... & Crocker, J. (2024). Pervasive sublethal effects of agrochemicals on insects at environmentally relevant concentrations. *Science*, *386*(6720), 446-453.
- Gross, K., & Rosenheim, J. A. (2011). Quantifying secondary pest outbreaks in cotton and their monetary cost with causal-inference statistics. *Ecological Applications*, *21*(7), 2770-2780.
- Giulioni C, Maurizi V, Castellani D, et al. The environmental and occupational influence of pesticides on male fertility: A systematic review of human studies. *Andrology*. 2022;*10*(7):1250-1271.
- INSERM Collective Expertise Centre. Effects of pesticides on health: New data. Montrouge (FR): EDP Sciences; 2022. <https://www.inserm.fr/expertise-collective/pesticides-et-sante-nouvelles-donnees-2021/>
- Karalexi, M. A., Tagkas, C. F., Markozannes, G., Tseretopoulou, X., Hernández, A. F., Schüz, J., ... & Ntzani, E. E. (2021). Exposure to pesticides and childhood leukemia risk: A systematic review and meta-analysis. *Environmental pollution*, *285*, 117376.
- Landrigan, P. J., Fuller, R., Acosta, N. J., Adeyi, O., Arnold, R., Baldé, A. B., ... & Zhong, M. (2018). The Lancet Commission on pollution and health. *The lancet*, *391*(10119), 462-512.
- Wan, N. F., Fu, L., Dainese, M., Kiær, L. P., Hu, Y. Q., Xin, F., ... & Scherber, C. (2025). Pesticides have negative effects on non-target organisms. *Nature Communications*, *16*(1), 1360.
- Rumschlag, S. L., Casamatta, D. A., Mahon, M. B., Hoverman, J. T., Raffel, T. R., Carrick, H. J., ... & Rohr, J. R. (2022). Pesticides alter ecosystem respiration via phytoplankton abundance and community structure: Effects on the carbon cycle?. *Global Change Biology*, *28*(3), 1091-1102.
- Sim, J. X., Doolette, C. L., Vasileiadis, S., Drigo, B., Wyrsh, E. R., Djordjevic, S. P., ... & Lombi, E. (2022). Pesticide effects on nitrogen cycle related microbial functions and community composition. *Science of the Total Environment*, *807*, 150734.
- Stanley, D. A., Garratt, M. P., Wickens, J. B., Wickens, V. J., Potts, S. G., & Raine, N. E. (2015). Neonicotinoid pesticide exposure impairs crop pollination services provided by bumblebees. *Nature*, *528*(7583), 548-550.

Response to review manuscript NCOMMS-24-32226B: “Expected Effects of a Global Transformation of Agricultural Pest Management”

We would like to thank the anonymous reviewers for providing insightful comments and suggestions for improving and balancing the manuscript, adding important explanations, discussions and examples. Please find detailed point-by-point responses to each comment in the respective sections below. We wish to thank the reviewers for the opportunity to elaborate our arguments and further improve the manuscript.

Reviewer #4

Remarks to authors of reviewer 4

Dear Authors,

The Reviewer expresses its great appreciation, congratulation that you targeted such an important topic, with overarching approaches, developing questionnaire, collecting over 500 responses globally (based on selected global regions).

Extremely difficult task, enormous differences globally. But the scientific (and broader) community needs your planned paper.

First of all, the Reviewer is less competent for the data analysis, methods but he should like to support your Manuscript (improvement) from the viewpoint of readers (be it scientists, policy makers, students, environmentalists, etc).

Reviewer aims at contributing to better common understanding of the survey, its results and subsequent interpretation.

Response to remarks to authors.

Thank you very much for the assessment and the constructive comments on our manuscript. We are confident that we were able to address all of your comments in the revised manuscript. Please find a detailed point-by-point response below.

General comments reviewer 4

Comment 4.1

L 52: "insect pests".... Please change for "animal pests" otherwise you disregard slugs, mites, birds, vertebrates, etc. UN FAO IPPC and EU definitions say "pests are all harmful organisms".

Response to 4.1

Thanks, we have changed the definition as you have suggested.

Comment 4.2

L 60 (and later in the entire text: "pesticides" as a definition is fully misunderstood or differently used globally. Reviewer understands Pesticides (to control pests, be it human disease vectors, urban rats, crop pests) and be it synthetic chemical plant protection product or natural plant extracts, plant oils, sulfur, microbes, entomopathogenic fungi, etc.

Reviewer suggests to DEFINE right in the beginning what "pesticides" stand for in your Manuscript, Assume "chemical/synthetic" active substances. Or?

Response to 4.2

Thanks, we have followed your suggestion and now write “We here specifically focus on pre-harvest pesticides in agriculture, which typically provide affordable and efficient pest management solutions that can be applied in a wide range of production contexts” (ll. 61-63).

We specifically do not exclude copper and sulphur-based pesticides in our discussion, which are used in organic agriculture, since they are widely used, it has been shown that these can also have negative effects, and there are also efforts under way to reduce their use (Lamichhane et al., 2018).

Comment 4.3

L63: Ecosystem functioning seems to be a bit cloudy, maybe ecological functions and subsequent ecosystem services?

Response to 4.3

Thanks, we broadly wanted to cover both ecosystem functions and services (e.g., Sekercioglu, 2010).

To make this clearer, we have reformulated the sentence in response to your comment and now write: “However, pesticides can adversely affect biodiversity as well as ecosystem functioning and services (e.g., IPBES, 2019, Li et al., 2020, Frank, 2024).” (ll. 63-64).

Comment 4.4

L 66: Reviewer strongly disagrees with "resistance development" as a specific risk. Resistance development in populations of target organisms is a "natural process", response of target populations to specific/multiple selection pressure (be it pesticide, agronomic practice, behavior, etc).

Response to 4.4

Thank, we completely agree that resistance development is a natural process in response to selection pressure. We here wanted to highlight that current pest management systems accelerate this process (e.g., Palumbi, 2001 for an overview) and thus will make it more difficult to continue in current systems in the future.

In order to address your concern that this sentence is misunderstood we have now separated it from the previous statement and reformulated. We now write: “Finally, resistances to widely used active ingredients are increasing, narrowing available pest management strategies (Palumbi, 2001).” (ll. 67-68).

Comment 4.5

L 68-70: Increasing pest pressure under climate change is likely one sided approach. Some pests occur, others disappear, their natural enemies may develop their increased population, thus it is more complex.

Response to 4.5

Thanks, we acknowledge that the previous formulation was simplifying the complex problem too much. In response to your comment, we therefore now specify (referring to Deutsch et al., 2018 and Chaloner et al., 2021): “In the future, the importance of effective pest management will further increase with elevated global food demand, changing consumption patterns and expected changes in the distribution and abundance of insect pests and plant pathogens under climate change (Muller et al., 2020, Deutsch et al., 2018, Gouel and Guimbard (2019), NAS (2021), Chaloner et al., (2021)” (ll. 69-71).

Comment 4.6

L 75: "more hazardous" seems to be not scientific. Pls look at UN FAO and other definitions.

Response to 4.6

Thanks, we agree, but this is a literal citation from the EU policy document where the target is defined as “[...] a 50% reduction in the use and risk of chemical pesticides and a 50% reduction in the use of more hazardous pesticides” (see for example: https://food.ec.europa.eu/plants/pesticides/sustainable-use-pesticides/pesticide-reduction-targets-progress_en).

To emphasize this, we have now put the definition in quotation marks and highlighted this by referring to the policy definitions in the reference.

Comment 4.7

L77: "For a long time"... is a bit questionable. Maybe until recent years (2025)...

Response to 4.7

Thanks, we are here referring to the long history of other (non-synthetic) means of pest control, which have been previously used by default (as synthetic pest control was not available) and have been formalized as Integrated Pest Management in recent decades (e.g., Kogan 1998).

In response to your comment, we have reformulated and now write: “For decades, agricultural production systems worldwide have been promoted in which pesticides are partially or entirely substituted with other means of pest control, e.g., biological, mechanical, breeding or agronomic solutions.” (ll. 79-81).

Comment 4.8

L 127 "minimal use": Reviewer suggest to avoid non-scientific "minimal"? use.

Response to 4.8

Thanks, we are here referring to Integrated Pest Management principles, which recommend pesticide use only after all other options have been exhausted (this is therefore from a technical viewpoint and in simplified words the minimal use of pesticides possible).

In response to your comment, we have now clarified this in the manuscript: “The baseline was defined as the current status of pest management in each production system and region, and the target scenario was defined as pest management with no or minimal use of pesticides (i.e., pesticide use as a last resort, as defined in Integrated Pest Management principles) using combinations of alternative practices in pest management.” (ll. 129-132).

Comment 4.9

L 132 "global and societal": Rev. understands global as spatial scale, societal as context environment. Might be rephrased.

Response to 4.9

Thank you, we have now rephrased and write: “Large-scale global expert surveys are commonly used to address important questions of global relevance to society, for which data are limited.” (ll. 136-137).

Comment 4.10

L 170-172: Missing where, when?

Response to 4.10

Thanks, we have now reformulated and specified: “Thus, essential information, e.g., studies that holistically cover all potential impact domains and global regions, for the identification of priorities,

trade-offs and synergies and the definition of strategies for the advancement of sustainable pest management is currently missing.” (ll. 172-175).

Comment 4.11

L 243: Reviewer calls the attention of the authors, that globally wrong fake news (input-intensive/extensive" phrasings are widespread. I strongly suggest to separate "material and energy input" from knowledge input.

Response to 4.11

Thanks, we have rephrased and now write: “Experts on production systems in Europe and North America, which are highly productive and operate with similarly high material and capital input, envision lower potential benefits due to sustainable pest management - across nearly all indicators.” (ll. 246-248).

Comment 4.12

L410: pls correct for NGT (New Gemonic Technics), see relevant EU documents.

Response to 4.12

Thanks, we have adjusted the wording in line with your suggestion.

Comment 4.13

L 428: "Pesticides Free"??? Means free of natural products, microbials, etc? At least, pls correct the popular non-scientific and misleading wording.

Response to 4.13

Thanks, this is referring to recently emerging systems that operate without synthetic pesticides. Following your comment, we have now specified this in the text: “The recent emergence of production systems that operate without synthetic pesticides in Europe shows that the combination of integrated pest management approaches, together with a mix of private and public funding can enable large-scale systems that offer a “third way” between conventional and organic agriculture (Finger and Möhring, 2024)” (ll. 436-440).

Comment 4.14

L 462-465: Sorry, very misleading non-factual, non scientific sentence, pls. correct.

Response to 4.14

Thanks, in response to your concerns we have now deleted this sentence.

Comment 4.15

L 561-562: Pls rephrase (as per "pesticide" definition). Therefor NO??? or MINIMAL??? should be rephrased. (What is minimal?).

Response to 4.15

Thanks, in response to your comment, we have now added an explanation “Here “no or minimal use” refers to the notion of pesticide use as a last resort, for example, as defined in Integrated Pest Management principles.” (ll. 577-578) (also see the response to comment 4.8).

Comment 4.16

L 575: "Pests and Pathogens", pls rewrite. ("Pests, incl. plant pathogens") I assume authors did not want to include animal and human pathogens?

Response to 4.16

Thanks, we have rephrased as you have suggested.

Comment 4.17

L 578: "Natural pest control" is non-scientific. The examples are fine but might be non-chemical pest control (Conservation, augmentative biocontrol, use of natural active substances, low risk products, etc. should be referred to).

Response to 4.17

Thank you for the recommendation. In response to your comment, we have now deleted the phrase "natural pest control" here.

Comment 4.18

Good luck for publishing your great outputs.

Response to 4.18

Thank you for the support!

References

- Chaloner, T. M., Gurr, S. J., & Bebber, D. P. (2021). Plant pathogen infection risk tracks global crop yields under climate change. *Nature Climate Change*, *11*(8), 710-715.
- Deutsch, C. A., Tewksbury, J. J., Tigchelaar, M., Battisti, D. S., Merrill, S. C., Huey, R. B., & Naylor, R. L. (2018). Increase in crop losses to insect pests in a warming climate. *Science*, *361*(6405), 916-919.
- Finger, R., & Möhring, N. (2024). The emergence of pesticide-free crop production systems in Europe. *Nature Plants*, *10*(3), 360-366.
- Frank, E. G. (2024). The economic impacts of ecosystem disruptions: Costs from substituting biological pest control. *Science*, *385*(6713), eadg0344.
- Gouel, C., & Guimbard, H. (2019). Nutrition transition and the structure of global food demand. *American Journal of Agricultural Economics*, *101*(2), 383-403.
- IPBES (2019). Intergovernmental science-policy platform on biodiversity and ecosystem services. Global Assessment Report on Biodiversity and Ecosystem Services of the Intergovernmental Science-Policy Platform on Biodiversity and Ecosystem Services.
- Kogan, M. (1998). Integrated pest management: historical perspectives and contemporary developments. *Annual review of entomology*, *43*(1), 243-270.
- Lamichhane, J. R., Osdaghi, E., Behlau, F., Köhl, J., Jones, J. B., & Aubertot, J. N. (2018). Thirteen decades of antimicrobial copper compounds applied in agriculture. A review. *Agronomy for sustainable development*, *38*(3), 28.
- Li, Y., Miao, R., & Khanna, M. (2020). Neonicotinoids and decline in bird biodiversity in the United States. *Nature Sustainability*, *3*(12), 1027-1035.
- Muller, A., Schader, C., El-Hage Scialabba, N., Brüggemann, J., Isensee, A., Erb, K. H., ... & Niggli, U. (2017). Strategies for feeding the world more sustainably with organic agriculture. *Nature communications*, *8*(1), 1-13.

NAS (2021). The Challenge of Feeding the World Sustainably: Summary of the US-UK Scientific Forum on Sustainable Agriculture. The National Academies Press, Washington, DC.

Palumbi, S. R. (2001). Humans as the world's greatest evolutionary force. *Science*, 293(5536), 1786-1790.

Sekercioglu, C. H. (2010). Ecosystem functions and services. *Conservation biology for all*, 2010, 45-72.

Reviewer #5

Thank you very much for the thorough assessment and the constructive comments on our manuscript. We are positive that we were able to address all of your comments in the revised manuscript. Please find a detailed point-by-point response below.

General comments reviewer 5:

Comment 5.1

1. Lines 56-57: “for the five major food crops worldwide”. Which are these five major food crops?

Response to 5.1

Thanks, we are here following the definitions of the cited reference (Savary et al., 2019). These are wheat, rice, maize, potato and soybean.

In response to your comment, we have now added this information in the text: “They are estimated to vary between 26% and 40% (Oerke, 2006; based on literature and field trial data) and between 17% and 30% (Savary et al., 2019; based on expert assessments) for the five major food crops (wheat, rice, maize, potato and soybean) worldwide with a large variation across regions.” (ll. 54-57).

Comment 5.2

1. Lines 60-61: “2.6 million tons in 2020” The latest FAO data can be used.

Response to 5.2

Thanks, in response to your comment we have now updated this to the most recent FAO data from 2025 (for the year 2023).

Comment 5.3

2. Lines 71-72: “the Post-2020 Global Biodiversity Framework of the United Nation sets a reduction target of 50% for overall pesticide risks until 2030”. Is this target achievable or is it meant for only academic discussions? It has been five plus decades since integrated pest management was globally adopted as the cardinal principal of plant protection strategy, yet pesticide use in agriculture has surged, and the same has been highlighted by the authors. The authors have used “sustainable pest management” terminology and not “integrated pest management”. IPM is based on the principle of “sustainable pest management”? The authors acknowledge this in the “Supplementary Information” at point 4. Although IPM endeavoured to promote sustainable pest management, thereby reducing synthetic pesticide use, it would have been better to analyse the obstacles to integrated pest management and the global transformation of IPM for sustainable agricultural production. 3. As happened with IPM, there are tens, if not hundreds, of definitions, and the same will be the fate of the concept “sustainable pest management”. Authors could have focused on IPM to conduct a survey of experts AND FARMERS on “Transformation of Agricultural Pest Management”. Sustainable pest management has been operationalised as “pest management systems with 4. no or minimal use of pesticides, using (a combination of) ... alternative pest management practices”.

Response to 5.3

Thanks, for your comment.

Regarding the first part of your comment: We totally agree that the reduction targets are ambitious, given the fact that pesticide use has been stable or increasing in important agricultural regions of the world. As you noted, we point out in the article that more action is needed in order to fulfill these targets. This is exactly the key motivation of our article: we want to support relevant actors in taking

next steps for advancing these targets by providing critical information on expected effects of a transformation of pest management – this information is currently missing for important regions and indicators worldwide.

In order to further emphasize this in response to your comment, we now state “Achieving the targets requires a shift in policies” (l. 84).

Regarding the second part of your comment: Thanks a lot for your comment. As you have noted, hundreds of definitions as well as different, important paradigms and production systems for reaching reduction targets exist. We agree that Integrated Pest Management is the most prominent one. But by only focusing on Integrated Pest Management, we would have excluded other approaches that aim at a reduction of pesticide use, such as agroecological pest management, which has gained in importance in the last decades. Different approaches are strongly advocated by some researchers in the domain due to some concerns around limitations of IPM (e.g., Deguine et al., 2021, 2023). We therefore decided to not only focus on Integrated Pest Management and use a more inclusive definition that captures the essence of these major approaches. We share your concern, and have tested the definition before the survey was conducted in the group of interdisciplinary co-authors, and in a workshop with stakeholders from different sectors (Material and Methods, section 4). Further, after the survey was conducted, we tested how respondents actually perceived this definition. Our results show that the perceptions of the respondents aligned with our intentions (see Material and Methods, section 4; Supplementary Materials, Section A4).

In order to respond to your concern, we have now added this more explicitly in the manuscript. In the revised version we now added: “We therefore refer to “Sustainable Pest Management” and not only, for example, Integrated Pest Management in order to provide a more inclusive definition that also captures other important approaches, such as agroecological pest management. We tested before and after conducting the survey that our intentions aligned with participants’ perceptions (Material and Methods section 4, Supplementary Materials, Section A4).” (ll. 561-565).

Comment 5.4

5. What are alternative adoptable pest management practices available with and disseminated by survey institutes that should have been part of the survey? Thereafter, a farmers' survey should have been conducted to assess the adoptability and sustainability of these practices at the farm level across different farming systems. Without involving farmers reached by the sampled institutes and 517 experts, the exercise is top-down. What are alternative pest management technologies for farmers in different global hotspots? The practices they adopt or reject, and pesticide use, would have made this survey impactful. What attitude farmers have towards alternative pest management tactics in different regions across different crops for reducing pesticide use and making pest management sustainable could have been another study variable?

Response to 5.4

Thanks for your comment. We completely agree that this is a top-down approach and that this approach should be complemented with more detailed studies on a local and farmer level. Databases, such as EPPO (<https://gd.eppo.int/>) or the Farmer’s Toolbox for Integrated Pest Management (<https://datam.jrc.ec.europa.eu/datam/mashup/IPM/index.html>) provide a more general (but patchy) overview of available practices to farmers in different regions, and the SUPPORT project (<https://he-support.eu/>) is currently conducting a more detailed survey of IPM practices of farmers in some crops and countries of the EU. However, conducting such a study on a farm-level for all global regions and all indicators that we covered here would require an enormous effort and large funding (and suggest this as a task for future research). Further, even data on a global and regional level that we provide here for the first time, is not available yet. We therefore see our study as a first step. We assess levels, heterogeneity and drivers of pest management and expected effects from a transformation on a global

and regional level with a sample of highly qualified experts from different regions, disciplines and sectors, who have a broad overview of ongoing challenges in their region. Our insights can then be used subsequently by researchers to focus on cases that are of special interest and have a high potential. Zooming in on a more local level after our study will be key to identifying solutions. We had previously recognized this in the discussion, and in response to your comment we have further emphasized this now: “Studies that provide information on the choice and development of locally adapted practices and solutions will be crucial for the development of sustainable pest management. This will require farm-level surveys and research.” (ll. 517-520).

Please note that the experts indeed assessed current levels of sustainable pest management in the survey (see Table 2) and that we account for this variable in our analysis of potential drivers (Fig. 2).

You make an important point, and we totally agree that comparing attitudes of farmers in different regions regarding pest management practices could be an important point for future research. In response to your comment, we have now added: “Future research should consider other important characteristics of production systems, such as farm and farmer characteristics and attitudes to different types of pest management, which we could not include in our global survey.” (ll. 510-512).

Comment 5.5

6. 517 experts are from the international organisations (Supplementary Table 5). What have been the outcomes of programmes implemented by these organisations in “reducing pesticide use”? Experts from National Agricultural Research Systems of countries having large cultivated land should also have been part of the sampling plan..

Response to 5.5

Thank you. This might have been a misunderstanding.

Our respondents come from a broad range of institutions, universities, extension agencies, governmental departments and industry – as you have suggested. They are just not all listed in Supplementary Table S5 due to data privacy reasons.

We explain the reasons in more detail below and have also adjusted the manuscript text in response to your comment since this was not described in sufficient clarity before.

Our sampling strategy relied on two pillars (see Supplementary Note 2):

First, we identified 454 senior experts on pest management across disciplines and professions from leading universities and research institutions worldwide (mainly from research, but also extension services, policy, and industry). We contacted them to respond to the survey and asked them to additionally forward the survey to senior members of their organizations in other regions worldwide and received 223 complete responses. Personal information on participants is strictly confidential due to ethical and legal restrictions, and we can therefore not report names or institutions of single participants. However, we provide a list of the contacted institutions in Table S5.

- ➔ In Table S5, we give an overview of institutions we contacted. However, we contacted the 454 single senior experts. Due to data privacy, we cannot disclose their names or institutions (they come from leading universities and extension agencies from all over the world). So those institutions are not in Table S5. We therefore additionally give an aggregate overview of the type of institutions, the tenure and the scientific output of respondents in Supplementary Note 2.
- ➔ Since this was not sufficiently clear in the manuscript before, we have now clarified this in response to your comment: “Personal information on participants is strictly confidential due to ethical and legal restrictions, and we can therefore not report participant names. However, we provide a list of the contacted institutions in Table S5. Note that Table S5 gives an overview

of contacted institutions but not of the universities and extension agencies the identified senior experts belonged to (excluded due to data privacy).” (Supp. Materials II. 74-78).

Second, we identified international and national academic experts through the literature database Scopus. More specifically, we contacted all corresponding authors of articles published in peer-reviewed journals on the subject of “sustainable pest management” over the last 10 years. We chose “sustainable pest management” as a key word, as our clear research focus is on the assessment of potential effects from a transition to sustainable pest management. Double entries were then discarded, leaving us with 1531 contacts in total. Out of the 1531 identified corresponding authors, we received 294 complete responses, amounting to a response rate of 19.3%. This strategy and response rate is in line with recent sampling strategies of researchers conducting global expert assessments in adjacent fields, such as Economics (e.g., 4). We provide information about the type of institutions, the tenure and the scientific output of corresponding authors, as well as indicated funding sources.

Comment 5.6

7. This article is simply an academic exercise. We cannot test hypotheses solely based on the survey of 517 experts. Experts' and farmers' data triangulation would have added to the robustness of the results for drawing inferences..

Response to 5.6

Thanks. We see our study as a first step. We assess levels, heterogeneity and drivers of pest management and expected effects from a transformation on a global and regional level with a sample of highly qualified experts from different regions, disciplines and sectors, who have a broad overview of ongoing challenges in their region. Our insights can then be used subsequently by researchers to focus on cases that are of special interest and have a high potential. Zooming in on a more local level after our study will be key to identifying solutions.

In response to your comment, we have now made this clearer in the manuscript: “Studies that provide information on the choice and development of locally adapted practices and solutions will be crucial for the development of sustainable pest management. This will require farm-level surveys and research.” (ll. 517-520).

(Please also see our response to comment 5.4).

Comment 5.7

8. Data on “pesticide use and pests” should have been added from the selected countries of the 517 experts, which would add value to the article.

Response to 5.7

Thanks, you are totally correct, pesticide use and pests are of course very important potential drivers – and this is why we actually account for both in our regression analysis.

For pesticide use we even use a risk indicator that accounts not only for pesticide use volume but also for the environmental hazards of used pesticides (which can be very heterogeneous) → This is the variable “Level Pesticide Risk” used in our regression analysis – it is based on the “Risk Score” from Tang et al. (2021).

For pests, we use regional estimates on pest pressure → This is the variable “Potential Pest Pressure” in our regression analysis, based on the estimates of Oerke (2006).

You can find a description of the variables in Table 2. Since we have noted that the variable description in the regression analysis and in the Table did not use the same names it is clear that this led to confusion. In response to your comment, we have therefore aligned variable names now.

Further, since it has not been clear enough that we account for pesticide use and pest pressure we now explicitly highlight this, in response to your comment, in the manuscript text: “To assess the factors that drive the expected effects of the transformation, we spatially matched the survey data with a wide set of country- and regional-level data on key characteristics of agricultural production systems (see the Methods section). We then analyzed the data using multiple regression analyses (see Fig. 2 and Supplementary Table 3 for detailed regression results). This allows us to control for key characteristics of the production system (e.g., level of pesticide risk, pest pressure).” (ll. 308-312).

Comment 5.8

9. Highest potential of sustainable pest management in low-income regions: Why “Experts on highly productive and input-intensive production systems in Europe and North America envision lower potential benefits due to sustainable pest management -across nearly all indicators?” Many times, experts are far removed from the farms/ farm households. Experts mostly do not collect primary data themselves in the developing countries, so how can they conclude “..... potential win-win scenarios due to the reduction of pesticide use and the transformation of pest management” AND WHY EXPERTS FROM INDUSTRIALIZED AGRICULTURE SEE LOWER POTENTIAL. What about the input-intensive production systems in many Green Revolution areas of developing countries, for example, Indian Punjab and Haryana, which are highly productive?.

Response to 5.8

Thanks, we here aggregate results on a continent level, since this allows us to compare general trends in different regions of the world within the limited space of the article.

But we totally agree with you that different continents can be very heterogeneous with regards to their agricultural systems and that such a broad aggregation hides a lot of this heterogeneity.

We further agree with you that this heterogeneity should be better highlighted in the article. In response to your comment, we have therefore created a new Figure (Supplementary Figure 14) which shows that expected effects within regions are very heterogeneous.

In response to your comment, we now further highlight this in the manuscript at different points:

First, we highlight that the regional results are averages (and can therefore hide heterogeneity):

“Experts on production systems in Europe and North America, which are very productive and operate with high material and capital input, on average, envision lower potential benefits due to sustainable pest management - across nearly all indicators. In contrast, experts in South America, Africa and Asia, on average, identified higher levels of potential benefits.” (ll. 246-251).

Then we now additionally, explicitly state that these regional averages hide a lot of heterogeneity in expected effects and refer to the additional figure that we have created in response to your comment: “Further note that the averages are aggregated on a large regional scale and that important heterogeneity within regions exist (Supplementary Fig. 14). This result highlights the importance of looking into potential drivers of expected effects.” (ll. 278-281).

In order to account for this heterogeneity, we conduct the regression analysis (Fig. 2), which accounts for characteristics of production systems on a finer scale.

Comment 5.9

10. Lines 162-170: In the results section, the authors are again explaining the rationale for the study. The results section should exclusively focus on survey results..

Response to 5.9

Thanks, in general we agree with your recommendation. However, the rationale was just shortly explained in the beginning of the study. Here we explain in more detail what existing gaps are and how this motivates the approach of our study. We think that this part is very important for readers to understand our approach, the necessity for this paper, the gap in the literature and the contributions the paper makes, since the article addresses a multidisciplinary audience and outside the field of pest management experts there is a widespread conception that reducing pesticides equals no pest control (which is wrong, since alternative pest management measures can be used, if available).

Comment 5.10

11. Lines 398-399: “identifying and providing access to effective and cost-efficient substitutes for pesticides”. It is a generic statement. Experts should have been asked to list the i) effective, ii) adoptable, iii) farmers’ compatible, and iv) cost-effective non-pesticidal pest management practices available.

Response to 5.10

Thanks, we completely agree that more detail is very valuable and important. However, we were conducting a survey on a global level - with very heterogeneous agricultural systems and potentially hundreds of specific practices, and space in the survey was limited. Moreover, we did not only want to focus on agricultural practices, but also account for measures in other domains (legal/information/economic/market) which have been shown to affect farmers pest management decisions (e.g., Finger et al., 2024). Due to the limited space we had to reduce potential answers to the eight most common categories of measures.

We again see this as a first step in a field where data on global comparisons is very limited (also see response to comment 5.4). Our goal was to assess which measure should be prioritized and if global heterogeneity exists. This is important information for relevant actors in the domain.

Our work allows researchers in the future to focus on specific measures and regions of interest with detailed questions, as you have suggested. We fully agree that this is very important and should be the next step.

In response to this comment and comment 5.4, we have now emphasized this more in the manuscript. We now write: “Studies that provide information on the choice and development of locally adapted practices and solutions will be crucial for the development of sustainable pest management. This will require farm-level surveys and research.” (ll. 517-520).

Comment 5.11

12. Many statements in the survey text are double barrel statement/questions. For example: “What extent are sustainable pest management.....as defined above already....in your area of expertise..”

How can a expert answer if one, two or three agronomic solutions are adopted?

Affordable and efficient. These are two constructs? Again, double barrel .

Environmental and health effects!.

Response to 5.11

Thanks, we agree that the clarity of the questionnaire is really important. Asking broader and more open questions was important due to the large scale of the survey that covers globally heterogeneous production systems and experts from different disciplines. As we agree that clarity of questions is an

important issue, we have taken several steps and conducted a range of robustness checks before and after the survey was conducted to ensure consistency and quality of responses. Below we first explain the general measures we have taken and then respond in detail to your comments. We hope that we are able to convince you that we took a broad range of measures and robustness checks to ensure the quality of the survey results.

General response:

First, we have taken several measures to test the questionnaire before it was sent out. The co-authors, who are from diverse regions and fields of expertise have thoroughly tested it, further we have discussed it at a workshop with all co-authors and relevant actors from different sectors in the field.

Next, we have taken a variety of measures to verify participants' perception of the questions and ensure that they align with our intended questions (see Fig. 3) for an overview. Notably, we have talked to selected participants after they had filled out the questionnaire in detail (Materials and Methods section 4) and we analyzed all comments given by respondents and ensured that we addressed them in the analysis (Supplementary Note 4). Finally, we compared results to expectations from the literature and found that they largely align in terms of levels and heterogeneity.

Detailed responses:

Regarding your first comment: As you have noted (see comment and response to comment 5.3) the listed measures largely align with important non-chemical pest control measures in important approaches such as Integrated Pest Management and Agroecological Pest Management. Implementing these approaches always requires a combination of several measures. The question therefore asks to what degree this concept (of reducing pesticide use and substituting it) is applied in the context of the experts' region and production system. The resulting variable does not have a unit per se, but can be used to compare responses of experts across regions and production systems, and control for this in the regression analysis (Fig. 2).

Regarding your second comment: Affordable and efficient is not necessarily a contradiction. This is based on the concept of opportunity costs in Economics, where farmers would choose another option for pest management if opportunity costs are lower. Opportunity costs can change with the price of a measure but also with its efficiency in protecting crops.

Regarding your third comment: Please also see the response to comment 5.10. These represent broad categories of potential measures. One potential measure that is commonly named in literature is increasing the awareness of stakeholders that pesticides might not only protect crops but also have unintended effects, e.g., on the environment and human health.

In response to your second and third comment, we now also make this clearer in the manuscript: "For all regions and production systems, we consistently find that "identifying and providing access to effective and cost-efficient substitutes for pesticides" (i.e., accounting for prices and efficiency in protecting crops) is the topmost priority for experts to support the transformation (21/100 points, SD = 0.60) followed by "education and extension on pest management" (18/100 points, SD = 0.56), "economic support" (13/100 points, SD = 0.52), "legislative support" (12/100 points, SD = 0.45) and "awareness building" (i.e., of unintended effects) (12/100 points, SD = 0.42) as the top five responses (see Fig. S8 for an overview of all responses)." (ll. 404-411).

For all of these points, we did not see any indications in our robustness checks (in the data or the comments) that respondents had troubles in perceiving the intended meaning of the questions or that respective answers diverted from expectations from literature.

In response to your comment, we further emphasized the results of the robustness checks in the revised manuscript: "Importantly, we chose a definition that allows for pesticide use due to its importance for effective pest control in certain contexts (see Methods section for a detailed

explanation how the survey was designed and its quality was tested through several robustness checks; see Supplementary Note 5 for the full survey text).” (ll. 132-136).

References

- Finger, R., Sok, J., Ahovi, E., Akter, S., Bremmer, J., Dachbrodt-Saaydeh, S., ... & Möhring, N. (2024). Towards sustainable crop protection in agriculture: A framework for research and policy. *Agricultural Systems*, 219, 104037.
- Fuchs, R., Brown, C., & Rounsevell, M. (2020). Europe’s Green Deal offshores environmental damage to other nations. *Nature*, 586, 671-673.
- Deguine, J. P., Aubertot, J. N., Flor, R. J., Lescourret, F., Wyckhuys, K. A., & Ratnadass, A. (2021). Integrated pest management: good intentions, hard realities. A review. *Agronomy for Sustainable Development*, 41(3), 38.
- Deguine, J. P., Aubertot, J. N., Bellon, S., Côte, F., Lauri, P. E., Lescourret, F., ... & Lamichhane, J. R. (2023). Agroecological crop protection for sustainable agriculture. *Advances in agronomy*, 178, 1-59.
- Oerke, E. C. (2006). Crop losses to pests. *The Journal of Agricultural Science*, 144, 31.
- Savary, S., Willocquet, L., Pethybridge, S. J., Esker, P., McRoberts, N., & Nelson, A. (2019). The global burden of pathogens and pests on major food crops. *Nature ecology & evolution*, 3(3), 430-439.
- Tang, F. H., Lenzen, M., McBratney, A., & Maggi, F. (2021). Risk of pesticide pollution at the global scale. *Nature Geoscience*, 1-5.

Response to review manuscript NCOMMS-24-32226C: “Expected Effects of a Global Transformation of Agricultural Pest Management”

We would like to thank the anonymous reviewers for providing insightful comments and suggestions for improving and balancing the manuscript. Please find detailed point-by-point responses to each comment in the respective sections below.

Please find the responses to the editorial comments in the other file

Reviewer #4

Remarks to authors of reviewer 4

Dear Authors,

Thank you for addressing my remarks, questions in my first Review step. I feel you adequately finetuned all those lines, sentences, words I questioned. With that readers will be able to better and understand (and understand equally, unambiguously) the context of your work and its key messages.

Since I provided details, highlighted the excellence, great value of your manuscript, I will not repeat it.

One point, what you mentioned several times in the corrected manuscript and in your response to the Reviewers, is indeed, your present manuscript is the first step that will and should be followed by regionalized, participatory (involvement of farmers, farming communities) further surveys, analysis. ion.

Response to remarks to authors.

Thanks again for your insightful comments that helped us to further improve the manuscript.

Reviewer #5

Thank you very much for the thorough assessment and the constructive comments on our manuscript. We are positive that we were able to address all of your comments in the revised manuscript. Please find a detailed point-by-point response below.

Remarks to the authors of reviewer 5:

Dear Authors: Thank you for your detailed response to my concerns and for clearing my doubts. It has added to my awareness-knowledge and why-to-do knowledge, regarding the direction in which pest management in agriculture is progressing. Pest management and the use of pesticides according to good agricultural practices are the most difficult aspects for farmers to master, especially if today a particular recommendation is good for pest management, tomorrow a study will find it is not so good/sustainable, thus creating a dilemma among farmers.

Response to remarks to the authors

Thanks again for the constructive remarks. We are glad that we were able to respond to your comments.

General comments reviewer 5:

Comment 5.3

Agroecological pest management/ Agroecological Crop Protection (Deguine et al. 2023) combines ecology, agroecology, Integrated Pest Management (IPM), Organic Farming (OF) and permaculture. However, I am of the firm belief that we should not get overburdened with terminology from IPM to APM/ACP. How to implement it at farm level & reduce the pesticide use, despite the introduction of low-dose pesticide, insect resistant transgenic crops and implementation of IPM, Pesticide Reduction Programme/targets?

Response to 5.3

Thanks, that is exactly the challenge we are discussing in several instances in the paper.

We agree with you that not terminology is key but the essential approaches. We therefore use the broader term of sustainable pest management and defined it in line with the basic principles of the approaches you named.

Comment 5.5

I can understand data privacy and not disclosing the names of the experts. However, institutions and universities of all the sampled experts should be given in Table S5.

Response to 5.5

Thank you for understanding the limitations from data privacy. Due to these concerns we cannot release the names of the universities, since this will directly allow to identify experts in certain cases, especially in smaller departments, where only one or two experts on the topic are employed.

We therefore chose to show all of the relevant information for study quality you could derive from the name of the institution and person (and a lot more) in an anonymized form that is in line with data privacy. More explicitly, we show the region of experts and if they work in a public institution, we show the tenure of the experts and their success in publications (h-factor and number of citations), if they received industry funding, their discipline, their scope of expertise, and their level of expertise in the different domains.

Comment 5.7

Dear authors, I think we agree that the primary objective of IPM/ Agroecological pest management/sustainable pest management is a reduction in pesticide use by volume and frequency. You have highlighted in the introduction: pesticide use and its negative direct and indirect consequences as your problem statement. *“The current pest management systems mainly rely on the use of pesticides with a global consumption of 3.7 million tons in 2023..... pesticides can adversely affect biodiversity and as well as ecosystem functioning and services. Ecosystem services such as natural pest control, insect-pollination and soil functions contribute to agricultural production. Additionally, pesticides can have adverse health effects, especially on farmers and bystanders. Finally, resistances to widely used active substances ingredients are increasing.* In S Table 3, you have used “Level- Pesticide Risk”, not “pesticide load per unit arable land, treatment index .

Response to 5.7

Thanks for your comment, we are glad that we agree on this important topic. We have now adjusted the notation and aligned it with the original wording in Tang et al. (2021).

Comment 5.8

Dear authors, I am reproducing from your response, “Experts on production systems in Europe and North America, which are very productive and operate with high material and capital input, on average, envision lower potential benefits due to sustainable pest management - across nearly all indicators. In contrast, experts in South America, Africa and Asia, on average, identified higher levels of potential benefits.” This is subjective. However, you have reported what experts from different production systems have averred to.

Response to 5.8

Thanks for your comment. In general, we have taken great care to separate findings from interpretations and have clearly marked when it is an interpretation. In order to respond to your comment, we have made this even clearer in the section. We now write:

“Experts on production systems in Europe and North America, on average, envision lower potential benefits due to sustainable pest management - across nearly all indicators. Production Systems in Europe and North America, on average, are very productive and operate with high material and capital input. In contrast, experts in South America, Africa and Asia, on average, identified higher levels of potential benefits.”

Comment 5.11

Dear authors: At this stage, it is not possible to rephrase the double barrel questions. “Environmental and health effects!. *Some pesticides, which are not environmentally hazardous do have adverse impact on human health, for example, mancozeb, belonging to the WHO hazard category U (Unlikely to present acute hazard), but has severe health effects (Probable Human Carcinogen).* So “Environment effects” and “health effects” should have been two separate questions, as was done in Sections E and F of the questionnaire. *The same is the case with “affordable” and “efficient”.*

Response to 5.11

Thanks for your comment. We agree that it is not possible to rephrase something. But as we explained, it was a deliberate choice to make broader categories, as the constraint in a survey is the time respondents have to take to answer it. We agree that there might be single cases where such categories do not always overlap. However, our broad robustness checks indicate that this was on average not a problem, and has not been brought up by respondents as a potential problem for choosing responses.

Further note that the questions you mention are not really double barreled if you take a look at them in detail.

1. “Affordable and efficient” was used in the context of assessing effects of economical, alternative solutions for sustainable pest management. In an economic sense, only affordable (but very inefficient) or only efficient (but not affordable) is not sufficient for uptake – solutions have to be both affordable and efficient (benefits higher than costs, and net returns higher than those of alternative solutions). Therefore, separating these here was not viable for us.

2. We clearly separate the environment and human health when asking about potential effects. The only instance where we mix these terminologies is in the context of potential solutions for increasing uptake of sustainable pest management (last question type in the questionnaire). We ask if increasing awareness of potential environmental and health effects could be one solution for higher uptake. We were here not interested in the separate solutions of increasing awareness on environmental and health effects, but in the general approach of improving awareness of potential negative effects of pesticides. So again, separating these here did not fit our research question.

References

Tang, F. H., Lenzen, M., McBratney, A., & Maggi, F. (2021). Risk of pesticide pollution at the global scale. *Nature Geoscience*, 1-5.

Reviewer #5

Thank you very much for the thorough assessment and the constructive comments on our manuscript. We are positive that we were able to address all of your comments in the revised manuscript. Please find a detailed point-by-point response below.

Dear Authors: Thank you for your detailed response to my concerns and for clearing my doubts. It has added to my awareness-knowledge and why-to-do knowledge, regarding the direction in which pest management in agriculture is progressing. Pest management and the use of pesticides according to good agricultural practices are the most difficult aspects for farmers to master, especially if today a particular recommendation is good for pest management, tomorrow a study will find it is not so good/sustainable, thus creating a dilemma among farmers.

General comments reviewer 5:

Comment 5.1

1. Lines 56-57: “for the five major food crops worldwide”. Which are these five major food crops?

Response to 5.1

Thanks, we are here following the definitions of the cited reference (Savary et al., 2019). These are wheat, rice, maize, potato and soybean.

In response to your comment, we have now added this information in the text: “They are estimated to vary between 26% and 40% (Oerke, 2006; based on literature and field trial data) and between 17% and 30% (Savary et al., 2019; based on expert assessments) for the five major food crops (wheat, rice, maize, potato and soybean) worldwide with a large variation across regions.” (ll. 54-57).

Reviewer Addressed

Comment 5.2

1. Lines 60-61: “2.6 million tons in 2020” The latest FAO data can be used.

Response to 5.2

Thanks, in response to your comment we have now updated this to the most recent FAO data from 2025 (for the year 2023).

**Reviewer
Addressed**

Comment 5.3

2. Lines 71-72:“the Post-2020 Global Biodiversity Framework of the United Nation sets a reduction target of 50% for overall pesticide risks until 2030”. Is this target achievable or is it meant for only academic discussions? It has been five plus decades since integrated pest management was globally adopted as the cardinal principal of plant protection strategy, yet pesticide use in agriculture has surged, and the same has been highlighted by the authors. The authors have used “sustainable pest management” terminology and not “integrated pest management”. IPM is based on the principle of “sustainable pest management”? The authors acknowledge this in the “Supplementary Information” at point 4. Although IPM endeavoured to promote sustainable pest management, thereby reducing synthetic pesticide use, it would have been better to analyse the obstacles to integrated pest management and the global transformation of IPM for sustainable agricultural production. 3. As happened with IPM, there are tens, if not hundreds, of definitions, and the same will be the fate of the concept “sustainable pest management”. Authors could have focused on IPM to conduct a survey of experts AND FARMERS on “Transformation of Agricultural Pest Management”. Sustainable pest management has been operationalised as “pest management systems with

4. no or minimal use of pesticides, using (a combination of) ... alternative pest management practices”.

Response to 5.3

Thanks, for your comment.

Regarding the first part of your comment: We totally agree that the reduction targets are ambitious, given the fact that pesticide use has been stable or increasing in important agricultural regions of the world. As you noted, we point out in the article that more action is needed in order to fulfill these targets. This is exactly the key motivation of our article: we want to support relevant actors in taking next steps for advancing these targets by providing critical information on expected effects of a transformation of pest management – this information is currently missing for important regions and indicators worldwide.

In order to further emphasize this in response to your comment, we now state “Achieving the targets requires a shift in policies” (l. 84).

Regarding the second part of your comment: Thanks a lot for your comment. As you have noted, hundreds of definitions as well as different, important paradigms and production systems for reaching reduction targets exist. We agree that Integrated Pest Management is the most prominent one. But by only focusing on Integrated Pest Management, we would have excluded other approaches that aim at a reduction of pesticide use, such as agroecological pest management, which has gained in importance in the last decades. Different approaches are strongly advocated by some researchers in the domain due to some concerns around limitations

of IPM (e.g., Deguine et al., 2021, 2023). We therefore decided to not only focus on Integrated Pest Management and use a more inclusive definition that captures the essence of these major approaches. We share your concern, and have tested the definition before the survey was conducted in the group of interdisciplinary co-authors, and in a workshop with stakeholders from different sectors (Material and Methods, section 4). Further, after the survey was conducted, we tested how respondents actually perceived this definition. Our results show that the perceptions of the respondents aligned with our intentions (see Material and Methods, section 4; Supplementary Materials, Section A4).

In order to respond to your concern, we have now added this more explicitly in the manuscript. In the revised version we now added: “We therefore refer to “Sustainable Pest Management” and not only, for example, Integrated Pest Management in order to provide a more inclusive definition that also captures other important approaches, such as agroecological pest management. We tested before and after conducting the survey that our intentions aligned with participants’ perceptions (Material and Methods section 4, Supplementary Materials, Section A4).” (ll. 561-565).

Reviewer

Agroecological pest management/ Agroecological Crop Protection (Deguine et al. 2023) combines ecology, agroecology, Integrated Pest Management (IPM), Organic Farming (OF) and permaculture. However, I am of the firm belief that we should not get overburdened with terminology from IPM to APM/ACP. How to implement it at farm level & reduce the pesticide use, despite the introduction of low-dose pesticide, insect resistant transgenic crops and implementation of IPM, Pesticide Reduction Programme/targets? Comment 5.4

5. What are alternative adoptable pest management practices available with and disseminated by survey institutes that should have been part of the survey? Thereafter, a farmers' survey should have been conducted to assess the adoptability and sustainability of these practices at the farm level across different farming systems. Without involving farmers reached by the sampled institutes and 517 experts, the exercise is top-down. What are alternative pest management technologies for farmers in different global hotspots? The practices they adopt or reject, and pesticide use, would have made this survey impactful. What attitude farmers have towards alternative pest management tactics in different regions across different crops for reducing pesticide use and making pest management sustainable could have been another study variable?

Response to 5.4

Thanks for your comment. We completely agree that this is a top-down approach and that this approach should be complemented with more detailed studies on a local and farmer level. Databases, such as EPPO (<https://gd.eppo.int/>) or the Farmer’s Toolbox for Integrated Pest Management (<https://datam.jrc.ec.europa.eu/datam/mashup/IPM/index.html>) provide a more general (but patchy) overview of available practices to farmers in different regions, and the SUPPORT project (<https://he-support.eu/>) is currently conducting a more detailed survey of IPM practices of farmers in some crops and countries of the EU. However, conducting such a study on a farm-level for all global regions and all indicators that we covered here would require an

enormous effort and large funding (and suggest this as a task for future research). Further, even data on a global and regional level that we provide here for the first time, is not available yet. We therefore see our study as a first step. We assess levels, heterogeneity and drivers of pest management and expected effects from a transformation on a global and regional level with a sample of highly qualified experts from different regions, disciplines and sectors, who have a broad overview of ongoing challenges in their region. Our insights can then be used subsequently by researchers to focus on cases that are of special interest and have a high potential. Zooming in on a more local level after our study will be key to identifying solutions. We had previously recognized this in the discussion, and in response to your comment we have further emphasized this now: “Studies that provide information on the choice and development of locally adapted practices and solutions will be crucial for the development of sustainable pest management. This will require farm-level surveys and research.” (ll. 517-520).

Please note that the experts indeed assessed current levels of sustainable pest management in the survey (see Table 2) and that we account for this variable in our analysis of potential drivers (Fig. 2).

You make an important point, and we totally agree that comparing attitudes of farmers in different regions regarding pest management practices could be an important point for future research. In response to your comment, we have now added: “Future research should consider other important characteristics of production systems, such as farm and farmer characteristics and attitudes to different types of pest management, which we could not include in our global survey.” (ll. 510-512).

Reviewer

No further comments

Comment 5.5

6. 517 experts are from the international organisations (Supplementary Table 5). What have been the outcomes of programmes implemented by these organisations in “reducing pesticide use”? Experts from National Agricultural Research Systems of countries having large cultivated land should also have been part of the sampling plan..

Response to 5.5

Thank you. This might have been a misunderstanding.

Our respondents come from a broad range of institutions, universities, extension agencies, governmental departments and industry – as you have suggested. They are just not all listed in Supplementary Table S5 due to data privacy reasons.

We explain the reasons in more detail below and have also adjusted the manuscript text in response to your comment since this was not described in sufficient clarity before.

Our sampling strategy relied on two pillars (see Supplementary Note 2):

First, we identified 454 senior experts on pest management across disciplines and professions from leading universities and research institutions worldwide (mainly from research, but also extension services, policy, and industry). We contacted them to respond to the survey and asked them to additionally forward the survey to senior members of their organizations in other regions worldwide and received 223 complete responses. Personal information on participants is strictly confidential due to ethical and legal restrictions, and we can therefore not report names or institutions of single participants. However, we provide a list of the contacted institutions in Table S5.

- ➔ In Table S5, we give an overview of institutions we contacted. However, we contacted the 454 single senior experts. Due to data privacy, we cannot disclose their names or institutions (they come from leading universities and extension agencies from all over the world). So those institutions are not in Table S5. We therefore additionally give an aggregate overview of the type of institutions, the tenure and the scientific output of respondents in Supplementary Note 2.
- ➔ Since this was not sufficiently clear in the manuscript before, we have now clarified this in response to your comment: “Personal information on participants is strictly confidential due to ethical and legal restrictions, and we can therefore not report participant names. However, we provide a list of the contacted institutions in Table S5. Note that Table S5 gives an overview of contacted institutions but not of the universities and extension agencies the identified senior experts belonged to (excluded due to data privacy).” (Supp. Materials ll. 74-78).

Second, we identified international and national academic experts through the literature database Scopus. More specifically, we contacted all corresponding authors of articles published in peer-reviewed journals on the subject of “sustainable pest management” over the last 10 years. We chose “sustainable pest management” as a key word, as our clear research focus is on the assessment of potential effects from a transition to sustainable pest management. Double entries were then discarded, leaving us with 1531 contacts in total. Out of the 1531 identified corresponding authors, we received 294 complete responses, amounting to a response rate of 19.3%. This strategy and response rate is in line with recent sampling strategies of researchers conducting global expert assessments in adjacent fields, such as Economics (e.g., 4). We provide information about the type of institutions, the tenure and the scientific output of corresponding authors, as well as indicated funding sources.

Authors ‘response

“In Table S5, we give an overview of institutions we contacted. However, we contacted the 454 single senior experts. Due to data privacy, we cannot disclose their names or institutions (they come from leading universities and extension agencies from all over the world). So those institutions are not in Table S5. We therefore additionally give an aggregate overview of the type of institutions, the tenure and the scientific output of respondents in Supplementary Note 2.”

Reviewer

I can understand data privacy and not disclosing the names of the experts. However, institutions and universities of all the sampled experts should be given in Table S5.

Comment 5.6

7. This article is simply an academic exercise. We cannot test hypotheses solely based on the survey of 517 experts. Experts' and farmers' data triangulation would have added to the robustness of the results for drawing inferences..

Response to 5.6

Thanks. We see our study as a first step. We assess levels, heterogeneity and drivers of pest management and expected effects from a transformation on a global and regional level with a sample of highly qualified experts from different regions, disciplines and sectors, who have a broad overview of ongoing challenges in their region. Our insights can then be used subsequently by researchers to focus on cases that are of special interest and have a high potential. Zooming in on a more local level after our study will be key to identifying solutions.

In response to your comment, we have now made this clearer in the manuscript: “Studies that provide information on the choice and development of locally adapted practices and solutions will be crucial for the development of sustainable pest management. This will require farm-level surveys and research.” (ll. 517-520).

(Please also see our response to comment 5.4).

Reviewer

No further comments

Comment 5.7

8. Data on “pesticide use and pests” should have been added from the selected countries of the 517 experts, which would add value to the article.

Response to 5.7

Thanks, you are totally correct, pesticide use and pests are of course very important potential drivers – and this is why we actually account for both in our regression analysis.

For pesticide use we even use a risk indicator that accounts not only for pesticide use volume but also for the environmental hazards of used pesticides (which can be very heterogeneous) → This is the variable “Level Pesticide Risk” used in our regression analysis – it is based on the “Risk Score” from Tang et al. (2021).

For pests, we use regional estimates on pest pressure → This is the variable “Potential Pest Pressure” in our regression analysis, based on the estimates of Oerke (2006).

You can find a description of the variables in Table 2. Since we have noted that the variable description in the regression analysis and in the Table did not use the same names it is clear that this led to confusion. In response to your comment, we have therefore aligned variable names now.

Further, since it has not been clear enough that we account for pesticide use and pest pressure we now explicitly highlight this, in response to your comment, in the manuscript text: “To assess the factors that drive the expected effects of the transformation, we spatially matched the survey data with a wide set of country- and regional-level data on key characteristics of agricultural production systems (see the Methods section). We then analyzed the data using multiple regression analyses (see Fig. 2 and Supplementary Table 3 for detailed regression results). This allows us to control for key characteristics of the production system (e.g., level of pesticide risk, pest pressure).” (ll. 308-312).

Reviewer

Dear authors, I think we agree that the primary objective of IPM/ Agroecological pest management/sustainable pest management is a reduction in pesticide use by volume and frequency. You have highlighted in the introduction: pesticide use and its negative direct and indirect consequences as your problem statement. “The current pest management systems mainly rely on the use of pesticides with a global consumption of 3.7 million tons in 2023..... pesticides can adversely affect biodiversity and as well as ecosystem functioning and services. Ecosystem services such as natural pest control, insect-pollination and soil functions contribute to agricultural production. Additionally, pesticides can have adverse health effects, especially on farmers and bystanders. Finally, resistances to widely used active substances ingredients are increasing. In S Table 3, you have used “Level- Pesticide Risk”, not “pesticide load per unit arable land, treatment index .”

Comment 5.8

9. Highest potential of sustainable pest management in low-income regions: Why “Experts on highly productive and input-intensive production systems in Europe and North America envision lower potential benefits due to sustainable pest management -across nearly all indicators?” Many times, experts are far removed from the farms/ farm households. Experts mostly do not collect primary data themselves in the developing countries, so how can they conclude “..... potential win-win scenarios due to the reduction of pesticide use and the transformation of pest management” AND WHY EXPERTS FROM INDUSTRIALIZED AGRICULTURE SEE LOWER POTENTIAL. What about the input-intensive production systems in many Green Revolution areas of developing countries, for example, Indian Punjab and Haryana, which are highly productive?.

Response to 5.8

Thanks, we here aggregate results on a continent level, since this allows us to compare general trends in different regions of the world within the limited space of the article.

But we totally agree with you that different continents can be very heterogeneous with regards to their agricultural systems and that such a broad aggregation hides a lot of this heterogeneity.

We further agree with you that this heterogeneity should be better highlighted in the article. In response to your comment, we have therefore created a new Figure (Supplementary Figure 14) which shows that expected effects within regions are very heterogeneous.

In response to your comment, we now further highlight this in the manuscript at different points:

First, we highlight that the regional results are averages (and can therefore hide heterogeneity):

“Experts on production systems in Europe and North America, which are very productive and operate with high material and capital input, on average, envision lower potential benefits due to sustainable pest management - across nearly all indicators. In contrast, experts in South America, Africa and Asia, on average, identified higher levels of potential benefits.” (ll. 246-251).

Then we now additionally, explicitly state that these regional averages hide a lot of heterogeneity in expected effects and refer to the additional figure that we have created in response to your comment: “Further note that the averages are aggregated on a large regional scale and that important heterogeneity within regions exist (Supplementary Fig. 14). This result highlights the importance of looking into potential drivers of expected effects.” (ll. 278-281).

In order to account for this heterogeneity, we conduct the regression analysis (Fig. 2), which accounts for characteristics of production systems on a finer scale.

Reviewer 5.8

Dear authors, I am reproducing from your response, “Experts on production systems in Europe and North America, which are very productive and operate with high material and capital input, on average, envision lower potential benefits due to sustainable pest management - across nearly all indicators. In contrast, experts in South America, Africa and Asia, on average, identified higher levels of potential benefits.” This is subjective. However, you have reported what experts from different production systems have averred to. Comment 5.9

10. Lines 162-170: In the results section, the authors are again explaining the rationale for the study. The results section should exclusively focus on survey results..

Response to 5.9

Thanks, in general we agree with your recommendation. However, the rationale was just shortly explained in the beginning of the study. Here we explain in more detail what existing gaps are and how this motivates the approach of our study. We think that this part is very important for readers to understand our approach, the necessity for this paper, the gap in the literature and the contributions the paper makes, since the article addresses a multidisciplinary audience and outside the field of pest management experts there is a widespread conception that reducing pesticides equals no pest control (which is wrong, since alternative pest management measures can be used, if available).

Reviewer

Okay

Comment 5.10

11. Lines 398-399: “identifying and providing access to effective and cost-efficient substitutes for pesticides”. It is a generic statement. Experts should have been asked to list the i) effective, ii) adoptable, iii) farmers’ compatible, and iv) cost-effective non-pesticidal pest management practices available.

Response to 5.10

Thanks, we completely agree that more detail is very valuable and important. However, we were conducting a survey on a global level - with very heterogeneous agricultural systems and potentially hundreds of specific practices, and space in the survey was limited. Moreover, we did not only want to focus on agricultural practices, but also account for measures in other domains (legal/information/economic/market) which have been shown to affect farmers pest management decisions (e.g., Finger et al., 2024). Due to the limited space we had to reduce potential answers to the eight most common categories of measures.

We again see this as a first step in a field where data on global comparisons is very limited (also see response to comment 5.4). Our goal was to assess which measure should be prioritized and if global heterogeneity exists. This is important information for relevant actors in the domain.

Our work allows researchers in the future to focus on specific measures and regions of interest with detailed questions, as you have suggested. We fully agree that this is very important and should be the next step.

In response to this comment and comment 5.4, we have now emphasized this more in the manuscript. We now write: “Studies that provide information on the choice and development of locally adapted practices and solutions will be crucial for the development of sustainable pest management. This will require farm-level surveys and research.” (ll. 517-520).

Reviewer

Agree with the authors

Comment 5.11

12. Many statements in the survey text are double barrel statement/questions. For example: “What extent are sustainable pest management.....as defined above already....in your area of expertise..”

How can a expert answer if one, two or three agronomic solutions are adopted?

Affordable and efficient. These are two constructs? Again, double barrel .

Environmental and health effects!.

Response to 5.11

Thanks, we agree that the clarity of the questionnaire is really important. Asking broader and more open questions was important due to the large scale of the survey that covers globally heterogeneous production systems and experts from different disciplines. As we agree that clarity of questions is an important issue, we have taken several steps and conducted a range of robustness checks before and after the survey was conducted to ensure consistency and quality of responses. Below we first explain the general measures we have taken and then respond in detail to your comments. We hope that we are able to convince you that we took a broad range of measures and robustness checks to ensure the quality of the survey results.

Affordable and efficient. These are two constructs? Again, double barrel .

Environmental and health effects!.

Reviewer

Dear authors: At this stage, it is not possible to rephrase the double barrel questions. “Environmental and health effects!. *Some pesticides, which are not environmentally hazardous do have adverse impact on human health, for example, mancozeb, belonging to the WHO hazard category U (Unlikely to present acute hazard), but has severe health effects (Probable Human Carcinogen).* So “Environment effects” and “health effects” should have been two separate questions, as was done in Sections E and F of the questionnaire. *The same is the case with” affordable” and “efficient”.*

General response:

First, we have taken several measures to test the questionnaire before it was sent out. The co-authors, who are from diverse regions and fields of expertise have thoroughly tested it, further we have discussed it at a workshop with all co-authors and relevant actors from different sectors in the field.

Next, we have taken a variety of measures to verify participants’ perception of the questions and ensure that they align with our intended questions (see Fig. 3) for an overview. Notably, we have talked to selected participants after they had filled out the questionnaire in detail (Materials and Methods section 4) and we analyzed all comments given by respondents and ensured that we addressed them in the analysis (Supplementary Note 4). Finally, we compared results to expectations from the literature and found that they largely align in terms of levels and heterogeneity.

Detailed responses:

Regarding your first comment: As you have noted (see comment and response to comment 5.3) the listed measures largely align with important non-chemical pest control measures in important approaches such as Integrated Pest Management and Agroecological Pest Management.

Implementing these approaches always requires a combination of several measures. The question therefore asks to what degree this concept (of reducing pesticide use and substituting it) is applied in the context of the experts' region and production system. The resulting variable does not have a unit per se, but can be used to compare responses of experts across regions and production systems, and control for this in the regression analysis (Fig. 2).

Regarding your second comment: Affordable and efficient is not necessarily a contradiction. This is based on the concept of opportunity costs in Economics, where farmers would choose another option for pest management if opportunity costs are lower. Opportunity costs can change with the price of a measure but also with its efficiency in protecting crops.

Regarding your third comment: Please also see the response to comment 5.10. These represent broad categories of potential measures. One potential measure that is commonly named in literature is increasing the awareness of stakeholders that pesticides might not only protect crops but also have unintended effects, e.g., on the environment and human health.

In response to your second and third comment, we now also make this clearer in the manuscript: "For all regions and production systems, we consistently find that "identifying and providing access to effective and cost-efficient substitutes for pesticides" (i.e., accounting for prices and efficiency in protecting crops) is the topmost priority for experts to support the transformation (21/100 points, SD = 0.60) followed by "education and extension on pest management" (18/100 points, SD = 0.56), "economic support" (13/100 points, SD = 0.52), "legislative support" (12/100 points, SD = 0.45) and "awareness building" (i.e., of unintended effects) (12/100 points, SD = 0.42) as the top five responses (see Fig. S8 for an overview of all responses)." (ll. 404-411).

For all of these points, we did not see any indications in our robustness checks (in the data or the comments) that respondents had troubles in perceiving the intended meaning of the questions or that respective answers diverted from expectations from literature.

In response to your comment, we further emphasized the results of the robustness checks in the revised manuscript: "Importantly, we chose a definition that allows for pesticide use due to its importance for effective pest control in certain contexts (see Methods section for a detailed explanation how the survey was designed and its quality was tested through several robustness checks; see Supplementary Note 5 for the full survey text)." (ll. 132-136).